# Robust Optimization for Mitigating Reward Hacking with Correlated Proxies

**Zixuan Liu, Xiaolin Sun & Zizhan Zheng**
Department of Computer Science
Tulane University
New Orleans, LA 70118, USA
`{zliu41,xsun12,zzheng3}@tulane.edu`

## Abstract

Designing robust reinforcement learning (RL) agents in the presence of imperfect reward signals remains a core challenge. In practice, agents are often trained with proxy rewards that only approximate the true objective, leaving them vulnerable to reward hacking, where high proxy returns arise from unintended or exploitative behaviors. Recent work formalizes this issue using $r$-correlation between proxy and true rewards, but existing methods like occupancy-regularized policy optimization (ORPO) optimize against a fixed proxy and do not provide strong guarantees against broader classes of correlated proxies. In this work, we formulate reward hacking as a robust policy optimization problem over the space of all $r$-correlated proxy rewards. We derive a tractable max-min formulation, where the agent maximizes performance under the worst-case proxy consistent with the correlation constraint. We further show that when the reward is a linear function of known features, our approach can be adapted to incorporate this prior knowledge, yielding both improved policies and interpretable worst-case rewards. Experiments across several environments show that our algorithms consistently outperform ORPO in worst-case returns, and offer improved robustness and stability across different levels of proxy-true reward correlation. These results show that our approach provides both robustness and transparency in settings where reward design is inherently uncertain. The code is available at `https://github.com/ZixuanLiu4869/reward_hacking`.

## 1 Introduction

Real-world reinforcement learning (RL) systems often struggle with reward specification: it is notoriously difficult to craft a reward function that perfectly captures the intended goals in all scenarios (Amodei et al., 2016; Ibarz et al., 2018; Stray et al., 2024). In practice, designers rely on proxy rewards that approximate the true objective (Tien et al., 2023). However, agents optimizing these imperfect proxies can lead to unintended exploitative behaviors, achieving high proxy returns while yielding poor true outcomes, a phenomenon known as reward hacking (Leike et al., 2017; Everitt et al., 2017; 2021; Koch et al., 2021). Such reward hacking behaviors are not merely hypothetical; they have led to undesirable or even catastrophic consequences in safety-critical settings (e.g., autonomous driving) (Krakovna et al., 2018; Knox et al., 2023) and are alarmingly common in real-world deployments (Kleinberg et al., 2024; Franchi et al., 2023; Milli et al., 2021; Obermeyer et al., 2019). Beyond reward hacking, interpretability and transparency of RL policies are increasingly recognized as critical requirements for real-world acceptance (Vouros, 2022; Puiutta & Veith, 2020; Iyer et al., 2018). Policymakers and practitioners in safety-critical domains require systems not only to be robust but also interpretable; they must understand which specific decision-making criteria lead to undesirable outcomes to effectively mitigate risks and ensure compliance with safety regulations (Rudin, 2019; Druce et al., 2021; Doshi-Velez & Kim, 2017). These challenges highlight the need for RL algorithms to address two fundamental challenges: robustness to uncertain or poorly-specified rewards, and interpretability to facilitate oversight and compliance by human stakeholders, especially in high-stakes, real-world environments like traffic control (Vinitsky et al.,

2018), healthcare decision-making (Fox et al., 2020; Holzinger et al., 2017), and pandemic response strategies (Kompella et al., 2020).

Recent work has begun to formalize reward hacking and develop principled mitigations. (Laidlaw et al., 2025) define a proxy reward to be *r-correlated* with the true reward if it maintains a correlation coefficient $r > 0$ on state-action pairs encountered by a certain reference policy. Notably, their definition permits the proxy and true reward to diverge arbitrarily in parts of the state-action space not visited by the reference policy, precisely the regions an RL agent might exploit under intensive optimization. Using this framework, reward hacking is formalized as the situation in which optimizing an $r$-correlated proxy yields a policy with lower true reward than that of the reference policy. Building on this definition, they propose Occupancy-Regularized Policy Optimization (ORPO) as a mitigation strategy. ORPO augments the standard RL objective with a regularization term that penalizes deviations between the learned policy's occupancy measure and that of the reference policy.

Despite significant progress, existing solutions to reward hacking show several limitations. First, their effectiveness relies heavily on the choice of the specific proxy reward. However, designing perfect proxies is challenging, and in real-world scenarios, reward proxies are often derived heuristically or empirically from noisy or limited data (Jeon et al., 2020; Sadigh et al., 2017), leading to uncertainty or variability in the exact correlation with true rewards. Therefore, robustness to variations in proxy rewards is crucial for dependable deployment. While the regularization method used by ORPO provides a lower bound on improvement in true reward, its guarantee on the worst-case performance against an adversarially chosen proxy is weak. Second, current methods like ORPO typically treat a reward function as a black box and learn a complex policy with no easily interpretable structure, making it hard to understand why the resulting policy avoids reward hacking or to trust its behavior in novel situations. Further, they cannot be easily adapted to incorporate prior knowledge of the true reward. These shortcomings underscore the need for a more robust and transparent approach to reward hacking in RL.

In this work, we formalize reward hacking as a robust RL problem under proxy reward uncertainty and develop new algorithms to address the above gaps. The key idea is to optimize against an adversarial proxy reward rather than trusting a single proxy. We assume the true reward could be any function that remains $r$-correlated with the proxy, and we train the agent to perform well against the worst-case such proxy. This approach explicitly accounts for uncertainty in proxy design and guards against unintended exploitative behaviors. Concretely, we propose a max-min formulation in which the policy chooses its strategy to maximize its guaranteed true return while an adversary minimizes the true return by selecting a reward function from the set of all $r$-correlated proxies. By solving this problem, the agent learns a policy that is robust to all plausible deviations of the proxy reward within the correlation bound. We derive a closed-form solution for the adversary's worst-case reward assignment given any candidate policy, which allows efficient evaluation of the inner minimization and provides insight into how proxy reward flaws are most damaging. Building on this result, we introduce a practical algorithm for Max-Min Policy Optimization that iteratively updates the policy against this worst-case reward signal. Moreover, to improve the tractability and transparency of the inner optimization, we introduce a Linear Max-Min variant of our method. In this variant, we assume the true reward lies in a class of linear functions over known features, an assumption that has been extensively studied in prior work on successor representations and successor features (Dayan, 1993; Barreto et al., 2017; 2018), and which allows us to characterize the worst-case proxy reward as a sparse linear combination of those features. While the policy itself remains parameterized by general neural networks, the learned worst-case reward function becomes interpretable in terms of its feature weights. This provides insight into which aspects of the proxy reward space the policy is robust to or vulnerable against, making it valuable for applications where understanding the failure modes of the reward design is important.

Finally, we empirically evaluate the proposed approaches on several challenging environments. Across all domains, our Max-Min and Linear Max-Min policies outperform ORPO in terms of worst-case reward, indicating substantially improved robustness. Moreover, under a large range of proxy-true correlation scenarios, our methods exhibit higher average reward and lower variance compared to ORPO, meaning the performance of our policies remains more consistent and reliable. These findings demonstrate the practical significance of our robust formulation, paving the way for safer and more trustworthy RL deployment in real-world applications.

Our main contributions can be summarized as follows: 1) We propose a novel robust RL formulation that explicitly models reward hacking as a max-min optimization problem over proxy rewards constrained by correlation with the true rewards. 2) We develop a practical algorithm for the max-min problem, which is further extended to linear rewards with improved robustness and interpretability. 3) We provide a theoretical convergence guarantee for the max-min objective with a sample-complexity bound for the occupancy estimation. We also show that accurate occupancy estimation is pivotal for robustness. 4) Experiment results demonstrate improved robustness and worst-case rewards across five real-world inspired reward hacking environments.

## 2 PRELIMINARIES

**Reinforcement Learning.** A reinforcement learning (RL) problem can be formulated as an infinite-horizon Markov Decision Process (MDP) defined by the tuple $(\mathcal{S}, \mathcal{A}, p, \mu_0, R, \gamma)$, where $\mathcal{S}$ and $\mathcal{A}$ denote the state and action spaces, $p(s' \mid s, a)$ is the transition probability from state $s$ to $s'$ given action $a$, $\mu_0$ is the initial state distribution and $\gamma \in [0, 1)$ is the discounted factor. The agent interacts with the environment over discrete time steps $t = 0, 1, 2, \ldots$. At each time step, it selects an action $a_t \in \mathcal{A}$ based on the current state $s_t \in \mathcal{S}$ according to a policy $\pi(a \mid s)$, which defines a distribution over actions conditioned on the state. Upon taking action $a_t$, the agent receives a reward $R(s_t, a_t) \in \mathbb{R}$ and transitions to the next state $s_{t+1}$ according to $p(s_{t+1} \mid s_t, a_t)$. The goal of the agent is to maximize the expected cumulative discounted return:

$$J(\pi, R) = (1 - \gamma) \mathbb{E}_\pi \left[ \sum_{t=0}^\infty \gamma^t R(s_t, a_t) \right],$$ (1)

where $\gamma \in [0, 1)$ is the discount factor, and the expectation is taken over trajectories generated by following policy $\pi$. We define the *state-action occupancy measure* $\mu_\pi$ of a policy $\pi$ as: $\mu_\pi(s, a) = (1 - \gamma) \mathbb{E}_\pi \left[ \sum_{t=0}^\infty \gamma^t \mathbb{I}\{s_t = s, a_t = a\} \right]$, which represents the discounted visitation frequency of each state-action pair under policy $\pi$. Using the occupancy measure, the return can be equivalently expressed as: $J(\pi, R) = \mathbb{E}_{(s,a) \sim \mu_\pi}[R(s, a)]$.

**Correlated Proxies and Reward Hacking.** Below we give an overview of the recently proposed $r$-correlated proxy framework proposed in (Laidlaw et al., 2025) for detecting and mitigating reward hacking, which our work is built upon. A detailed discussion of related work on reward hacking and robust RL is given in Appendix D. In particular, they consider a setting where the agent is given a reference policy $\pi_{\text{ref}}$ and a proxy reward $R_{\text{proxy}}$, while the true reward is hidden. They further assume that the proxy reward is $r$-*correlated* with the true reward under the reference policy, that is:

$$\mathbb{E}_{\mu_{\pi_{\text{ref}}}} \left[ \left( \frac{R_{\text{proxy}} - J(\pi_{\text{ref}}, R_{\text{proxy}})}{\sigma_{R_{\text{proxy}}}} \right) \left( \frac{R_{\text{true}} - J(\pi_{\text{ref}}, R_{\text{true}})}{\sigma_{R_{\text{true}}}} \right) \right] = r,$$ (2)

where $\sigma_{R_{\text{proxy}}}^2 = \mathbb{E}_{\mu_{\pi_{\text{ref}}}} \left[ (R_{\text{proxy}} - J(\pi_{\text{ref}}, R_{\text{proxy}}))^2 \right]$ and $\sigma_{R_{\text{true}}}^2 = \mathbb{E}_{\mu_{\pi_{\text{ref}}}} \left[ (R_{\text{true}} - J(\pi_{\text{ref}}, R_{\text{true}}))^2 \right]$ are the variances of the proxy and true rewards, respectively, under the reference policy. Reward hacking is said to occur when a policy $\pi$ optimized for an $r$-correlated proxy reward achieves lower true reward than the reference policy: $J(\pi, R_{\text{true}}) < J(\pi_{\text{ref}}, R_{\text{true}})$. To mitigate reward hacking, they propose Occupancy-Regularized Policy Optimization (ORPO) to optimize a regularized policy objective given below, which is shown to provide a lower bound on improvement in true reward:

$$\max_\pi J(\pi, R_{\text{proxy}}) - \lambda \sqrt{\chi^2(\mu_\pi \| \mu_{\pi_{\text{ref}}})},$$ (3)

where $\chi^2(\mu_\pi \| \mu_{\pi_{\text{ref}}})$ denotes the $\chi^2$-squared divergence between the occupancy measures of $\pi$ and $\pi_{\text{ref}}$, and the regularization strength $\lambda = \sigma_{R_{\text{proxy}}} \sqrt{1 - r^2}$. This encourages the learned policy to stay close to the reference distribution when the proxy reward is weakly correlated with the true reward.

## 3 METHOD

In this section, we discuss our robust policy optimization approach for mitigating reward hacking. In contrast to regularization-based methods such as ORPO, we consider a max-min formulation that identifies a robust policy with respect to the worst-case reward across all reward functions that

are $r$-correlated with the proxy reward. We further extend our framework to settings where the reward function is a linear combination of known features with unknown weights. Our approach effectively leverages this structural information, when known a priori, to improve both robustness and interpretability, a task that is particularly challenging for regularization-based techniques.

## 3.1 MAX-MIN POLICY OPTIMIZATION

Similar to ORPO, we assume that the agent is given a proxy reward $R_{\text{proxy}}$ and a reference policy $\pi_{\text{ref}}$, while the true reward is hidden. Rather than regularizing the policy under a fixed proxy reward, we consider the *entire space of rewards* $\mathcal{R}_{\text{corr}}$ that satisfy the correlation constraint with respect to a known proxy reward, as defined in Equation 4:

$$\mathcal{R}_{\text{corr}} = \left\{ R : (s,a) \to \mathbb{R} \,\middle|\, \mathbb{E}_{\mu_{\pi_{\text{ref}}}}\left[\frac{R-M}{V} \cdot R_{\text{proxy}}\right] = r, \, J(\pi_{\text{ref}}, R) = M, \, \sigma_R^2 = V^2 \right\}. \quad (4)$$

$M$ and $V$ denote the fixed mean and standard deviation of the reward function $R$ under the reference policy $\pi_{\text{ref}}$. For simplicity, we define $R_{\text{proxy}}$ to be the normalized proxy reward $R_{\text{proxy}}(s,a) := \frac{\tilde{R}_{\text{proxy}}(s,a) - J(\pi_{\text{ref}}, \tilde{R}_{\text{proxy}})}{\sigma_{\tilde{R}_{\text{proxy}}}}$, where $\tilde{R}_{\text{proxy}}$ is the original (unnormalized) proxy reward. After normalization, we have $J(\pi_{\text{ref}}, R_{\text{proxy}}) = 0$ and $\text{Var}_{\mu_{\pi_{\text{ref}}}}(R_{\text{proxy}}) = 1$, which simplifies the correlation constraint in Equation 4. The hyperparameter $r$ controls the degree of alignment between the proxy and true reward. It allows us to interpolate between strong robustness (small $r$) and high proxy fidelity (large $r$), enabling a principled robustness-accuracy trade-off. We remark that it is without loss of generality to consider fixed $M$ and $V$, which we will further elaborate on later.

We propose a *worst-case optimization framework* where the policy is trained to maximize expected performance under the least favorable reward within $\mathcal{R}_{\text{corr}}$. Assuming that the true reward lies somewhere within this set, this approach improves robustness by ensuring that the policy does not overfit to any single optimistic interpretation of the proxy reward. Formally, the objective becomes:

$$\max_{\pi} \min_{R \in \mathcal{R}_{\text{corr}}} J(\pi, R) = \max_{\pi} \min_{R \in \mathcal{R}_{\text{corr}}} \mathbb{E}_{(s,a) \sim \mu_{\pi}}[R(s,a)]. \quad (5)$$

However, a challenge arises: the objective $\mathbb{E}_{\mu_{\pi}}[R(s,a)]$ depends on the state-action occupancy $\mu_{\pi}$, whereas the constraints defining $\mathcal{R}_{\text{corr}}$ are expressed in terms of $\mu_{\pi_{\text{ref}}}$. This mismatch complicates direct optimization. To resolve this, we apply a *change-of-measure* technique (Hu & Hong, 2013; Lam, 2016) to rewrite the expectation under $\mu_{\pi_{\text{ref}}}$. Specifically, let $L(s,a)$ denote the Radon-Nikodym derivative: $L(s,a) = \frac{\mu_{\pi}(s,a)}{\mu_{\pi_{\text{ref}}}(s,a)}$. By definition, $L(s,a) \geq 0$ and $\mathbb{E}_{\mu_{\pi_{\text{ref}}}}[L(s,a)] = 1$. Applying the change-of-measure formula, we can express the return as: $\mathbb{E}_{\mu_{\pi}}[R(s,a)] = \int_{\mathcal{S} \times \mathcal{A}} \mu_{\pi}(s,a)R(s,a)\,d(s,a) = \int_{\mathcal{S} \times \mathcal{A}} \mu_{\pi_{\text{ref}}}(s,a)\frac{\mu_{\pi}(s,a)}{\mu_{\pi_{\text{ref}}}(s,a)}R(s,a)\,d(s,a) = \mathbb{E}_{\mu_{\pi_{\text{ref}}}}[L(s,a)R(s,a)]$. Thus, both the objective and the constraints can be rewritten as expectations with respect to $\mu_{\pi_{\text{ref}}}$.

For notational simplicity, we will suppress variables $(s,a)$ and write for example, $L$ as $L(s,a)$. Under this reparameterization, the inner minimization in Equation 5 can be reformulated as:

$$\min_{R \in \mathcal{R}_{\text{corr}}} \mathbb{E}_{\mu_{\pi_{\text{ref}}}}[L \cdot R]. \quad (6)$$

Although the feasible set in Problem 6 is not convex due to the equality constraint on the variance, we still derive an optimal solution using a Lagrangian formulation. Our approach leverages tools from duality theory, commonly used in robust optimization (Delage & Ye, 2010; Goh & Sim, 2010). We further justify the validity of our solution in Appendix E.2. Specifically, the Lagrangian functional associated with this problem is defined as: $l_0(\lambda_1, \lambda_2, \lambda_3, R) = \mathbb{E}_{\mu_{\pi_{\text{ref}}}}[L \cdot R - \lambda_1 \frac{R-M}{V} \cdot R_{\text{proxy}} - \lambda_2 R - \lambda_3 R^2] + \lambda_1 r + \lambda_2 M + \lambda_3 (M^2 + V^2)$, where $\lambda_1, \lambda_2, \lambda_3$ are the Lagrange multipliers corresponding to the correlation constraint, mean constraint, and variance constraint, respectively. Then the original problem in Equation 6 is equivalent to the following problem:

$$\max_{\lambda_1, \lambda_2, \lambda_3} \min_{R \in \mathcal{R}_{\text{corr}}} l_0(\lambda_1, \lambda_2, \lambda_3, R). \quad (7)$$

We now solve the inner minimization problem in Equation 7 by finding the optimal $R$ for fixed dual variables $(\lambda_1, \lambda_2, \lambda_3)$. Taking the functional derivative of the Lagrangian $l_0$ with respect to $R(s,a)$ gives: $\frac{\partial l_0}{\partial R} = \mu_{\pi_{\text{ref}}}(s,a)[(L - \lambda_1 \frac{R_{\text{proxy}}}{V} - \lambda_2) - 2\lambda_3 R]$. When $\mu_{\pi_{\text{ref}}}(s,a) > 0$, setting the derivative

of the Lagrangian to zero yields the optimal adversarial reward function:

$$R^*(s,a) = \frac{L(s,a) - \lambda_1 \frac{R_{\text{proxy}}}{V} - \lambda_2}{2\lambda_3}. \tag{8}$$

However, for state-action pairs where $\mu_{\pi_{\text{ref}}}(s,a) = 0$, i.e., those not visited under the reference policy, the correlation and moment constraints become vacuous. In these regions, the adversarial reward $R^*(s,a)$ can be driven arbitrarily poor, reflecting that no constraint prevents the adversary from assigning highly penalizing values to rarely visited or unobserved state-action pairs. Nevertheless, consider the case where $\mu_{\pi_{\text{ref}}}(s,a) > 0$, we can substitute the optimal $R^*$ from Equation 8 into the Lagrangian $l_0$ and get the dual objective. After some process detailed in Appendix E.1, we get the optimal solution to problem (6), so the original max-min problem (5) reduces to:

$$\max_{\pi} \ r \cdot V \cdot \mathbb{E}_{\mu_\pi}[R_{\text{proxy}}] - V \cdot \sqrt{1-r^2}\sqrt{\chi^2(\mu_\pi \,\|\, \mu_{\pi_{\text{ref}}}) - \mathbb{E}^2_{\mu_\pi}[R_{\text{proxy}}]} + M. \tag{9}$$

Thus, the final policy optimization objective becomes maximizing the proxy reward, regularized by a penalty that depends on the distributional shift between $\mu_\pi$ and $\mu_{\pi_{\text{ref}}}$ and the expectation of the current policy under proxy reward $\mathbb{E}_{\mu_\pi}[R_{\text{proxy}}]$, and the correlation strength $r$. We observe that the constants $M$ and $V$ do not affect the optimal policy: while they influence the absolute value of the worst-case reward for a given policy $\pi$, they only apply a linear transformation (scaling by $V$ and shifting by $M$) and do not change the relative ordering of policies. Therefore, for simplicity, we set $V = 1$ and $M = 0$ in our implementation. This also provides a fair way to compare the worst-case rewards of different policies. Notice that the optimization objective in Equation 9 closely resembles the ORPO objective proposed in Equation 3. However, there are two key differences: (1) our regularization strength is $\frac{\sqrt{1-r^2}}{r}$ instead of $\sigma_{R_{\text{proxy}}}\sqrt{1-r^2}$, and (2) our penalty term is $\chi^2(\mu_\pi \,\|\, \mu_{\pi_{\text{ref}}}) - \mathbb{E}^2_{\mu_\pi}[R_{\text{proxy}}]$ rather than simply $\chi^2(\mu_\pi \,\|\, \mu_{\pi_{\text{ref}}})$. The proof that $\chi^2(\mu_\pi \,\|\, \mu_{\pi_{\text{ref}}}) - \mathbb{E}^2_{\mu_\pi}[R_{\text{proxy}}] \geq 0$ holds can be found in Appendix E.3. A detailed comparison between our policy gradient and that of ORPO is provided in Appendix E.8.

To further illustrate how our framework in Equation 9 helps prevent reward hacking, i.e., how optimizing a proxy reward can translate into an improvement in the true reward over the reference policy, as discussed in Section 2, we formalize the following theorem:

**Theorem 1.** *Suppose that the true reward function $R_{true}$ lies in the correlation-constrained uncertainty set $\mathcal{R}_{corr}$. Then, for any policy $\pi$ such that $\mu_\pi \ll \mu_{\pi_{ref}}$ (i.e., $\mu_{\pi_{ref}}(s,a) = 0 \Rightarrow \mu_\pi(s,a) = 0$), we have*

$$J(\pi, R_{true}) - J(\pi_{ref}, R_{true}) \ \geq \ r \cdot \mathbb{E}_{\mu_\pi}[R_{proxy}] - \sqrt{1-r^2}\sqrt{\chi^2(\mu_\pi \,\|\, \mu_{\pi_{ref}}) - \mathbb{E}^2_{\mu_\pi}[R_{proxy}]}.$$

Proof can be found in Appendix E.9. From Theorem 1, we see that our objective optimizes a pessimistic lower bound on the true improvement over the reference policy. By Definition 4.2 in (Laidlaw et al., 2025), reward hacking occurs when $J(\pi, R_{\text{true}}) < J(\pi_{\text{ref}}, R_{\text{true}})$. Although this is precisely the quantity we would like to maximize, we cannot do so directly because the true reward is unobserved, and therefore we must instead optimize the max–min objective in Equation 9. Theorem 1 shows that our objective is always lower than (but anchored to) the true improvement, which explains why our framework can promote robustness against potential reward hacking: improving our surrogate objective necessarily improves a conservative bound on $J(\pi, R_{\text{true}}) - J(\pi_{\text{ref}}, R_{\text{true}})$.

**Remark:** Our optimization problem in Equation 5 is standard in distributionally robust optimization (DRO). However, it remains underexplored in the context of RL, with only one relevant work that considers uncertainty sets based on the first and second moments of the reward distribution (Nguyen et al., 2022). While their formulation appears similar, their results are not directly applicable to our max-min framework, and we still need to explicitly solve our formulation. We also note that under certain assumptions, the ORPO objective (Equation 3) can be reinterpreted as a special case of the max-min formulation in (Nguyen et al., 2022) (Theorem 1), providing a complementary view of the connection between these approaches. Nevertheless, our optimization objective remains structurally different. Moreover, in the pessimism offline RL setting, where distribution shift is the central challenge, the $\chi^2$ regularization together with maxmin formulation has also been explored (Zhan et al., 2022; Huang et al., 2024) from a perspective different from ours. However, frameworks such as $\chi$PO (Huang et al., 2024) are not applicable in our setting because they require the regularizer to be $f$-divergence. The square-root term in Equation 9 does not satisfy this requirement.

## 3.2 STRUCTURED REWARD SPACES VIA FEATURE LINEARIZATION

A natural concern with worst-case optimization is *over-conservatism*: if the reward uncertainty set $\mathcal{R}_{\text{corr}}$ is too broad, the resulting policy may become overly cautious or deviate from realistic task objectives. Additionally, the learned worst-case rewards may themselves be implausible or uninterpretable. To address these issues, we introduce *structure* into the reward space by assuming that all rewards are *linear combinations of known features*, an assumption that has been widely adopted in prior work (Dayan, 1993; Barreto et al., 2017; 2018). Specifically, we assume: $R(s,a) = \boldsymbol{\theta}^\top \boldsymbol{\phi}(s,a)$, where $\boldsymbol{\phi}(s,a) = [\phi_1(s,a), \phi_2(s,a), \ldots, \phi_M(s,a)]^\top \in \mathbb{R}^M$ denotes a vector of $M$ known or engineered feature functions, and $\boldsymbol{\theta} = [\theta_1, \theta_2, \ldots, \theta_M]^\top \in \mathbb{R}^M$ represents the uncertain feature weights. The linearization yields two key benefits: 1) **Realism and Interpretability:** In many real-world tasks, reward functions are naturally approximated as linear combinations over interpretable features. For example, in a traffic control environment, features might include total commute time, vehicle speed, acceleration, and inter-vehicle headway distances. 2) **Better-Constrained Robustness:** By restricting uncertainty to structured, feature-based rewards, the worst-case optimization problem becomes more grounded and avoids pathological, unrealistic reward functions.

In this section, we assume that the agent is aware of the set of features but not their true weights. We show that our robust optimization framework can be naturally extended to incorporate the structure in rewards to improve robustness. In our experiments, we further demonstrate that linear rewards help interpret a policy's performance even when it is trained without such prior knowledge. Under our assumption, the uncertainty set reduces to the set of feature weights $\boldsymbol{\theta} \in \mathbb{R}^M$ satisfying:

$$\mathcal{R}_{\text{corr}}^{\text{lin}} = \left\{ \boldsymbol{\theta} \in \mathbb{R}^M \,\middle|\, \mathbb{E}_{\mu_{\pi_{\text{ref}}}}[\boldsymbol{\theta}^\top \boldsymbol{\phi} \cdot R_{\text{proxy}}] = r, \ \mathbb{E}_{\mu_{\pi_{\text{ref}}}}[\boldsymbol{\theta}^\top \boldsymbol{\phi}] = 0, \ \mathbb{E}_{\mu_{\pi_{\text{ref}}}}[(\boldsymbol{\theta}^\top \boldsymbol{\phi})^2] = 1 \right\}. \quad (10)$$

To simplify the analysis, we assume without loss of generality that the worst-case reward $R(s,a) = \boldsymbol{\theta}^\top \boldsymbol{\phi}(s,a)$ is normalized to have zero mean and unit variance under the reference policy $\pi_{\text{ref}}$. This corresponds to setting $M = 0$ and $V = 1$, which, as shown in our earlier derivation, does not affect the resulting optimal policy. As before, $R_{\text{proxy}}$ denotes the normalized proxy reward under $\pi_{\text{ref}}$, satisfying $\mathbb{E}_{\mu_{\pi_{\text{ref}}}}[R_{\text{proxy}}] = 0$ and $\text{Var}_{\mu_{\pi_{\text{ref}}}}[R_{\text{proxy}}] = 1$.

We now derive the corresponding max-min optimization under the structured reward assumption:

$$\max_{\pi} \min_{\boldsymbol{\theta} \in \mathcal{R}_{\text{corr}}^{\text{lin}}, \boldsymbol{\theta} \geq 0} \mathbb{E}_{(s,a) \sim \mu_\pi} \left[ \boldsymbol{\theta}^\top \boldsymbol{\phi}(s,a) \right]. \quad (11)$$

Similar to previous steps, we introduce the Radon-Nikodym derivative $L(s,a) = \frac{\mu_\pi(s,a)}{\mu_{\pi_{\text{ref}}}(s,a)}$, use a change-of-measure, and define the Lagrangian functional for the inner minimization in Equation 11 as: $l_1(\lambda_1, \lambda_2, \lambda_3, \boldsymbol{\theta}) = \boldsymbol{\theta}^\top \left( \sum_{(s,a)} u_{\lambda_1, \lambda_2}(s,a) \boldsymbol{\phi}(s,a) \right) - \lambda_3 \boldsymbol{\theta}^\top Q \boldsymbol{\theta} + \lambda_1 r + \lambda_3$, where $u_{\lambda_1, \lambda_2} = \mu_\pi - \lambda_1 \mu_{\pi_{\text{ref}}} R_{\text{proxy}} - \lambda_2 \mu_{\pi_{\text{ref}}}$, $Q = \sum_{(s,a)} \mu_{\pi_{\text{ref}}}(s,a) \boldsymbol{\phi}(s,a) \boldsymbol{\phi}(s,a)^\top$. A detailed derivation can be found in Appendix E.4. Then solving the inner minimization over $\boldsymbol{\theta}$ in Equation 11 is equivalent to:

$$\max_{\lambda_1, \lambda_2, \lambda_3} \min_{\boldsymbol{\theta} \geq 0} \quad l_1(\lambda_1, \lambda_2, \lambda_3, \boldsymbol{\theta}) = \boldsymbol{\theta}^\top \left( \sum u_{\lambda_1, \lambda_2} \boldsymbol{\phi} \right) - \lambda_3 \boldsymbol{\theta}^\top Q \boldsymbol{\theta} + \lambda_1 r + \lambda_3. \quad (12)$$

Notice that $l_1(\lambda_1, \lambda_2, \lambda_3, \boldsymbol{\theta})$ is a convex quadratic function of $\boldsymbol{\theta}$ (assuming $\lambda_3 \leq 0$) subject to linear inequality constraints $\boldsymbol{\theta} \geq 0$. Thus, it is a standard convex quadratic program (QP) with non-negativity constraints (Boyd & Vandenberghe, 2004). However, it is not possible to derive a universal closed-form solution for the optimal $\boldsymbol{\theta}^*$ under arbitrary $Q$. To further simplify the problem and obtain a closed-form solution, we transform the feature vector $\boldsymbol{\phi}$ into a whitened version $\tilde{\boldsymbol{\phi}}$ such that the matrix $Q$ becomes the identity matrix $I$ and we formally show this in Appendix E.5. Specifically, we perform a whitening transformation using the Cholesky decomposition (Boyd & Vandenberghe, 2004). Let $W = Q^{-\frac{1}{2}}, \tilde{\boldsymbol{\phi}}(s,a) = W \boldsymbol{\phi}(s,a)$, where $Q^{-\frac{1}{2}}$ denotes a matrix square root of $Q^{-1}$ (which exists since $Q$ is positive semi-definite and non-singular, which is detailed in Appendix E.5). Then the original problem in Equation 12 can be further simplified into:

$$\max_{\lambda_1, \lambda_2, \lambda_3} \min_{\tilde{\boldsymbol{\theta}} \geq 0} \quad l_1(\lambda_1, \lambda_2, \lambda_3, \tilde{\boldsymbol{\theta}}) = \tilde{\boldsymbol{\theta}}^\top \left( \sum_{(s,a)} u_{\lambda_1, \lambda_2}(s,a) \tilde{\boldsymbol{\phi}}(s,a) \right) - \lambda_3 \tilde{\boldsymbol{\theta}}^\top \tilde{\boldsymbol{\theta}} + \lambda_1 r + \lambda_3. \quad (13)$$

where we now optimize over the parameter $\tilde{\boldsymbol{\theta}}$ using the transformed features $\tilde{\boldsymbol{\phi}}$. For notational simplicity, we will drop the tilde and henceforth use $\boldsymbol{\phi}$ to represent the whitened feature $\tilde{\boldsymbol{\phi}}$, and $\boldsymbol{\theta}$ to represent $\tilde{\boldsymbol{\theta}}$. Then we can get a closed-form solution (we detail the steps in Appendix E.6) for optimal $\boldsymbol{\theta}^*$ as: $\boldsymbol{\theta}^* = \max\left(0, -\frac{\sum_{(s,a)} u_{\lambda_1,\lambda_2}(s,a)\phi(s,a)}{2\lambda_3}\right)$, where the $\max(\cdot, 0)$ is applied elementwise. Details for solving the outer maximization in Equation 13 can be found in Appendix E.7. After obtaining the optimal dual variables $(\lambda_1^*, \lambda_2^*, \lambda_3^*)$, we can substitute them back into the optimal $\theta^*$ to construct the worst-case reward, which is the optimal solution of the inner problem of Equation 11 given $\pi$. Then we can solve the outer maximization over the policy $\pi$ using standard RL algorithms.

**ORPO with Linear Rewards.** While ORPO provides a general guarantee based on occupancy measure regularization, it does not exploit any structural assumptions about the reward function. In particular, even when the true reward is linear in a set of features, ORPO does not explicitly incorporate this structure into its policy optimization or theoretical analysis. While the lower bound (Theorem 5.1 in (Laidlaw et al., 2025)) continues to hold, it is unclear how to leverage this structure to obtain a tighter lower bound or to guide policy updates more effectively. This suggests a missed opportunity: by explicitly modeling the reward as a linear function, it becomes possible to derive stronger guarantees, interpret worst-case reward directions, and efficiently optimize against them. Our Linear Maxmin method fills this gap by parameterizing reward uncertainty directly in the space of reward weights, enabling both robustness and greater transparency.

### 3.3 OCCUPANCY ESTIMATION AND CONVERGENCE

A core step in both our algorithms and ORPO is to estimate the Radon-Nikodym derivative $L(s,a)$. To this end, following prior works (Laidlaw et al., 2025; Kang et al., 2018; Ho & Ermon, 2016), we fit a discriminator network $d_\phi(s,a)$ with $L_\phi(s,a) = \exp d_\phi(s,a)$. We learn $\phi$ by minimizing:

$$\phi = \arg\min_\phi \ \mathbb{E}_{\mu_{\pi_{\text{ref}}}}[\log(1 + e^{d_\phi(s,a)})] + \mathbb{E}_{\mu_\pi}[\log(1 + e^{-d_\phi(s,a)})]. \tag{14}$$

It is known that the optimal discriminator satisfies $d^*(s,a) = \log\frac{\mu_\pi(s,a)}{\mu_{\pi_{\text{ref}}}(s,a)}$ and we estimate $L_\phi(s,a)$ as $\tilde{L}_\phi(s,a) = \exp \tilde{d}_\phi(s,a)$ with $\tilde{d}_\phi(s,a) \approx d^*(s,a)$. As discussed in Section 3.1, if the policy $\pi$ visits state-action pairs that the reference policy $\pi_{\text{ref}}$ rarely or never visits, the adversarial reward can be arbitrarily poor. In theory, the estimated $\tilde{L}(s,a)$ is expected to grow arbitrarily large in this case, which should discourage the learned policy from exploiting such regions. However, we observe empirically (Section 4.2) that the ORPO policy still visits some of these low-coverage regions under $\pi_{\text{ref}}$. This is because in the original ORPO implementation, the discriminator is not fully optimized during policy learning. Specifically, the discriminator receives only a small number of gradient updates per RL iteration, resulting in underfitting and inaccurate estimates of the Radon-Nikodym derivative $\tilde{L}(s,a)$. To address this, we substantially increase the number of gradient updates per iteration and carefully tune the learning rate. Our goal is to strike a practical balance between training time and discriminator quality, which we discuss in Appendix F.1. We further show that the following theorem, which establishes that the discriminator estimation achieves a sample complexity of $\mathcal{O}(n^{-1/4})$, where $n$ denotes the sample size.

**Theorem 2** (Occupancy ratio $L_\phi$ error bound). *Under assumptions, let $\tilde{L} := e^{\tilde{d}}$ be the empirical estimation and $L^\star = e^{d^\star}$ be the true ratio. Then, with probability at least $1 - \delta$,*

$$\mathbb{E}_{\mu_{\pi_{\text{ref}}}}\big[\,|\tilde{L} - L^\star|\,\big] \ \leq \ C\left(\gamma' + \left(\frac{\log(M/\delta)}{n}\right)^{1/2}\right)^{1/2}.$$

*where $C$, $\gamma'$ and $M$ are some constants.*

The full argument is presented in Appendix G.1, where we adopt the optimistic cover notion (Definition 3) from (Huang et al., 2023) as a technical tool and establish the new concentration analysis as well as the resulting complexity bounds specific for estimating the loss in Equation 14.

To compute the final objective for our Max-Min policy in Equation 9, we estimate the $\chi^2$ divergence, the normalized proxy reward $R_{\text{proxy}}$, and the first and second moments $\mathbb{E}_{\mu_\pi}[R_{\text{proxy}}]$ and $\mathbb{E}_{\mu_\pi}^2[R_{\text{proxy}}]$. These components together define the robust optimization objective used to update the policy. A

simplified Max-Min policy optimization procedure is outlined in Algorithm 1. We provide detailed descriptions of each estimation step, as well as the complete algorithmic implementation for both Max-Min and Linear Max-Min in Appendices F.2 and F.3. We further obtain a convergence bound of $\mathcal{O}(1/T + 1/N + n^{-1/4})$ for our Max-Min algorithm, by viewing (9) as maximizing a general utility considered in (Zhang et al., 2022; Barakat et al., 2024). Here $T$ is the number of iterations and $N$ is the batch size for policy update. Detailed proofs and the convergence analysis for Linear Max-Min are in Appendix G.

---

**Algorithm 1** Max-Min Policy Optimization (Simplified)

---

1: Initialize policy parameters $\theta$
2: Initialize reference policy $\pi_{\text{ref}}$ and collect trajectories
3: Estimate mean and variance of the proxy reward under $\pi_{\text{ref}}$
4: **for** each iteration **do**
5:     Collect trajectories from current policy $\pi_\theta$
6:     Normalize the proxy rewards for state-action pairs in the collected trajectories
7:     Estimate the expected proxy reward and its second moment under the current policy
8:     Estimate the discriminator using Equation 14 and $\chi^2$ divergence between $\mu_\pi$ and $\mu_{\pi_{\text{ref}}}$
9:     Update the policy using PPO to maximize the Max-Min objective in Equation 9
10: **end for**

---

## 4 EXPERIMENT

### 4.1 EXPERIMENT SETUP

We evaluate our method across five realistic benchmark environments: *Traffic*, *Pandemic*, *Glucose Monitoring*, *Tomato Watering GridWorld* and *RLHF*. These environments were originally proposed in (Pan et al., 2022; Leike et al., 2017) and represent diverse forms of proxy reward hacking, including misweighting, ontological mismatch, and scope misalignment (Pan et al., 2022). A detailed description of the environments and their respective reward structures is provided in Appendix F.4. In each of the five environments, we train policies using both our `Max-Min` and `Linear Max-Min` algorithms for 5 random seeds. For baselines, we compare against the `ORPO` policy. To isolate the impact of discriminator training, we also include an ablation: `ORPO*`, where we train the `ORPO` policy using the same full discriminator training schedule as in our algorithms. This variant shares the same architecture and optimization settings as the original `ORPO`, differing only in the extent of discriminator training. Including this baseline allows us to evaluate the specific contribution of discriminator optimization to policy robustness. For the RLHF environment, we additionally include the `Ensemble` baseline (Eisenstein et al., 2023), a reward-centric approach designed to mitigate reward hacking in RLHF. We include more detailed experimental settings in Appendix F.5 and a discussion of training time and complexity of all algorithms in Appendix F.6.

Since the correlation $r$ may only be approximately estimated and there is currently no principled method for selecting its optimal value. We adopted a similar approach used by `ORPO` (Laidlaw et al., 2025). For each environment, we first performed a grid search over several different values of $r$, and for each fixed $r$, we trained the policy using our algorithm. We then selected the $r$ value that leads to the policy with the best expected worst-case return (detailed in Appendix H.2), which is 0.3 for Traffic, 0.7 for Pandemic, 0.9 for Glucose, 0.4 for Tomato, and 0.4 for RLHF. Results on all searched $r$ can be found in Appendix H.5. Notice that `ORPO` selects the optimal $r$ that yields the best expected return under the true reward, which is infeasible in practice when the true reward is unknown during training. On the other hand, when the exact correlation $r$ is unknown, our approach also raises a concern about how to interpret which worst-case reward is actually meaningful. We include a detailed discussion about how to choose $r$ in practice in Appendix I.

As for evaluation metrics, we report both the expected proxy and true rewards, along with the expected worst-case reward as described in Section 3.1. Note that some policies may visit state-action pairs that are not covered by the reference policy $\pi_{\text{ref}}$. In such cases, we exclude those trajectories and report the occupancy measure of the unseen state-action pairs. Additionally, we evaluate each policy using two variants of the expected linear worst-case reward introduced in Section 3.2. The first uses only the features present in the proxy reward, while the second variant, denoted *Linear Worst\**, leverages features from the true reward, some of which remain unseen during training. This

Table 1: Evaluation results on Traffic, Pandemic, Glucose, and RLHF environments. All policies are trained using **only the proxy reward**. In Traffic, the proxy reward is based on *vel*, *accel*, *headway* (1, 1, 0.1), while the true reward uses *commute*, *accel*, *headway* (1, 1, 0.1). In Pandemic, the proxy reward includes *infection*, *lower stage*, *smooth changes* (10, 0.1, 0.01), while the true reward additionally includes *political* with weight 10 after *infection*. In Glucose, the proxy uses *expected patient cost*, and the true reward uses *magni_bg*. In RLHF, the proxy uses a 70M LLM, and the true reward uses a 8B LLM. We report $\theta$ in the same order as feature weights. **Occ** denotes total occupancy over state-action pairs unseen by $\pi_{\text{ref}}$, where discriminator outputs infinity.

| Env | Traffic | | | | | |
|---|---|---|---|---|---|---|
| **Method** | **True** | **Proxy** | **Worst** | **Linear Worst ($\theta$)** | **Linear Worst* ($\theta$)** | **Occ ↓** |
| ORPO | 16.91±0.12 | 3.41±0.13 | -1.96e+04±0.02e+04 | -0.69±0.01 (0.71, 0.21, 0.69) | -0.83±0.02 (0.63, 0.12, 0.97) | 3.82e-04 ±0.13e-04 |
| ORPO* | 10.26±0.09 | 1.35±0.09 | -1.35e+04±0.02e+04 | -0.44±0.02 (0.46, 0.18, 0.86) | -0.45±0.01 (0.58, 0.06, 0.81) | 1.84e-04±0.07e-04 |
| Max-Min | 12.70±0.06 | 3.63±0.09 | -268.31±4.14 | -0.06±0.01 (0.01, 0.02, 0.96) | -0.06±0.01 (0.001, 0.02, 0.99) | 0.00±0.00 |
| Linear Max-Min | 16.46±0.10 | 2.40±0.11 | -1.19e+04±0.01e+04 | 0.20±0.01 (0.64, 0.07, 0.76) | -0.12±0.01 (0.91, 0.01, 0.67) | 0.00±0.00 |
| **Env** | **Pandemic** | | | | | |
| **Method** | **True** | **Proxy** | **Worst** | **Linear Worst ($\theta$)** | **Linear Worst* ($\theta$)** | |
| ORPO | -1.04±0.21 | 1.75±0.19 | -5.31e+06±0.01e+06 | -2.41±0.02 (0.23, 0.95, 0.17) | -2.65±0.02 (0.02, 0.95, 0.92, 0.08) | |
| ORPO* | 1.18±0.19 | 1.18±0.19 | -4.46e+06±0.03e+06 | -1.36±0.01 (0.25, 0.97, 0.13) | -1.36±0.01 (0.25, 0.97, 0.13) | |
| Max-Min | 1.25±0.18 | 1.25±0.18 | -63.29±3.35 | -1.11±0.01 (0.14, 0.99, 0.01) | -1.11±0.01 (0.14, 0, 0.99, 0.01) | |
| Linear Max-Min | 3.65±0.11 | 7.60±0.13 | -6.82e+05±0.01e+05 | 0.65±0.01 (0.001, 0.23, 0.02) | -0.17±0.02 (0.01, 0.97, 0.22, 0.09) | |
| **Env** | **Glucose** | | | **RLHF** | | |
| **Method** | **True(×10³)** | **Proxy** | **Worst** | **True** | **Proxy** | **Worst** |
| ORPO | 6.0±0.1 | 100.48±0.54 | -27.54±0.32 | 8.30±1.07 | 0.63±0.21 | -1.84±0.03 |
| ORPO* | 6.3±0.2 | 116.36±0.56 | -8.79±0.27 | N/A | N/A | N/A |
| Max-Min | 6.3±0.1 | 102.66±0.58 | -1.71±0.25 | 5.38±0.92 | 0.84±0.11 | -0.10±0.01 |
| Ensemble | N/A | N/A | N/A | 2.31±1.23 | 1.26±0.11 | -1.70±0.04 |

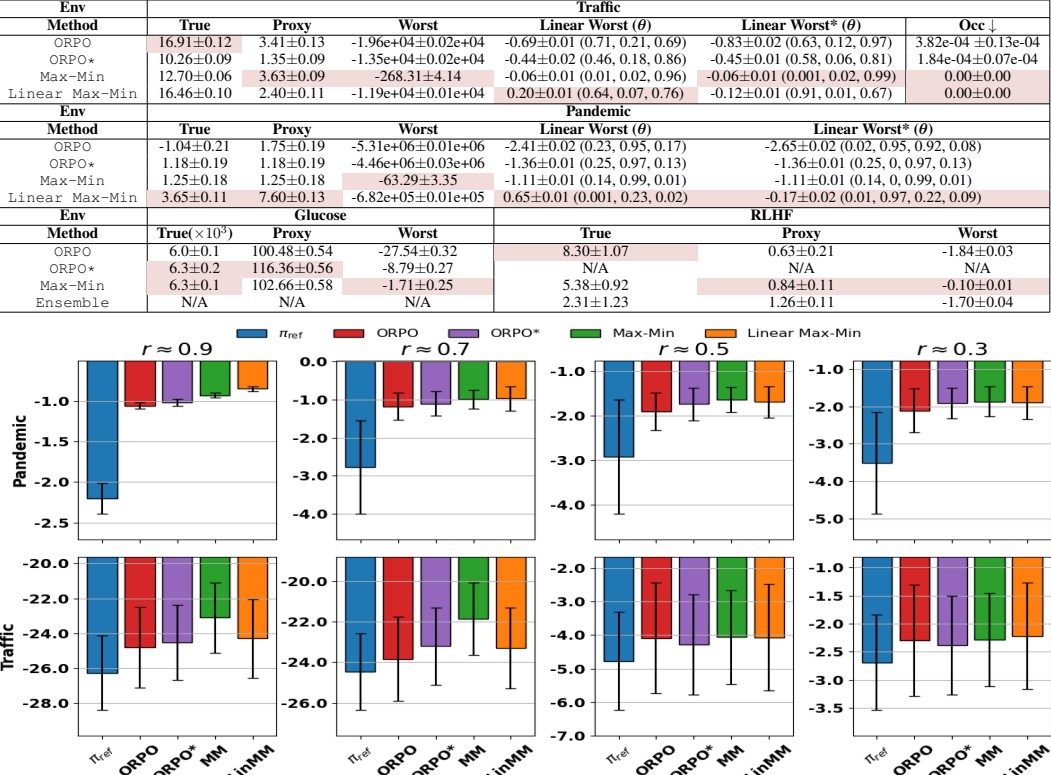

Figure 1: Mean reward and standard deviation under sampled $\theta$ and true reward features at different proxy–true reward correlation levels $r$ for the Traffic and Pandemic environments. Our methods (`Max-Min` and `Linear Max-Min`) yield more stable and higher average performance across all choices of $r$.

setup mimics a more realistic real-world scenario in which the true reward function may depend on features not explicitly modeled at training time. Comparing performance under this setting allows us to assess the robustness of each policy to unseen or misaligned reward structures. All rewards are normalized with respect to the reference policy $\pi_{\text{ref}}$ to ensure a consistent scale across metrics, enabling fair and meaningful comparisons. Note that all worst-case rewards are reported using the fixed correlation level $r$ specified during training.

## 4.2 RESULTS

**Worst-Case Performance.** Table 1 presents the evaluation results on the Traffic, Pandemic, Glucose, and RLHF environments. Additional results for Tomato are provided in Appendix H. Note that we omit the `Linear Max-Min` policy from the Glucose and RLHF environments for the following reasons. In the Glucose and RLHF environment, both the proxy and true rewards used in prior work (Laidlaw et al., 2025; Baker et al., 2025) are based on a single feature, making the linear reward formulation trivial. Although the original Glucose simulator provides multiple candidate features related to patient health, selecting an appropriate feature combination without prior knowledge of clinical intent is nontrivial. Therefore, in both settings, we report only the results for the `Max-Min` policy alongside the baselines. Our `Max-Min` and `Linear Max-Min` policies achieve better expected worst-case performance under both general and linear adversarial rewards, while remaining competitive with baselines in terms of expected true and proxy rewards. Notably, the `Max-Min` policy attains the highest expected worst-case return, followed by `Linear Max-Min`.

Conversely, `Linear Max-Min` yields the highest expected linear worst-case reward, followed by `Max-Min`, demonstrating the robustness of both approaches under worst-case scenarios. For the Linear Worst* evaluation, which uses reward features unseen during training, we observe minimal degradation in `Max-Min` policy's performance, indicating its strong robustness to feature variation. In contrast, the performance of `Linear Max-Min` declines in this case, suggesting its advantage diminishes when prior assumptions about feature structure are inaccurate. We also find that `ORPO*` exhibits better worst-case performance than the original `ORPO`. In particular, training the discriminator more thoroughly significantly reduces the occupancy of state-action pairs that are not visited by the reference policy, indicating that more accurate estimation of the Radon-Nikodym derivative leads to improved policy robustness. Notably, in the Pandemic and Glucose environment, we observe no such unvisited state-action pairs, and the discriminator outputs remain small across all policies. This could be due to either the discriminator network not being fully optimized or its inability to capture rare events that fall outside the support of $\pi_{\text{ref}}$. Developing more reliable techniques for handling such rare or unseen state-action pairs remains an open direction for future work. We also report the adversarial weight vectors $\boldsymbol{\theta}$ for each policy. These weights reveal which features are most vulnerable to proxy exploitation under the learned policy and can be used to diagnose and revise the proxy reward function, thereby improving robustness. This highlights the interpretability benefits of our framework. Moreover, several patterns emerge from the results, which is detailed in Appendix H.2. We also notice that the `Ensemble` baseline in the RLHF setting achieves only limited improvement in expected true return over the reference policy and attains a lower expected worst-case return than our method. These results indicate that using reward ensembles alone is insufficient to effectively mitigate reward hacking compared to our approach. However, such reward-centric methods, including InfoRM (Miao et al., 2024) and RRM (Liu et al., 2024a), can be easily integrated into our framework. In particular, these approaches can be used to construct a stronger proxy reward, which can then be plugged into our method to further improve performance.

**Robustness Across Correlation Levels.** To further assess the robustness of each policy across a broader range of proxy–true correlation scenarios, we also compute the Linear Worst* for each policy under varying $r$ values. Specifically, for each $r$, we sample 1000 vectors $\boldsymbol{\theta}$ such that $\boldsymbol{\theta} \in \mathcal{R}_{\text{corr}}^{\text{lin}}$, and report the average return and variance achieved by each policy over these sampled rewards. Importantly, the variation in $r$ is applied **only during evaluation**; all policies are fixed and trained using the specific $r$ values reported in Appendix F.5. Unlike evaluations that only consider several reward functions, this approach evaluates policy performance across the entire reward set $\mathcal{R}_{\text{corr}}^{\text{lin}}$, providing a more comprehensive measure of robustness and better reflecting real-world scenarios where the true reward and correlation $r$ are unknown. Figure 1 shows the average reward and variance achieved by each method under different levels of proxy–true reward correlation $r$. As expected, the reference policy $\pi_{\text{ref}}$ (blue) performs the worst across all correlation levels in both environments. In Traffic, its variance is relatively small, suggesting consistently poor but stable behavior. In contrast, variance is highest in the Pandemic environment, indicating increased policy fragility. Notably, `ORPO*` (purple) consistently achieves lower variance than `ORPO` (red) across both environments and outperforms it in terms of average reward at $r \approx 0.9$ and $r \approx 0.7$ in Traffic, and across nearly all $r$ values in Pandemic. This underscores the importance of accurate discriminator training for improving both stability and robustness. `Max-Min` (green) demonstrates the highest average reward and lowest variance across a wide range of $r$ values in both environments, showing strong resilience to reward misspecification. While `Linear Max-Min` (orange) achieves the best performance at specific correlation levels, particularly $r \approx 0.3$ in Traffic and $r \approx 0.7\text{-}0.9$ in Pandemic. As $r$ decreases and the proxy becomes less informative, differences in average reward among methods shrink, while variance increases. These results highlight the significance of variance control in low-correlation regimes and demonstrate that `Max-Min` and `Linear Max-Min` offer robust and stable performance under high uncertainty.

## 5 CONCLUSION

In this work, we formalize reward hacking as a robust optimization problem and introduce both a Max-Min formulation with a closed-form adversarial reward and a Linear Max-Min variant that further improves interpretability and tractability. We develop efficient algorithms and empirically validate that both Max-Min and Linear Max-Min policies achieve stronger worst-case performance and improved stability compared to prior baselines such as ORPO across diverse environments. We further discuss limitations and broader impacts of our method in Appendices B and C.

ACKNOWLEDGMENTS

This work was supported in part by NSF grant CNS-2146548 and a grant from the Louisiana Board of Regents. We thank the anonymous reviewers for their insightful and constructive feedback.

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

APPENDIX

## A    LLM USAGE

In this work, we used ChatGPT to improve grammar, wording, and paragraph flow throughout the paper after completing an initial draft. We also used ChatGPT's research capabilities to help surface potentially relevant prior work for the related work and introduction sections. All references were independently verified by the authors. No algorithms, proofs, or experimental results were generated by ChatGPT, and no proprietary or sensitive data were shared with the tool. All technical contributions and analyses are solely the authors' work.

## B    LIMITATIONS AND FUTURE WORK

Despite the effectiveness of our framework, several limitations remain. First, although our Max-Min formulation can be extended naturally beyond linear reward structures, incorporating more expressive representations such as neural networks makes the inner optimization problem significantly harder. In such cases, the inner minimization may no longer admit a closed-form solution, necessitating iterative training between the policy and adversary. This increases computational complexity and undermines the efficiency advantages of our current formulation. Developing scalable solutions for general reward representations remains an open direction.

Second, in our Linear Max-Min algorithm, computing $Q$ results in $O(d^2)$ space and $O(d^3)$ time complexity. While low-rank approximations could potentially reduce computational cost, such methods often discard small eigenvalues. However, in our setting, these small eigenvalues become critical due to the inversion in the whitening step, and removing them may severely distort the worst-case reward direction. Therefore, naive low-rank approximations may not be applicable in our setting, and we emphasize the need for principled, scalable extensions when applying our method to settings with very high-dimensional feature spaces.

Third, for complex environments, constructing effective features for the reward function is often challenging without prior domain knowledge. For example, in the Glucose environment, a large number of health-related indices are provided. However, without medical expertise or knowledge of glucose monitoring, it is difficult to determine which combination of indices best captures patient health or blood glucose trends. Using arbitrarily selected features in such cases can lead to proxy rewards that exhibit little or no correlation with the true reward. While our max-min formulation can still offer robustness under such misspecification, the resulting policy is nevertheless expected to perform poorly due to the fundamental misalignment between the proxy and true objectives. Therefore, designing meaningful reward features remains a fundamental and unresolved challenge, and we will include this as a limitation of our method in the main text. Moreover, in some environments, such as Tomato, the reward function is not explicitly feature-based. Although our general max-min algorithm still applies in this setting, incorporating non feature-based reward structure into the uncertainty set remains an open problem.

Fourth, like ORPO, our framework assumes access to a fixed proxy reward, a reference policy, and a pre-specified correlation parameter $r$, all provided offline. This setup limits the ability of the algorithm to adapt or refine its reward model based on new information. However, we observe that the adversarial rewards generated by our method, particularly the structured linear ones, can serve as diagnostic tools to identify vulnerable reward features. These insights could be leveraged to guide human-in-the-loop refinement or adaptive querying of stronger feedback models (e.g., large language models). Extending our framework to close the loop between diagnostic robustness and reward learning is an exciting direction for future work.

Fifth, while our experimental results demonstrate that the proposed method improves robustness across a range of proxy-true reward correlation levels, an alternative and perhaps more direct strategy would be to train the policy against multiple proxy rewards sampled at varying levels of correlation $r$. In principle, optimizing the average performance across a diverse set of proxies could yield a policy that is robust to a wider distribution of potential reward misspecifications. However, this approach presents several practical challenges. First, there is a trade-off between computational cost and coverage: sampling too few proxies may fail to represent the full space of plausible reward deviations, while sampling many proxies significantly increases training time. Second, efficiently generating reward functions that satisfy a fixed correlation constraint with the proxy reward becomes non-trivial in high-dimensional or continuous state-action spaces. Designing scalable and effective

reward sampling mechanisms (such as leveraging diffusion models) under correlation constraints remains an open problem and a promising direction for future research.

## C    BROADER IMPACTS

Designing reward functions that faithfully reflect designer intent remains a fundamental challenge in deploying reinforcement learning (RL) systems in the real world. When reward misspecification occurs, agents can behave in undesirable or even dangerous ways. Our work addresses this issue by proposing a robust policy optimization framework that explicitly accounts for uncertainty in the reward function, improving worst-case performance across a range of plausible reward proxies. This approach has the potential to increase the safety and reliability of RL systems in safety-critical applications such as healthcare, autonomous driving, and digital infrastructure, where poorly specified incentives can lead to unintended consequences. In addition to robustness, our linear variant contributes to policy interpretability by yielding explicit weightings over features that can be inspected and audited. This can help practitioners identify vulnerable components in their reward specification and make better-informed decisions when refining proxies. However, while our method is primarily intended to prevent reward exploitation, one could conceivably use adversarial reward modeling to stress-test or attack policies. We believe the benefits of improved safety and robustness outweigh this risk, especially when combined with interpretability. Overall, this work contributes to the safe and trustworthy deployment of RL by equipping practitioners with more robust and explainable optimization tools.

## D    RELATED WORK

### D.1    REWARD HACKING

Early work in AI safety underscored the pitfalls of optimizing an imperfect proxy reward. Amodei et al. (Amodei et al., 2016) famously illustrate how an agent can "game" its reward function: for example, a cleaning robot rewarded for not seeing any messes might simply close its cameras or create messes to clean up, maximizing the proxy reward while betraying the designer's intent. Other examples of such reward hacking include an agent in a racing game that spins in circles to collect points instead of completing the race (Skalse et al., 2022), social media recommendation systems that promote emotionally extreme content to increase engagement (Harari, 2024), and Large Language Models (LLMs) that generate trivial or hard-coded solutions to pass unit tests rather than producing general, correct code (Baker et al., 2025). Krakovna et al. (Krakovna, 2018) have catalogued many such failure cases across diverse domains. Several studies have analyzed the causes of reward hacking (Amodei et al., 2016; Krakovna, 2019; Skalse et al., 2022), often interpreting it as a manifestation of Goodhart's Law (Goodhart & Goodhart, 1984): when a proxy metric becomes a target for optimization, it ceases to be a good measure. In reinforcement learning, this risk is particularly acute because agents can exploit even small imperfections in the reward specification. Pan et al. (Pan et al., 2022) further propose a taxonomy of proxy reward misspecification into three types: *misweighting*, *ontological*, and *scope* errors.

To mitigate such risks, several reward-centric methods have been proposed (Hadfield-Menell et al., 2017; Ramé et al., 2024). Inverse Reward Design (Hadfield-Menell et al., 2017) aims to infer the intended true objective from a given proxy and its training context, helping agents generalize without exploiting flawed signals. Recent work by Rame et al. (Ramé et al., 2024) averages the parameters of multiple reward models to smooth out idiosyncratic errors, reduce the impact of individual proxy biases, and demonstrate reduced reward hacking on held-out tests. Another line of defense focuses on regularizing policy behavior to reduce sensitivity to reward flaws. Common approaches include penalizing divergence from a reference policy using KL-regularization (Liu et al., 2020). Recent research by Laidlaw et al. (Laidlaw et al., 2025) proposes Occupancy-Regularized Policy Optimization (ORPO), which applies a $\chi^2$ penalty on the state-action distribution to constrain deviation from a baseline policy and reduce exploitative behaviors. Another complementary paradigm is assistance games (Fern et al., 2014; Shah et al., 2020), in which human users remain actively involved and the agent's actions complement the user's to achieve optimal joint performance. Assistance games can mitigate reward hacking by removing incentives for deception since the agent's performance

depends on the human's latent (true) reward. Recent work has developed scalable assistance-game approaches in practice (Laidlaw et al., 2024).

Overall, existing approaches either attempt to correct the reward specification, regularize against a fixed proxy, or explicitly involve human interaction. In contrast, our method trains policies against an entire set of plausible proxy rewards, those that remain sufficiently correlated with the true reward, offering robustness to a broader range of misspecifications. Moreover, we show that this robust training objective can be reformulated as an equivalent regularized optimization problem, providing both theoretical and practical benefits.

## D.2 Reward Modeling in Reinforcement Learning

In standard RL benchmarks, the reward is usually assumed to be given, but real-world applications rarely offer a well-defined reward signal upfront. Therefore, designing an effective reward function, often referred to as reward modeling, is a critical yet challenging aspect of RL (Skalse et al., 2022; Liu et al., 2024b; Booth et al., 2023; Knox & MacGlashan, 2024). Moreover, evaluating whether a designed reward truly captures the designer's intent is non-trivial. Recent work (Muslimani et al., 2025) has proposed to measure reward alignment via a "trajectory alignment coefficient," which quantifies how closely the rankings of trajectories induced by a given reward match a human stakeholder's preferences. Such efforts underscore the importance of conceptual frameworks that go beyond treating the reward as a black box, instead focusing on principled reward design and evaluation.

To incorporate domain knowledge and improve interpretability, researchers have explored structured or rule-based reward modeling frameworks (Icarte et al., 2022; Brafman et al., 2018; Camacho et al., 2017). One prominent example is the use of reward machines (Icarte et al., 2022) that explicitly represent the reward function's logic. A reward machine exposes the internal structure of the reward (e.g. different sub-goal phases or conditions) to the agent, enabling techniques like automated reward shaping and task decomposition for more sample-efficient learning. Even before the advent of reward machines, prior works had leveraged logical task specifications to design rewards. For instance, translating Linear Temporal Logic (LTL) formulas into automata and rewarding the agent upon reaching designated accepting states (Camacho et al., 2017; Brafman et al., 2018). By defining rewards through such rules or logical templates, the intended behavior is encoded transparently, making the reward function more interpretable. Along similar lines, many approaches assume a structured parametric form for the reward function itself to aid transparency (Yu et al., 2025; Mu et al., 2024). In particular, it is common to model the reward as a linear combination of feature functions, a simplification used in inverse RL and preference-based reward learning to make reward inference tractable and explainable (Yu et al., 2025). Recent work in RL from human feedback also implements rule-based reward signals as linear models over interpretable features (Mu et al., 2024). Our approach follows this tradition: by assuming the reward is linear in a set of human-interpretable features, we improve the interpretability of the learned policies and reveal which feature components are robust or vulnerable.

## D.3 Robust Reinforcement Learning

Our work is also related to robust reinforcement learning, where the agent assumes the reward function (and/or transition dynamics) lies within a given uncertainty set, and it seeks to maximize performance against the worst-case realization from that set. This can be formulated as a zero-sum dynamic game between the agent and an adversary who selects the most adverse reward or dynamics; solving the robust MDP thus involves a challenging max-min optimization (Iyengar, 2005; Nilim & El Ghaoui, 2005). To alleviate the computational complexity, early works in this vein rely on a rectangularity assumption that is crucial for traceability. Thus, classical robust RL formulation typically considers rectangular uncertainty sets on rewards or transition probabilities, which lead to conservative solutions but permit efficient algorithms such as robust value iteration (Bagnell et al., 2001; Grand-Clément & Kroer, 2021) or modified policy iteration (MPI) (Kaufman & Schaefer, 2013).

Recent theoretical work has revealed an intimate connection between adversarial robustness and policy regularization in the context of rectangular uncertainty sets. Several researchers have shown that solving a robust MDP is equivalent to solving a certain regularized RL problem (Derman et al., 2021;

Eysenbach & Levine; Mcmahan et al., 2024). In particular, the worst-case effect of the adversary can often be captured via an additional penalty term in the agent's objective. Derman et al. (Derman et al., 2021) prove that any entropy- or $L^2$-regularized MDP can be interpreted as a robust MDP with uncertain rewards – in fact, a regularized MDP is a special case of a reward-robust MDP. Their analysis establishes a duality between a max-min reward-robust objective and a single-agent maximization of expected reward plus a regularization term. Eysenbach and Levine (Eysenbach & Levine) show that the optimal policy from a maximum entropy RL formulation is provably robust to some adversarial reward perturbations. More recently, these insights have been extended and formalized for general MDPs and even multi-agent settings. McMahan et al. (Mcmahan et al., 2024) study robust Markov games with $(s, a)$-rectangular uncertainty, and they prove that computing a robust equilibrium is polynomial-time equivalent to computing an equilibrium in a corresponding regularized game. In their framework, the added regularization term is exactly the support function of the uncertainty set, effectively the dual representation of the adversary's worst-case reward selection. This means that for common uncertainty sets (e.g., those inducing entropy or $\ell_p$-norm regularizers), one can replace the inner minimization over rewards with an explicit regularization term in the objective.

The setting in our work departs from the above literature by considering **non-rectangular** reward uncertainty. In particular, we assume a correlation-constrained uncertainty set for the reward function, meaning that the adversary's permissible deviations in reward are coupled across states. This structure can mitigate the conservativeness of the worst-case solution (the adversary cannot simultaneously push all state rewards to their extreme worst values) (Goyal & Grand-Clement, 2018), but it also means that the neat robustness-regularization duality from the rectangular-case no longer applies and the robust optimization must be solved (or approximated) directly. In summary, our work tackles a form of reward uncertainty which lies beyond the scope of existing robustness-as-regularization analysis.

### D.4 Successor Representations in Reinforcement Learning

The linear reward assumption and the use of discounted feature expectations are also closely related to the literature on successor representations/successor features. Successor representations and successor features represent values as inner products between reward weights and discounted occupancies or features of future states and actions. It was introduced as a generalization of the value function (Dayan, 1993). This idea was later generalized by (Gehring, 2015; Barreto et al., 2017; 2018) to handle high-dimensional, continious state spaces and to use the method for transfer learning. Specifically, (Barreto et al., 2017) formalize this idea into the successor features (SF) framework for transfer learning, assuming that tasks share dynamics but differ only in their reward functions parameterized as linear combinations of features. This yields a value function representation that effectively decouples the environment's transition dynamics from the reward parameters. (Barreto et al., 2018) further extend successor features to deep reinforcement learning and introduce generalized policy improvement over multiple tasks, demonstrating effective transfer by reusing learned successor features across a family of related tasks.

## E Proofs and Additional Theoretical Results

### E.1 Solve the Max-Min Objective

In this section, we show the complete proof for solving the following max-min problem:

$$\max_{\pi} \min_{R \in \mathcal{R}_{\text{corr}}} J(\pi, R) = \max_{\pi} \min_{R \in \mathcal{R}_{\text{corr}}} \mathbb{E}_{(s,a) \sim \mu_\pi}[R(s,a)]. \tag{15}$$

where $\mathcal{R}_{\text{corr}}$ is the entire space of rewards that satisfy the correlation constraint with respect to a known proxy reward, as defined below:

$$\mathcal{R}_{\text{corr}} = \left\{ R : (s,a) \to \mathbb{R} \,\middle|\, \mathbb{E}_{\mu_{\pi_{\text{ref}}}}\left[\frac{R-M}{V} \cdot R_{\text{proxy}}\right] = r, \, J(\pi_{\text{ref}}, R) = M, \, \sigma_R^2 = V^2 \right\} \tag{16}$$

$$= \left\{ R : (s,a) \to \mathbb{R} \,\middle|\, \mathbb{E}_{\mu_{\pi_{\text{ref}}}}[\frac{R-M}{V} \cdot R_{\text{proxy}}] = r, \, \mathbb{E}_{\mu_{\pi_{\text{ref}}}}[R] = M, \, \mathbb{E}_{\mu_{\pi_{\text{ref}}}}[R^2] = V^2 + M^2 \right\}.$$

$M$ and $V$ denote the fixed mean and standard deviation of the reward function $R$ under the reference policy $\pi_{\text{ref}}$. $R_{\text{proxy}}$ is the normalized proxy reward

$$R_{\text{proxy}}(s,a) := \frac{\tilde{R}_{\text{proxy}}(s,a) - J(\pi_{\text{ref}}, \tilde{R}_{\text{proxy}})}{\sigma_{\tilde{R}_{\text{proxy}}}}$$

where $\tilde{R}_{\text{proxy}}$ is the original (unnormalized) proxy reward. After normalization, we have $J(\pi_{\text{ref}}, R_{\text{proxy}}) = 0$ and $\text{Var}_{\mu_{\pi_{\text{ref}}}}(R_{\text{proxy}}) = 1$.

To solve the challenge that the objective $\mathbb{E}_{\mu_\pi}[R(s,a)]$ depends on the state-action occupancy $\mu_\pi$, whereas the constraints defining $\mathcal{R}_{\text{corr}}$ are expressed in terms of $\mu_{\pi_{\text{ref}}}$. We apply a *change-of-measure* technique (Hu & Hong, 2013; Lam, 2016) to rewrite the expectation under $\mu_{\pi_{\text{ref}}}$. Specifically, let $L(s,a)$ denote the Radon-Nikodym derivative:

$$L(s,a) = \frac{\mu_\pi(s,a)}{\mu_{\pi_{\text{ref}}}(s,a)}$$

By definition, $L(s,a) \geq 0$ and $\mathbb{E}_{\mu_{\pi_{\text{ref}}}}[L(s,a)] = 1$. Applying the change-of-measure formula, we can express the return as:

$$\begin{aligned}
\mathbb{E}_{\mu_\pi}[R(s,a)] &= \int_{\mathcal{S} \times \mathcal{A}} \mu_\pi(s,a) R(s,a) \, d(s,a) \\
&= \int_{\mathcal{S} \times \mathcal{A}} \mu_{\pi_{\text{ref}}}(s,a) \frac{\mu_\pi(s,a)}{\mu_{\pi_{\text{ref}}}(s,a)} R(s,a) \, d(s,a) \\
&= \mathbb{E}_{\mu_{\pi_{\text{ref}}}}[L(s,a) R(s,a)]
\end{aligned}$$

Thus, both the objective and the constraints can be rewritten as expectations with respect to the reference distribution $\mu_{\pi_{\text{ref}}}$. For notational simplicity, we will suppress the variables $(s,a)$ when necessary. Under this reparameterization, the max-min objective in Equation 15 can be reformulated as:

$$\max_\pi \min_{R \in \mathcal{R}_{\text{corr}}} \mathbb{E}_{\mu_{\pi_{\text{ref}}}}[L \cdot R]. \tag{17}$$

**Solve Inner Minimization Problem.** The Lagrangian functional associated with the inner minimization problem of 17 is defined as:

$$l_0(\lambda_1, \lambda_2, \lambda_3, R) = \mathbb{E}_{\mu_{\pi_{\text{ref}}}}\left[L \cdot R - \lambda_1 \frac{R-M}{V} \cdot R_{\text{proxy}} - \lambda_2 R - \lambda_3 R^2\right] + \lambda_1 r + \lambda_2 M + \lambda_3(M^2 + V^2)$$

where $\lambda_1, \lambda_2, \lambda_3$ are the Lagrange multipliers corresponding to the correlation constraint, mean constraint, and variance constraint, respectively. Then the inner minimization problem in Equation 17 is equivalent to the following problem:

$$\max_{\lambda_1, \lambda_2, \lambda_3} \min_{R \in \mathcal{R}_{\text{corr}}} l_0(\lambda_1, \lambda_2, \lambda_3, R) \tag{18}$$

We now solve the inner minimization problem in Equation 18 by finding the optimal $R$ for fixed dual variables $(\lambda_1, \lambda_2, \lambda_3)$. Taking the functional derivative of the Lagrangian $l_0$ with respect to $R(s,a)$ gives:

$$\frac{\partial l_0}{\partial R} = \mu_{\pi_{\text{ref}}}(s,a)\left[(L - \lambda_1 \frac{R_{\text{proxy}}(s,a)}{V} - \lambda_2) - 2\lambda_3 R\right]$$

When $\mu_{\pi_{\text{ref}}}(s,a) > 0$, setting the derivative of the Lagrangian to zero yields the optimal adversarial reward function:

$$\boxed{R^*(s,a) = \frac{L(s,a) - \lambda_1 \frac{R_{\text{proxy}}(s,a)}{V} - \lambda_2}{2\lambda_3}} \tag{19}$$

However, for state-action pairs where $\mu_{\pi_{\text{ref}}}(s, a) = 0$, i.e., those not visited under the reference policy, the correlation and moment constraints become vacuous. In these regions, the adversarial reward $R^*(s, a)$ can be driven arbitrarily poor, reflecting that no constraint prevents the adversary from assigning highly penalizing values to rarely visited or unobserved state-action pairs. Nevertheless, consider the case where $\mu_{\pi_{\text{ref}}}(s, a) > 0$, after substituting the optimal $R^*$ from Equation 19 into the Lagrangian $l_0$ in Equation 18 and simplifying, we obtain the following dual objective:

$$\max_{\lambda_1, \lambda_2, \lambda_3} l_0(\lambda_1, \lambda_2, \lambda_3, R^*) = \mathbb{E}_{\mu_{\pi_{\text{ref}}}}\left[\frac{(L(s,a) - \lambda_1 \frac{R_{\text{proxy}}(s,a)}{V} - \lambda_2)^2}{4\lambda_3}\right] + \lambda_1 r + \lambda_2 M + \lambda_3(M^2 + V^2)$$

(20)

We now compute the gradients of the dual objective with respect to the dual variables:

$$\frac{\partial l_0}{\partial \lambda_1} = -\mathbb{E}_{\mu_{\pi_{\text{ref}}}}\left[\frac{(L - \lambda_1 \frac{R_{\text{proxy}}(s,a)}{V} - \lambda_2)\frac{R_{\text{proxy}}(s,a)}{V}}{2\lambda_3}\right] + r$$

$$\frac{\partial l_0}{\partial \lambda_2} = -\mathbb{E}_{\mu_{\pi_{\text{ref}}}}\left[\frac{L - \lambda_1 \frac{R_{\text{proxy}}(s,a)}{V} - \lambda_2}{2\lambda_3}\right] + M$$

$$\frac{\partial l_0}{\partial \lambda_3} = -\mathbb{E}_{\mu_{\pi_{\text{ref}}}}\left[\frac{(L - \lambda_1 \frac{R_{\text{proxy}}(s,a)}{V} - \lambda_2)^2}{4\lambda_3^2}\right] + M^2 + V^2$$

(21)

Setting these gradients to zero yields the system of equations:

$$\mathbb{E}_{\mu_{\pi_{\text{ref}}}}\left[\frac{(L - \lambda_1 \frac{R_{\text{proxy}}(s,a)}{V} - \lambda_2)\frac{R_{\text{proxy}}(s,a)}{V}}{2\lambda_3}\right] = r,$$

(22)

$$\mathbb{E}_{\mu_{\pi_{\text{ref}}}}\left[\frac{L - \lambda_1 \frac{R_{\text{proxy}}(s,a)}{V} - \lambda_2}{2\lambda_3}\right] = M,$$

(23)

$$\mathbb{E}_{\mu_{\pi_{\text{ref}}}}\left[\frac{(L - \lambda_1 \frac{R_{\text{proxy}}(s,a)}{V} - \lambda_2)^2}{4\lambda_3^2}\right] = M^2 + V^2.$$

(24)

Expanding and simplifying each condition:

**Solving for $\lambda_2$:** Starting with Equation 23,

$$\mathbb{E}_{\mu_{\pi_{\text{ref}}}}[L] - \lambda_1 \mathbb{E}_{\mu_{\pi_{\text{ref}}}}[\frac{R_{\text{proxy}}(s, a)}{V}] - \lambda_2 = 2\lambda_3 M$$

Recall from our normalization that $\mathbb{E}_{\mu_{\pi_{\text{ref}}}}[R_{\text{proxy}}] = 0$. Thus,

$$\lambda_2 = \mathbb{E}_{\mu_{\pi_{\text{ref}}}}[L] - 2\lambda_3 M$$

Since $L(s, a) = \frac{\mu_\pi(s,a)}{\mu_{\pi_{\text{ref}}}(s,a)}$, and using properties of Radon-Nikodym derivatives, we have:

$$\mathbb{E}_{\mu_{\pi_{\text{ref}}}}[L] = 1$$

Thus, we find:

$$\boxed{\lambda_2 = 1 - 2\lambda_3 M}$$

**Solving for $\lambda_1$:** Substituting $\lambda_2 = 1 - 2\lambda_3 M$ into Equation 22,

$$2r\lambda_3 = \mathbb{E}_{\mu_{\pi_{\text{ref}}}}[(L - \lambda_1 \frac{R_{\text{proxy}}}{V} - 1 + 2\lambda_3 M)\frac{R_{\text{proxy}}}{V}]$$

$$= \mathbb{E}_{\mu_{\pi_{\text{ref}}}}[L \cdot \frac{R_{\text{proxy}}}{V}] - \lambda_1 \mathbb{E}_{\mu_{\pi_{\text{ref}}}}[\frac{R_{\text{proxy}}^2}{V^2}] - (1 - 2\lambda_3 M)\mathbb{E}_{\mu_{\pi_{\text{ref}}}}[\frac{R_{\text{proxy}}}{V}]$$

Again, using normalization, $\mathbb{E}_{\mu_{\pi_{\text{ref}}}}[R_{\text{proxy}}] = 0$ and $\mathbb{E}_{\mu_{\pi_{\text{ref}}}}[R_{\text{proxy}}^2] = 1$, so we get:

$$2r\lambda_3 = \mathbb{E}_{\mu_{\pi_{\text{ref}}}}[L \cdot \frac{R_{\text{proxy}}}{V}] - \frac{\lambda_1}{V^2}$$

which rearranges to:

$$\boxed{\lambda_1 = V\mathbb{E}_{\mu_{\pi_{\text{ref}}}}[L \cdot R_{\text{proxy}}] - 2r\lambda_3 V^2}$$

**Solving for $\lambda_3$:** Substituting $\lambda_2 = 1 - 2\lambda_3 M$ into Equation 24,

$$4\lambda_3^2(M^2 + V^2) = \mathbb{E}_{\mu_{\pi_{\text{ref}}}}\left[(L - \lambda_1\frac{R_{\text{proxy}}}{V} - 1 + 2\lambda_3 M)^2\right]$$

$$= \mathbb{E}_{\mu_{\pi_{\text{ref}}}}[L^2] + \lambda_1^2\mathbb{E}_{\mu_{\pi_{\text{ref}}}}[\frac{R_{\text{proxy}}^2}{V^2}] + (1 - 2\lambda_3 M)^2 - 2\lambda_1\mathbb{E}_{\mu_{\pi_{\text{ref}}}}[L \cdot \frac{R_{\text{proxy}}}{V}]$$

$$- 2(1 - 2\lambda_3 M)\mathbb{E}_{\mu_{\pi_{\text{ref}}}}[L] + 2\lambda_1(1 - 2\lambda_3 M)\mathbb{E}_{\mu_{\pi_{\text{ref}}}}[\frac{R_{\text{proxy}}}{V}]$$

Again, using normalization ($\mathbb{E}[R_{\text{proxy}}] = 0$, $\mathbb{E}[R_{\text{proxy}}^2] = 1$, $\mathbb{E}[L] = 1$), this simplifies to:

$$4\lambda_3^2(M^2 + V^2) = \mathbb{E}_{\mu_{\pi_{\text{ref}}}}[L^2] + \frac{\lambda_1^2}{V^2} - 2\lambda_1\mathbb{E}_{\mu_{\pi_{\text{ref}}}}[L \cdot \frac{R_{\text{proxy}}}{V}] + 4\lambda_3^2 M^2 - 1$$

$$4\lambda_3^2 V^2 = \mathbb{E}_{\mu_{\pi_{\text{ref}}}}[L^2] + \frac{\lambda_1^2}{V^2} - 2\lambda_1\mathbb{E}_{\mu_{\pi_{\text{ref}}}}[L \cdot \frac{R_{\text{proxy}}}{V}] - 1$$

Now substitute $\lambda_1 = V\mathbb{E}_{\mu_{\pi_{\text{ref}}}}[L \cdot R_{\text{proxy}}] - 2r\lambda_3 V^2$ into this expression. After rearrangement and simplification, we obtain:

$$4\lambda_3^2(1 - r^2)V^2 = \mathbb{E}_{\mu_{\pi_{\text{ref}}}}[L^2] - \mathbb{E}_{\mu_{\pi_{\text{ref}}}}^2[L \cdot R_{\text{proxy}}] - 1$$

Thus,

$$\lambda_3 = \pm\frac{1}{2}\frac{\sqrt{\mathbb{E}_{\mu_{\pi_{\text{ref}}}}[L^2] - \mathbb{E}_{\mu_{\pi_{\text{ref}}}}^2[L \cdot R_{\text{proxy}}] - 1}}{V\sqrt{1 - r^2}}$$

We argue that $\lambda_3 < 0$ yields the optimal dual variable. To determine which root maximizes the above dual objective in Equation 20, we compute the second derivative from Equation 21:

$$\frac{\partial^2 l_0}{\partial\lambda_3^2} = \mathbb{E}_{\mu_{\pi_{\text{ref}}}}\left[\frac{(L - \lambda_1\frac{R_{\text{proxy}}(s,a)}{V} - \lambda_2)^2}{2\lambda_3^3}\right]$$

Since the numerator is always non-negative and when $\lambda_3 < 0$, we have $\frac{\partial^2 l_0}{\partial\lambda_3^2} < 0$, which implies the dual objective is concave in $\lambda_3$ around this root. Thus, selecting the negative root yields a local maximum of the dual objective.

Recognizing that $\mathbb{E}_{\mu_{\pi_{\text{ref}}}}[L^2] - 1$ corresponds to the $\chi^2$ divergence between the occupancy measures:

$$\chi^2(\mu_\pi \| \mu_{\pi_{\text{ref}}}) = \mathbb{E}_{\mu_{\pi_{\text{ref}}}}[L^2] - 1$$

and noting that:

$$\mathbb{E}_{\mu_{\pi_{\text{ref}}}}[L \cdot R_{\text{proxy}}] = \mathbb{E}_{\mu_\pi}[R_{\text{proxy}}]$$

we can express the solution for $\lambda_3$ as:

$$\boxed{\lambda_3 = -\frac{\sqrt{\chi^2(\mu_\pi \| \mu_{\pi_{\text{ref}}}) - \mathbb{E}_{\mu_\pi}^2[R_{\text{proxy}}]}}{2V\sqrt{1 - r^2}}.} \tag{25}$$

**Solve Outer Maximization Problem.** Now that we have solved for the optimal primal variable $R$ and dual variables $\lambda_1$, $\lambda_2$, and $\lambda_3$, we plug them back into the original max-min objective in Equation 17:

$$\max_\pi \mathbb{E}_{\mu_{\pi_{\text{ref}}}}[L \cdot R] = \max_\pi \mathbb{E}_{\mu_{\pi_{\text{ref}}}}\left[L(s,a) \cdot \frac{L(s,a) - \lambda_1\frac{R_{\text{proxy}}(s,a)}{V} - \lambda_2}{2\lambda_3}\right] \tag{26}$$

Using the earlier substitutions:

$$\lambda_1 = V \cdot \mathbb{E}_{\mu_\pi}[R_{\text{proxy}}] - 2r\lambda_3 V^2,$$
$$\lambda_2 = 1 - 2\lambda_3 M,$$
$$\lambda_3 = -\frac{1}{2} \cdot \frac{\sqrt{\chi^2(\mu_\pi \| \mu_{\pi_{\text{ref}}}) - \mathbb{E}_{\mu_\pi}^2[R_{\text{proxy}}]}}{V\sqrt{1 - r^2}},$$

We simplify the expression:

$$\max_\pi \mathbb{E}_{\mu_{\pi_{\text{ref}}}}[LR] = \max_\pi \frac{1}{2\lambda_3}\left(\mathbb{E}_{\mu_{\pi_{\text{ref}}}}[L^2] - \lambda_1 \mathbb{E}_{\mu_{\pi_{\text{ref}}}}\left[L \cdot \frac{R_{\text{proxy}}}{V}\right] - \lambda_2 \mathbb{E}_{\mu_{\pi_{\text{ref}}}}[L]\right)$$

Recall the identities:

$$\mathbb{E}_{\mu_{\pi_{\text{ref}}}}[L^2] = \chi^2(\mu_\pi \parallel \mu_{\pi_{\text{ref}}}) + 1,$$
$$\mathbb{E}_{\mu_{\pi_{\text{ref}}}}[L \cdot R_{\text{proxy}}] = \mathbb{E}_{\mu_\pi}[R_{\text{proxy}}],$$
$$\mathbb{E}_{\mu_{\pi_{\text{ref}}}}[L] = 1,$$

We substitute these and get:

$$\max_\pi \mathbb{E}_{\mu_{\pi_{\text{ref}}}}[LR] = \frac{1}{2\lambda_3}\left(\chi^2(\mu_\pi \parallel \mu_{\pi_{\text{ref}}}) + 1 - \lambda_1 \cdot \frac{\mathbb{E}_{\mu_\pi}[R_{\text{proxy}}]}{V} - \lambda_2\right)$$

Now substitute the expressions for $\lambda_1$ and $\lambda_2$:

$$\max_\pi \mathbb{E}_{\mu_{\pi_{\text{ref}}}}[LR] = \max_\pi \frac{1}{2\lambda_3}\left(\chi^2 + 1 - (\mathbb{E}_{\mu_\pi}[R_{\text{proxy}}] - 2r\lambda_3 V) \cdot \frac{\mathbb{E}_{\mu_\pi}[R_{\text{proxy}}]}{V} - (1 - 2\lambda_3 M)\right)$$

$$= \max_\pi \frac{1}{2\lambda_3}\left(\chi^2 - \frac{\mathbb{E}^2_{\mu_\pi}[R_{\text{proxy}}]}{V} + 2r\lambda_3 \mathbb{E}_{\mu_\pi}[R_{\text{proxy}}] + 2\lambda_3 M\right)$$

Now cancel out $2\lambda_3$ in numerator and denominator:

$$\max_\pi \mathbb{E}_{\mu_{\pi_{\text{ref}}}}[LR] = \max_\pi \mathbb{E}_{\mu_\pi}[R_{\text{proxy}}] \cdot r - \frac{1}{2\lambda_3} \cdot \left(\frac{\mathbb{E}^2_{\mu_\pi}[R_{\text{proxy}}]}{V} - \chi^2\right) + M$$

Now plug in the expression for $\lambda_3$:

$$\lambda_3 = -\frac{1}{2} \cdot \frac{\sqrt{\chi^2 - \mathbb{E}^2_{\mu_\pi}[R_{\text{proxy}}]}}{V\sqrt{1 - r^2}}$$

This gives the final outer problem for the original max-min objective in Equation 15:

$$\boxed{\max_\pi \quad r \cdot V \cdot \mathbb{E}_{\mu_\pi}[R_{\text{proxy}}] - V \cdot \sqrt{1 - r^2} \cdot \sqrt{\chi^2(\mu_\pi \parallel \mu_{\pi_{\text{ref}}}) - \mathbb{E}^2_{\mu_\pi}[R_{\text{proxy}}]} + M} \quad (27)$$

## E.2 PROOF OF OPTIMALITY

Recall the inner minimization problem of our max-min objective in Equation 17:

$$\min_{R \in \mathcal{R}_{\text{corr}}} \mathbb{E}_{\mu_{\pi_{\text{ref}}}}[L \cdot R]$$

where $L = \mu_\pi(s,a)/\mu_{\pi_{\text{ref}}}(s,a)$ is treated as fixed, and the feasible set is:

$$\mathcal{R}_{\text{corr}} = \left\{ R : (s,a) \to \mathbb{R} \,\middle|\, \mathbb{E}_{\mu_{\pi_{\text{ref}}}}[(R - M) \cdot R_{\text{proxy}}] = r \cdot V, \right.$$

$$\left. \mathbb{E}_{\mu_{\pi_{\text{ref}}}}[R] = M, \quad \mathbb{E}_{\mu_{\pi_{\text{ref}}}}[R^2] = V^2 + M^2 \right\}$$

The feasible region is not convex due to the *quadratic equality* constraint $\mathbb{E}_{\mu_{\pi_{\text{ref}}}}[R^2] = V^2 + M^2$. This defines the boundary of an $L^2$ ball (a hypersphere) in function space, which is not convex. Therefore, traditional convex programming tools and strong duality do not directly apply.

However, we still claim that the resulting $R^*$ derived in Appendix E.1 is globally optimal. This is supported by the following facts:

**Stationarity.** When considering $R^*$ for any fixed dual variables $\lambda_1, \lambda_2, \lambda_3$, we are looking at the inner minimization problem in Equation 18 as follows:

$$\min_{R \in \mathcal{R}_{\mathrm{corr}}} l_0(\lambda_1, \lambda_2, \lambda_3, R)$$

The term with $R(s,a)$ in $l_0$ is:

$$\mathbb{E}_{\mu_{\pi_{\mathrm{ref}}}}\left[L \cdot R - \lambda_1 \frac{R - M}{V} \cdot R_{\mathrm{proxy}} - \lambda_2 R - \lambda_3 R^2\right]$$

For this quadratic in $R$ to have a minimum (since it is a minimization problem for $R$), the coefficient of $R^2$ must be positive. In our case, the coefficient is $-\lambda_3$. Therefore, for the minimization problem to be well-posed and have a finite minimum, we must have $\lambda_3 < 0$. This condition ensures that the quadratic term in $R$ is a concave upward parabola, which means that a minimum exists. Moreover, in Appendix E.1, we explicitly state that $R^*(s,a)$ is derived by setting the derivative of the Lagrangian function $l_0$ in Equation 18 to zero with respect to $R(s,a)$. Thus, $R^*(s,a)$ is indeed the optimal value for the minimization problem for fixed $\lambda_1, \lambda_2, \lambda_3$ where $\lambda_3 < 0$. The Stationarity in this context implies that $R^*$ lies within the domain where the Lagrangian is well-defined and differentiable, which it does.

**Feasibility.** We also argue that the closed-form primal solution $R^*(\lambda^*)$, where $\lambda^*$ denotes the optimal dual solution, is feasible in the original sense, that is, it satisfies the three equality constraints in the feasible set $\mathcal{R}_{\mathrm{corr}}$. Specifically, as detailed in Appendix E.1, we substitute $R^*$ back into the dual objective $l_0$ and compute the gradient with respect to each dual variable. We then solve:

$$\frac{\partial l_0(\lambda_1, \lambda_2, \lambda_3, R^*(\lambda_1, \lambda_2, \lambda_3))}{\partial \lambda_i} = 0, \quad \text{for } i = 1, 2, 3,$$

to find the optimal values $\lambda_1^*, \lambda_2^*, \lambda_3^*$.

By the chain rule, we have:

$$\frac{\partial l_0(\lambda_1, \lambda_2, \lambda_3, R^*(\lambda_1, \lambda_2, \lambda_3))}{\partial \lambda_i} = \left\langle \frac{\partial l_0}{\partial R}, \frac{\partial R^*}{\partial \lambda_i} \right\rangle + \frac{\partial l_0}{\partial \lambda_i},$$

where the first term vanishes because $R^*$ is chosen to minimize $l_0$ for fixed $\lambda$ (i.e., $\partial l_0/\partial R = 0$ at $R^*$). Therefore, the derivative simplifies to:

$$\frac{\partial l_0(\lambda_1, \lambda_2, \lambda_3, R^*)}{\partial \lambda_i} = \frac{\partial l_0}{\partial \lambda_i}.$$

Setting these derivatives to zero yields:

$$\frac{\partial l_0}{\partial \lambda_1} = -\mathbb{E}_{\mu_{\pi_{\mathrm{ref}}}}[(R^* - M)R_{\mathrm{proxy}}] + rV = 0,$$

$$\frac{\partial l_0}{\partial \lambda_2} = -\mathbb{E}_{\mu_{\pi_{\mathrm{ref}}}}[R^*] + M = 0,$$

$$\frac{\partial l_0}{\partial \lambda_3} = -\mathbb{E}_{\mu_{\pi_{\mathrm{ref}}}}[(R^*)^2] + V^2 + M^2 = 0,$$

which exactly recover the original feasibility constraints. Hence, the solution $R^*(\lambda^*)$ is feasible by construction.

Therefore, $R^*$ satisfies both stationarity and feasibility. In general, stationarity and feasibility are not sufficient for global optimality when the feasible set is nonconvex. In our case, however, global optimality does hold, relying on the specific structure of the inner problem.

Recall the inner minimization problem discussed above, and we work in the Hilbert space $\mathcal{H} = L^2(\mu_{\pi_{\mathrm{ref}}})$. Using the normalization assumptions: $\mathbb{E}_{\mu_{\pi_{\mathrm{ref}}}}[R_{\mathrm{proxy}}] = 0$ and $\mathbb{E}_{\mu_{\pi_{\mathrm{ref}}}}[R_{\mathrm{proxy}}^2] = 1$, the constraints can be rewritten as inner products in $\mathcal{H}$:

- $\langle R, \mathbf{1} \rangle = M$ (mean constraint)
- $\langle R, R_{\mathrm{proxy}} \rangle = rV$ (correlation constraint)

- $\|R\|_2^2 = V^2 + M^2$ (norm constraint).

Let $\{e_0, e_1, e_2, \dots\}$ be an orthonormal basis of $\mathcal{H}$, where

- $e_0$ is proportional to the constant function of $\mathbf{1}$
- $e_1 = R_{\text{proxy}}$,
- and $\{e_k\}_{k \geq 2}$, spans the orthogonal complement of span $\{\mathbf{1}, R_{\text{proxy}}\}$.

$e_0$ and $e_1$ is orthonormal because $E_{\mu_{\pi_{\text{ref}}}}[R_{\text{proxy}}] = 0$. Expanding

$$R = \alpha_0 e_0 + \alpha_1 e_1 + \sum_{k \geq 2} \alpha_k e_k$$

Notice that the mean constraint and correlation constraints uniquely fix $\alpha_0$ and $\alpha_1$. The norm constraint then forces:

$$\sum_k \alpha_k = \rho^2$$

for some constant radius $\rho > 0$. Hence the remaining degrees of freedom lie on a sphere in the subspace orthogonal to $\mathbf{1}$ and $R_{\text{proxy}}$. This is to say, although $\mathcal{R}_{\text{corr}}$ is not convex in the ambient space, it is a spherical manifold (the boundary of an $L^2$-ball intersected with an affine subspace), which is compact and smooth. Moreover, the objective is linear in $R$:

$$\mathbb{E}_{\mu_{\pi_{\text{ref}}}}[L \cdot R] = \langle L, R \rangle = \text{const} + \langle L', R' \rangle$$

where $L'$ is the projection of $L = \frac{\mu_\pi}{\mu_{\pi_{\text{ref}}}}$ onto the subspace spanned by $\{e_k\}_{k \geq 2}$ and $R' = \sum_{k \geq 2} \alpha_k e_k$. Therefore the optimization reduces to

$$\min_{\|R'\|_2 = \rho} \langle L', R' \rangle$$

This is simply minimizing a linear function over a Euclidean sphere. In this setting, it is well-known that the only stationary points of a linear functional on a sphere are its global maximum and global minimum. There are no other local minima or saddle points. Thus, on this particular nonconvex feasible set, **any feasible stationary point is automatically a global optimizer.**

In summary, our previous analysis shows that:

1. For fixed $(\lambda_1, \lambda_2, \lambda_3)$ with $\lambda_3 < 0$, the Lagrangian is a strictly convex quadratic in $R$, so its stationary point $R^\star(\lambda)$ is the unique global minimizer of the inner problem with those multipliers.

2. Solving the dual and enforcing feasibility recovers the specific choice of multipliers $\lambda^\star$ for which $R^\star(\lambda^\star)$ lies on the sphere defined by the norm constraint.

3. Because the reduced problem is linear over a sphere, this feasible stationary point $R^\star(\lambda^\star)$ must be the global minimizer of the original inner problem.

### E.3 PROOF THAT $\chi^2(\mu_\pi \,\|\, \mu_{\pi_{\text{REF}}}) \geq \mathbb{E}^2_{\mu_\pi}[R_{\text{PROXY}}]$

To ensure that the inner term of the square root in Equation 25 remains non-negative, we need to show that

$$\chi^2(\mu_\pi \,\|\, \mu_{\pi_{\text{ref}}}) \geq \mathbb{E}^2_{\mu_\pi}[R_{\text{proxy}}]$$

**Proof.** Recall that

$$\mathbb{E}_{\mu_\pi}[R_{\text{proxy}}] = \mathbb{E}_{\mu_{\pi_{\text{ref}}}}[L \cdot R_{\text{proxy}}],$$

where $L(s, a) = \frac{\mu_\pi(s,a)}{\mu_{\pi_{\text{ref}}}(s,a)}$ is the Radon-Nikodym derivative. Since $R_{\text{proxy}}$ is normalized to have zero mean under $\mu_{\pi_{\text{ref}}}$, we have:

$$\mathbb{E}_{\mu_{\pi_{\text{ref}}}}[R_{\text{proxy}}] = 0.$$

Thus,

$$\mathbb{E}_{\mu_\pi}[R_{\text{proxy}}] = \mathbb{E}_{\mu_{\pi_{\text{ref}}}}[R_{\text{proxy}}(s,a)(L(s,a)-1)]$$
$$= \sum_{(s,a)} R_{\text{proxy}}(s,a)\mu_{\pi_{\text{ref}}}(s,a)(L(s,a)-1)$$

Applying the Cauchy-Schwarz inequality:

$$\left(\sum_{(s,a)} R_{\text{proxy}}(s,a)\mu_{\pi_{\text{ref}}}(s,a)(L(s,a)-1)\right)^2 \leq \left(\sum_{(s,a)} \mu_{\pi_{\text{ref}}}(s,a)R_{\text{proxy}}^2(s,a)\right)\left(\sum_{(s,a)} \mu_{\pi_{\text{ref}}}(s,a)(L(s,a)-1)^2\right)$$

By the assumptions: $\mathbb{E}_{\mu_{\pi_{\text{ref}}}}[R_{\text{proxy}}^2] = 1$, $\sum_{(s,a)} \mu_{\pi_{\text{ref}}}(s,a)(L(s,a)-1)^2 = \chi^2(\mu_\pi \| \mu_{\pi_{\text{ref}}})$.

We obtain:

$$\mathbb{E}_{\mu_\pi}^2[R_{\text{proxy}}] \leq \chi^2(\mu_\pi \| \mu_{\pi_{\text{ref}}})$$

as desired.

### E.4 Derive Lagrangian Functional for Linear Max-Min Objective

Recall that our max-min optimization under the structured reward assumption is as follows:

$$\max_\pi \min_{\boldsymbol{\theta} \in \mathcal{R}_{\text{corr}}^{\text{lin}}, \boldsymbol{\theta} \geq 0} \mathbb{E}_{(s,a)\sim\mu_\pi}\left[\boldsymbol{\theta}^\top \boldsymbol{\phi}(s,a)\right]. \tag{28}$$

where $\mathcal{R}_{\text{corr}}^{\text{lin}}$ is the uncertainty set defined as follow:

$$\mathcal{R}_{\text{corr}}^{\text{lin}} = \left\{\boldsymbol{\theta} \in \mathbb{R}^M \,\middle|\, \mathbb{E}_{\mu_{\pi_{\text{ref}}}}[\boldsymbol{\theta}^\top \boldsymbol{\phi} \cdot R_{\text{proxy}}] = r,\ \mathbb{E}_{\mu_{\pi_{\text{ref}}}}[\boldsymbol{\theta}^\top \boldsymbol{\phi}] = 0,\ \mathbb{E}_{\mu_{\pi_{\text{ref}}}}[(\boldsymbol{\theta}^\top \boldsymbol{\phi})^2] = 1\right\}. \tag{29}$$

We assume without loss of generality that the worst-case reward $R(s,a) = \boldsymbol{\theta}^\top \boldsymbol{\phi}(s,a)$ is normalized to have zero mean and unit variance under the reference policy $\pi_{\text{ref}}$. This corresponds to setting $M = 0$ and $V = 1$, which, as shown in our earlier derivation, does not affect the resulting optimal policy. As before, $R_{\text{proxy}}$ denotes the normalized proxy reward under $\pi_{\text{ref}}$, satisfying $\mathbb{E}_{\mu_{\pi_{\text{ref}}}}[R_{\text{proxy}}] = 0$ and $\text{Var}_{\mu_{\pi_{\text{ref}}}}[R_{\text{proxy}}] = 1$.

Similar to previous steps, we introduce the Radon-Nikodym derivative

$$L(s,a) = \frac{\mu_\pi(s,a)}{\mu_{\pi_{\text{ref}}}(s,a)}$$

We use a change-of-measure, and define the Lagrangian functional for the inner minimization in Equation 28 as:

$$l_1(\lambda_1,\lambda_2,\lambda_3,\boldsymbol{\theta}) = \mathbb{E}_{\mu_{\pi_{\text{ref}}}}\left[L \cdot \boldsymbol{\theta}^\top \boldsymbol{\phi}\right] - \lambda_1\left(\mathbb{E}_{\mu_{\pi_{\text{ref}}}}\left[R_{\text{proxy}} \cdot \boldsymbol{\theta}^\top \boldsymbol{\phi}\right] - r\right)$$
$$- \lambda_2 \mathbb{E}_{\mu_{\pi_{\text{ref}}}}\left[\boldsymbol{\theta}^\top \boldsymbol{\phi}\right] - \lambda_3\left(\mathbb{E}_{\mu_{\pi_{\text{ref}}}}\left[(\boldsymbol{\theta}^\top \boldsymbol{\phi})^2\right] - 1\right)$$
$$= \mathbb{E}_{\mu_{\pi_{\text{ref}}}}\left[(L - \lambda_1 R_{\text{proxy}} - \lambda_2)\boldsymbol{\theta}^\top \boldsymbol{\phi}\right] - \lambda_3 \mathbb{E}_{\mu_{\pi_{\text{ref}}}}\left[(\boldsymbol{\theta}^\top \boldsymbol{\phi})^2\right] + \lambda_1 r + \lambda_3$$
$$= \sum_{(s,a)} \mu_{\pi_{\text{ref}}}(s,a)\left[(L - \lambda_1 R_{\text{proxy}} - \lambda_2)\boldsymbol{\theta}^\top \boldsymbol{\phi} - (\boldsymbol{\theta}^\top \boldsymbol{\phi})^2\right] + \lambda_1 r + \lambda_3$$

Define the following terms for simplicity:

$$v(s,a) = \mu_\pi(s,a)$$
$$D(s,a) = \mu_{\pi_{\text{ref}}}(s,a) \cdot R_{\text{proxy}}(s,a)$$
$$C(s,a) = \mu_{\pi_{\text{ref}}}(s,a)$$
$$u_{\lambda_1,\lambda_2}(s,a) = v(s,a) - \lambda_1 D(s,a) - \lambda_2 C(s,a)$$

Then the Lagrangian function simplifies to:

$$l_1(\lambda_1, \lambda_2, \lambda_3, \boldsymbol{\theta}) = \sum_{(s,a)} \left[ u_{\lambda_1, \lambda_2}(s,a) \boldsymbol{\theta}^\top \boldsymbol{\phi}(s,a) - \lambda_3 C(s,a)(\boldsymbol{\theta}^\top \boldsymbol{\phi}(s,a))^2 \right] + \lambda_1 r + \lambda_3$$

$$= \boldsymbol{\theta}^\top \left( \sum_{(s,a)} u_{\lambda_1, \lambda_2}(s,a) \boldsymbol{\phi}(s,a) \right) - \lambda_3 \boldsymbol{\theta}^\top \left( \sum_{(s,a)} C(s,a) \boldsymbol{\phi}(s,a) \boldsymbol{\phi}(s,a)^\top \right) \boldsymbol{\theta} + \lambda_1 r + \lambda_3$$

where we expand the quadratic term:

$$\sum_{(s,a)} C(s,a)(\boldsymbol{\theta}^\top \boldsymbol{\phi}(s,a))^2 = \sum_{(s,a)} C(s,a)(\boldsymbol{\phi}(s,a)^\top \boldsymbol{\theta})^2$$

$$= \sum_{(s,a)} C(s,a) \boldsymbol{\theta}^\top \boldsymbol{\phi}(s,a) \boldsymbol{\phi}(s,a)^\top \boldsymbol{\theta}$$

$$= \boldsymbol{\theta}^\top \left( \sum_{(s,a)} C(s,a) \boldsymbol{\phi}(s,a) \boldsymbol{\phi}(s,a)^\top \right) \boldsymbol{\theta}$$

Let

$$Q = \sum_{(s,a)} C(s,a) \boldsymbol{\phi}(s,a) \boldsymbol{\phi}(s,a)^\top \tag{30}$$

then we can write the Lagrangian function as:

$$l_1(\lambda_1, \lambda_2, \lambda_3, \boldsymbol{\theta}) = \boldsymbol{\theta}^\top \left( \sum_{(s,a)} u_{\lambda_1, \lambda_2}(s,a) \boldsymbol{\phi}(s,a) \right) - \lambda_3 \boldsymbol{\theta}^\top Q \boldsymbol{\theta} + \lambda_1 r + \lambda_3$$

And the inner minimization problem in Equation 28 becomes:

$$\max_{\lambda_1, \lambda_2, \lambda_3} \min_{\boldsymbol{\theta} \geq 0} l_1(\lambda_1, \lambda_2, \lambda_3, R) \tag{31}$$

### E.5 PROOF FOR WHITENING TRANSFORMATION

To simplify the problem associated with the Lagrangian function above, we transform the feature vector $\boldsymbol{\phi}$ into a whitened version $\tilde{\boldsymbol{\phi}}$ such that the matrix $Q$ as defined in Equation 30 becomes the identity matrix $I$. Specifically, we perform a whitening transformation using the Cholesky decomposition (Boyd & Vandenberghe, 2004). Let

$$W = Q^{-\frac{1}{2}}, \quad \tilde{\boldsymbol{\phi}}(s,a) = W \boldsymbol{\phi}(s,a)$$

where $Q^{-\frac{1}{2}}$ denotes a matrix square root of $Q^{-1}$. Then we have:

$$\sum_{(s,a)} C(s,a) \tilde{\boldsymbol{\phi}}(s,a) \tilde{\boldsymbol{\phi}}(s,a)^\top = \sum_{(s,a)} C(s,a)(W\boldsymbol{\phi}(s,a))(W\boldsymbol{\phi}(s,a))^\top$$

$$= \sum_{(s,a)} C(s,a) W \boldsymbol{\phi}(s,a) \boldsymbol{\phi}(s,a)^\top W^\top$$

$$= W \left( \sum_{(s,a)} C(s,a) \boldsymbol{\phi}(s,a) \boldsymbol{\phi}(s,a)^\top \right) W^\top$$

$$= W Q W^\top$$

$$= Q^{-\frac{1}{2}} Q Q^{-\frac{1}{2}}$$

$$= I$$

as desired.

Note that the whitening step requires $Q$ to be invertible so that $Q^{-1/2}$ (and hence $Q^{-1}$) exists. It holds when $Q$ is positive semi-definite and non-singular. $Q$ is positive semi-definite since it is a sum of outer products $\phi(s,a)\phi(s,a)^\top$ weighted by non-negative coefficients (occupancy measure of $\pi_{\text{ref}} \geq 0$). For $Q$ to be non-singular, it is necessary that the span of $\{\phi(s,a) : \mu_{\pi_{\text{ref}}}(s,a) > 0\}$ covers $\mathbb{R}^n$, i.e., the features associated with state-action pairs visited by $\pi_{\text{ref}}$ must span the full feature space. To achieve these conditions, the reference policy should visit a diverse and representative subset of the state-action space with non-trivial occupancy. This is more likely when $\pi_{\text{ref}}$ is derived from either expert demonstrations that exhibit rich behavior or from stochastic or exploratory policies (e.g., entropy-regularized policies or policies trained with exploration bonuses). Moreover, the feature mapping $\phi(s,a)$ must exhibit sufficient variation across the visited state-action pairs. This typically holds when $\phi$ encodes task-relevant dynamics (e.g., learned embeddings or expressive hand-crafted features) and when $\pi_{\text{ref}}$ does not collapse to trivial or deterministic behavior. In our experiments (Appendix F.4), the reference policies for the Traffic and Pandemic environments are trained via behavioral cloning on large, diverse trajectories generated by human experts or hand-crafted controllers. The feature representations used in these environments, such as velocity, acceleration, and headway in Traffic, and infection level, disease stage, and smooth transitions in Pandemic, encode meaningful task-relevant dynamics. These demonstrations cover a wide range of task-relevant behaviors, and the induced occupancy over state-action pairs spans a high-dimensional subspace of the feature space. We empirically verified that the resulting $Q$ matrices in our experiments are full-rank and numerically well-conditioned. Though ensuring sufficient coverage of the feature space by the reference policy is generally challenging in practice.

### E.6 DERIVE OPTIMAL PRIMAL VARIABLE FOR LINEAR MAX-MIN OBJECTIVE

After whitening transformation as discussed in Appendix E.5, the problem in Equation 31 becomes:

$$\max_{\lambda_1,\lambda_2,\lambda_3} \min_{\tilde{\boldsymbol{\theta}} \geq 0} \quad l_1(\lambda_1,\lambda_2,\lambda_3,\tilde{\boldsymbol{\theta}}) = \tilde{\boldsymbol{\theta}}^\top \left( \sum_{(s,a)} u_{\lambda_1,\lambda_2}(s,a)\tilde{\phi}(s,a) \right) - \lambda_3\tilde{\boldsymbol{\theta}}^\top\tilde{\boldsymbol{\theta}} + \lambda_1 r + \lambda_3. \quad (32)$$

where we now optimize over the parameter $\tilde{\boldsymbol{\theta}}$ using the transformed features $\tilde{\phi}$. For notational simplicity, we will drop the tilde and henceforth use $\phi$ to represent the whitened feature $\tilde{\phi}$, and $\boldsymbol{\theta}$ to represent the whitened weights $\tilde{\boldsymbol{\theta}}$.

**Separable Structure.** In the whitened feature space, the objective becomes separable across coordinates of $\boldsymbol{\theta}$. Thus, the inner minimization problem in Equation 32 decouples into $M$ independent one-dimensional convex minimization problems, one for each feature coordinate $i \in \{1, 2, \dots, M\}$:

$$\min_{\theta_i \geq 0} \quad \left( \sum_{(s,a)} u_{\lambda_1,\lambda_2}(s,a)\phi_i(s,a) \right) \theta_i - \lambda_3\theta_i^2$$

Let us solve the $i$-th subproblem. Assuming $\lambda_3 < 0$, the objective is a convex quadratic function in $\theta_i$ (an upward-opening parabola). The unconstrained minimum occurs at:

$$\theta_i^* = -\frac{\sum_{(s,a)} u_{\lambda_1,\lambda_2}(s,a)\phi_i(s,a)}{2\lambda_3}$$

Considering the constraint $\theta_i \geq 0$, we have two cases:

- If the unconstrained minimum $\theta_i^* \geq 0$, then it is also the solution to the constrained problem.
- If $\theta_i^* < 0$, then the constrained minimum occurs at the boundary $\theta_i = 0$.

Thus, the final optimal $\theta_i^*$ is:

$$\theta_i^* = \max\left( 0, -\frac{\sum_{(s,a)} u_{\lambda_1,\lambda_2}(s,a)\phi_i(s,a)}{2\lambda_3} \right)$$

Collecting across all $i$, we express the final optimal solution $\boldsymbol{\theta}^*$ as:

$$\boldsymbol{\theta}^* = \max\left(0, -\frac{\sum_{(s,a)} u_{\lambda_1,\lambda_2}(s,a)\boldsymbol{\phi}(s,a)}{2\lambda_3}\right) \tag{33}$$

where the $\max(\cdot, 0)$ is applied elementwise.

### E.7 SOLVE THE DUAL OBJECTIVE FOR LINEAR MAX-MIN OBJECTIVE

Let the outer objective in Equation 32 be:

$$g(\lambda_1, \lambda_2, \lambda_3) = l_1(\lambda_1, \lambda_2, \lambda_3, \boldsymbol{\theta}^*)$$

Then we want to solve the following dual objective:

$$\max_{\lambda_1,\lambda_2,\lambda_3} g(\lambda_1, \lambda_2, \lambda_3) \tag{34}$$

Let

$$q_j(\lambda_1, \lambda_2) = \sum_{(s,a)} \left(v(s,a) - \lambda_1 D(s,a) - \lambda_2 C(s,a)\right) \phi_j(s,a)$$

denote the linear coefficient for each feature $j \in \{1, \ldots, M\}$.

The optimal $\theta_j^*$ is:

$$\theta_j^*(\lambda) = \max\left(0, \frac{q_j(\lambda_1, \lambda_2)}{2\lambda_3}\right)$$

Now, we compute the gradients:

**Gradient with respect to $\lambda_1$:**

$$\begin{aligned}
\frac{\partial g}{\partial \lambda_1}(\lambda) &= \frac{\partial l_1}{\partial \lambda_1}(\lambda, \boldsymbol{\theta}^*(\lambda)) \\
&= \sum_{(s,a)} \left(-D(s,a)(\boldsymbol{\theta}^{*T}\boldsymbol{\phi}(s,a))\right) + r \\
&= r - \sum_{j=1}^{M} D_{\phi,j} \cdot \theta_j^*(\lambda)
\end{aligned}$$

where

$$D_{\phi,j} = \sum_{(s,a)} D(s,a)\phi_j(s,a)$$

**Gradient with respect to $\lambda_2$:**

$$\begin{aligned}
\frac{\partial g}{\partial \lambda_2}(\lambda) &= \frac{\partial l_1}{\partial \lambda_2}(\lambda, \boldsymbol{\theta}^*(\lambda)) \\
&= \sum_{(s,a)} \left(-C(s,a)(\boldsymbol{\theta}^{*T}\boldsymbol{\phi}(s,a))\right) \\
&= -\sum_{j=1}^{M} C_{\phi,j} \cdot \theta_j^*(\lambda)
\end{aligned}$$

where

$$C_{\phi,j} = \sum_{(s,a)} C(s,a)\phi_j(s,a)$$

**Gradient with respect to $\lambda_3$:**

$$\frac{\partial g}{\partial \lambda_3}(\lambda) = \frac{\partial l_1}{\partial \lambda_3}(\lambda, \boldsymbol{\theta}^*(\lambda))$$

$$= \sum_{(s,a)} \left(-C(s,a)(\boldsymbol{\theta}^{*T}\boldsymbol{\phi}(s,a))^2\right) + 1$$

$$= 1 - \sum_{j=1}^{M}(\theta_j^*(\lambda))^2$$

where we use the whitening assumption $\sum_{(s,a)} C(s,a)\boldsymbol{\phi}(s,a)\boldsymbol{\phi}(s,a)^\top = I$.

Thus, the full gradients are:

$$\frac{\partial g}{\partial \lambda_1}(\lambda) = r - \sum_{j=1}^{M} D_{\phi,j} \cdot \max\left(0, \frac{q_j(\lambda_1, \lambda_2)}{2\lambda_3}\right)$$

$$\frac{\partial g}{\partial \lambda_2}(\lambda) = -\sum_{j=1}^{M} C_{\phi,j} \cdot \max\left(0, \frac{q_j(\lambda_1, \lambda_2)}{2\lambda_3}\right)$$

$$\frac{\partial g}{\partial \lambda_3}(\lambda) = 1 - \sum_{j=1}^{M} \left(\max\left(0, \frac{q_j(\lambda_1, \lambda_2)}{2\lambda_3}\right)\right)^2$$

We can solve for the optimal dual variables $(\lambda_1, \lambda_2, \lambda_3)$ using standard first-order optimization methods. Since $g(\lambda)$ is concave (under the condition $\lambda_3 < 0$), optimization is well-behaved and converges reliably. After obtaining the optimal primal variables $\boldsymbol{\theta}^*$ and dual variables $(\lambda_1^*, \lambda_2^*, \lambda_3^*)$, we can substitute them back into Equation 28 and solve the outer maximization over the policy $\pi$ using standard reinforcement learning algorithms, such as PPO (Schulman et al., 2017).

### E.8 POLICY GRADIENT DERIVATION

We now derive the gradient of the robust objective equation 27 with respect to the policy parameters $\theta$. Recall that the robust objective is:

$$\mathcal{J}(\mu_{\pi_\theta}) = r \cdot V \cdot \mathbb{E}_{\mu_{\pi_\theta}}[R_{\text{proxy}}] - V \cdot \sqrt{1-r^2} \cdot \sqrt{\chi^2(\mu_\pi \,\|\, \mu_{\pi_{\text{ref}}}) - \left(\mathbb{E}_{\mu_{\pi_\theta}}[R_{\text{proxy}}]\right)^2} + M$$

$$= \mathbb{E}_{\mu_{\pi_\theta}}[R_{\text{proxy}}] - \frac{\sqrt{1-r^2}}{r}\sqrt{\chi^2(\mu_{\pi_\theta} \,\|\, \mu_{\pi_{\text{ref}}}) - \left(\mathbb{E}_{\mu_{\pi_\theta}}[R_{\text{proxy}}]\right)^2}$$

where we set $M = 0$ and $V = 1$ without loss of generality. We also divide the entire objective by $r$, which is assumed to be positive ($r > 0$), so this rescaling preserves the optimization direction and does not affect the final policy solution. The $\chi^2$ divergence is defined as:

$$\chi^2(\mu_{\pi_\theta} \,\|\, \mu_{\pi_{\text{ref}}}) = \sum_{(s,a)} \frac{\mu_{\pi_\theta}(s,a)^2}{\mu_{\pi_{\text{ref}}}(s,a)} - 1$$

Applying the chain rule, we compute:

$$\nabla_\theta \mathcal{J} = \nabla_\theta \mathbb{E}_{\mu_{\pi_\theta}}[R_{\text{proxy}}] - \frac{\sqrt{1-r^2}}{r}\nabla_\theta\left(\sqrt{h(\mu_{\pi_\theta})}\right) \tag{35}$$

where we define:

$$h(\mu_{\pi_\theta}) = \chi^2(\mu_{\pi_\theta} \,\|\, \mu_{\pi_{\text{ref}}}) - \left(\mathbb{E}_{\mu_{\pi_\theta}}[R_{\text{proxy}}]\right)^2$$

Using the chain rule again:

$$\nabla_\theta \sqrt{h(\mu_{\pi_\theta})} = \frac{1}{2\sqrt{h(\mu_{\pi_\theta})}}\nabla_\theta h(\mu_{\pi_\theta})$$

Now compute $\nabla_\theta h(\mu_{\pi_\theta})$:

$$\nabla_\theta h(\mu_{\pi_\theta}) = \nabla_\theta \chi^2(\mu_{\pi_\theta} \| \mu_{\pi_{\text{ref}}}) - \nabla_\theta \left( \mathbb{E}_{\mu_{\pi_\theta}}[R_{\text{proxy}}]^2 \right)$$

$$= \nabla_\theta \left( \sum_{(s,a)} \frac{\mu_{\pi_\theta}(s,a)^2}{\mu_{\pi_{\text{ref}}}(s,a)} - 1 \right) - 2\mathbb{E}_{\mu_{\pi_\theta}}[R_{\text{proxy}}]\nabla_\theta \mathbb{E}_{\mu_{\pi_\theta}}[R_{\text{proxy}}]$$

The individual terms are:

$$\nabla_\theta \left( \sum_{(s,a)} \frac{\mu_{\pi_\theta}(s,a)^2}{\mu_{\pi_{\text{ref}}}(s,a)} - 1 \right) = 2 \sum_{(s,a)} \frac{\mu_{\pi_\theta}(s,a)}{\mu_{\pi_{\text{ref}}}(s,a)} \nabla_\theta \mu_{\pi_\theta}(s,a)$$

$$\nabla_\theta \mathbb{E}_{\mu_{\pi_\theta}}[R_{\text{proxy}}] = \sum_{(s,a)} \nabla_\theta \mu_{\pi_\theta}(s,a) R_{\text{proxy}}(s,a)$$

Thus:

$$\nabla_\theta h(\mu_{\pi_\theta}) = \sum_{(s,a)} \nabla_\theta \mu_{\pi_\theta}(s,a) \left( 2\frac{\mu_{\pi_\theta}(s,a)}{\mu_{\pi_{\text{ref}}}(s,a)} - 2\mathbb{E}_{\mu_{\pi_\theta}}[R_{\text{proxy}}]R_{\text{proxy}}(s,a) \right)$$

Then we compute $\nabla_\theta \mathbb{E}_{\mu_{\pi_\theta}}[R_{\text{proxy}}]$:

$$\nabla_\theta \mathbb{E}_{\mu_{\pi_\theta}}[R_{\text{proxy}}] = \nabla_\theta \sum_{(s,a)} \mu_{\pi_\theta}(s,a) R_{\text{proxy}}(s,a)$$

$$= \sum_{(s,a)} \nabla_\theta \mu_{\pi_\theta}(s,a) R_{\text{proxy}}(s,a)$$

Put them together, we get the final gradient in Equation 35 as:

$$\nabla_\theta \mathcal{J} = \nabla_\theta \mathbb{E}_{\mu_{\pi_\theta}}[R_{\text{proxy}}] - \frac{\sqrt{1-r^2}}{r} \nabla_\theta \left( \sqrt{h(\mu_{\pi_\theta})} \right)$$

$$= \sum_{(s,a)} \nabla_\theta \mu_{\pi_\theta}(s,a) R_{\text{proxy}}(s,a) - \frac{\sqrt{1-r^2}}{r} \frac{1}{2\sqrt{h(\mu_{\pi_\theta})}} \nabla_\theta h(\mu_{\pi_\theta})$$

$$= \sum_{(s,a)} \nabla_\theta \mu_{\pi_\theta}(s,a) R_{\text{proxy}}(s,a)$$

$$- \frac{\sqrt{1-r^2}}{r} \frac{1}{2\sqrt{h(\mu_{\pi_\theta})}} \sum_{(s,a)} \nabla_\theta \mu_{\pi_\theta}(s,a) \left( 2\frac{\mu_{\pi_\theta}(s,a)}{\mu_{\pi_{\text{ref}}}(s,a)} - 2\mathbb{E}_{\mu_{\pi_\theta}}[R_{\text{proxy}}]R_{\text{proxy}}(s,a) \right)$$

$$= \sum_{(s,a)} \nabla_\theta \mu_{\pi_\theta}(s,a) \left[ R_{\text{proxy}} - \frac{\sqrt{1-r^2}}{r} \frac{1}{\sqrt{h(\mu_{\pi_\theta})}} \left( \frac{\mu_{\pi_\theta}(s,a)}{\mu_{\pi_{\text{ref}}}(s,a)} - \mathbb{E}_{\mu_{\pi_\theta}}[R_{\text{proxy}}]R_{\text{proxy}}(s,a) \right) \right]$$

$$(36)$$

The full policy gradient for the ORPO algorithm, as presented in Appendix B of (Laidlaw et al., 2025), is given by:

$$\sum_{(s,a)} \nabla_\theta \mu_{\pi_\theta}(s,a) \left[ R_{\text{proxy}}(s,a) - \frac{\lambda}{\sqrt{\chi^2(\mu_{\pi_\theta} \| \mu_{\pi_{\text{ref}}})}} \cdot \frac{\mu_{\pi_\theta}(s,a)}{\mu_{\pi_{\text{ref}}}(s,a)} \right]. \tag{37}$$

**Interpretation.** The policy gradient consists of two terms:

- A standard term encouraging the policy to increase $R_{\text{proxy}}(s,a)$.
- A correction term that penalizes deviations from the reference occupancy $\mu_{\pi_{\text{ref}}}$, while also adjusting for alignment with the proxy reward.

This correction enforces robustness to potential reward hacking by optimizing against adversarially misaligned interpretations of the proxy reward.

Notice that our derived policy gradient in Equation 36 shares structural similarities with ORPO but is rooted in a formal robust optimization framework. Unlike ORPO, our formulation introduces an additional correction term involving both the occupancy ratio and the expected proxy reward, capturing how the proxy is aligned with the current policy's behavior. This structure more explicitly penalizes the combination of distributional shift and proxy overoptimization, discouraging policies from exploiting proxy-specific artifacts. Both methods share the goal of improving robustness, but our approach is derived from first principles by directly optimizing for worst-case performance over a correlation-constrained uncertainty set.

### E.9   PROOF OF THEOREM 1

*Proof.* For any reward function $R$, define the performance difference

$$\Delta J(\pi, R) \ := \ J(\pi, R) - J(\pi_{\text{ref}}, R).$$

By definition of the correlated uncertainty set, our distributionally robust objective considers

$$F(\pi) \ := \ \min_{R \in \mathcal{R}_{\text{corr}}} \Delta J(\pi, R).$$

Under the assumptions on the correlation, mean, and variance of rewards in $\mathcal{R}_{\text{corr}}$, Equation 27 provides a closed-form expression for this inner minimum. In particular, for any policy $\pi$ with $\mu_\pi \ll \mu_{\pi_{\text{ref}}}$, Equation 27 gives

$$F(\pi) \ = \ r \cdot \mathbb{E}_{\mu_\pi}[R_{\text{proxy}}] - \sqrt{1 - r^2} \sqrt{\chi^2\big(\mu_\pi \,\|\, \mu_{\pi_{\text{ref}}}\big) - \mathbb{E}^2_{\mu_\pi}[R_{\text{proxy}}]}.$$

Now assume that the true reward $R_{\text{true}}$ lies in $\mathcal{R}_{\text{corr}}$. Since $R_{\text{true}}$ is one feasible element of the uncertainty set, we must have

$$F(\pi) \ = \ \min_{R \in \mathcal{R}_{\text{corr}}} \Delta J(\pi, R) \ \leq \ \Delta J(\pi, R_{\text{true}}) \ = \ J(\pi, R_{\text{true}}) - J(\pi_{\text{ref}}, R_{\text{true}}).$$

Rearranging yields

$$J(\pi, R_{\text{true}}) - J(\pi_{\text{ref}}, R_{\text{true}}) \ \geq \ F(\pi),$$

and substituting the explicit form of $F(\pi)$ from Equation 27 gives the claimed inequality.   □

## F   ADDITIONAL IMPLEMENTATION DETAILS

### F.1   TRAINING DISCRIMINATOR NETWORK

A core step in our Max-Min optimization algorithm and ORPO is to estimate the Radon-Nikodym derivative $L(s, a)$, which is critical for computing the $\chi^2$ divergence, as detailed in Appendix F.2. To this end, we follow prior works (Laidlaw et al., 2025; Kang et al., 2018; Ho & Ermon, 2016) and train a discriminator network. Specifically, we sample a batch of trajectories $D_{\pi_{\text{ref}}}$ from the reference policy $\pi_{\text{ref}}$ and another batch $D_\pi$ from the current policy $\pi$. The batch sizes used for each are specified in Table 2. And then we use a discriminator architecture identical to that in (Laidlaw et al., 2025), denoted by $d_\phi(s, a)$, which is optimized according to:

$$\phi = \arg\min_\phi \ \mathbb{E}_{\mu_{\pi_{\text{ref}}}}[\log(1 + e^{d_\phi(s,a)})] + \mathbb{E}_{\mu_\pi}[\log(1 + e^{-d_\phi(s,a)})]$$

$$\approx \arg\min_\phi \ \mathbb{E}_{D_{\pi_{\text{ref}}}}[\log(1 + e^{d_\phi(s,a)})] + \mathbb{E}_{D_\pi}[\log(1 + e^{-d_\phi(s,a)})] \tag{38}$$

It is known that the optimal discriminator satisfies $d^*(s, a) = \log \frac{\mu_\pi(s,a)}{\mu_{\pi_{\text{ref}}}(s,a)}$ and we estimate $L(s, a)$ as $\tilde{L}(s, a) = e^{d_\phi(s,a)}$ with $d_\phi(s, a) \approx d^*(s, a)$. However, in the original ORPO implementation[1],

---

[1] https://github.com/cassidylaidlaw/orpo/tree/main

the discriminator is not fully optimized during policy learning. Specifically, the discriminator receives only a small number of gradient updates per reinforcement learning iteration, resulting in underfitting and inaccurate estimates of the Radon-Nikodym derivative $L(s, a)$.

This undertraining is evident in Figure 2, which shows the discriminator loss across RL iterations. The loss remains nearly constant (e.g., around 1.4 in the Traffic environment, which is the initial loss value as shown in Figure 3a ), indicating that the discriminator is not learning effectively. This limits its ability to distinguish between $\pi$ and $\pi_{\text{ref}}$, especially for state-action pairs where their occupancy distributions diverge.

To address this, we substantially increase the number of gradient updates per iteration and carefully tune the learning rate. Our goal is to strike a practical balance between training time and discriminator quality: while fully training the discriminator to convergence each iteration is computationally expensive, insufficient training leads to inaccurate divergence estimates and unstable optimization.

Figure 3 shows that in our implementation, the discriminator loss consistently decreases within each iteration, e.g., from an initial value around 1.4 to below 0.2 in the Traffic environment, indicating effective optimization and more accurate occupancy-ratio estimation. In the Glucose and Pandemic environments, however, we observe that training the discriminator for too long leads to slower convergence and little improvement in loss. In these cases, we apply early stopping to limit training time. The specific training schedules are provided in Table 2.

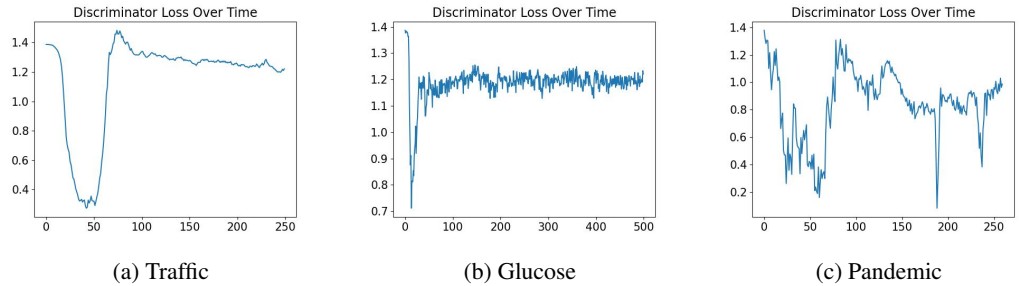

(a) Traffic        (b) Glucose        (c) Pandemic

Figure 2: Discriminator loss across RL iterations in the **original ORPO implementation**. The loss stays flat and high ($\sim$1.4 for the traffic environment), indicating the discriminator is not adequately trained. This undermines the accuracy of the estimated occupancy ratios.

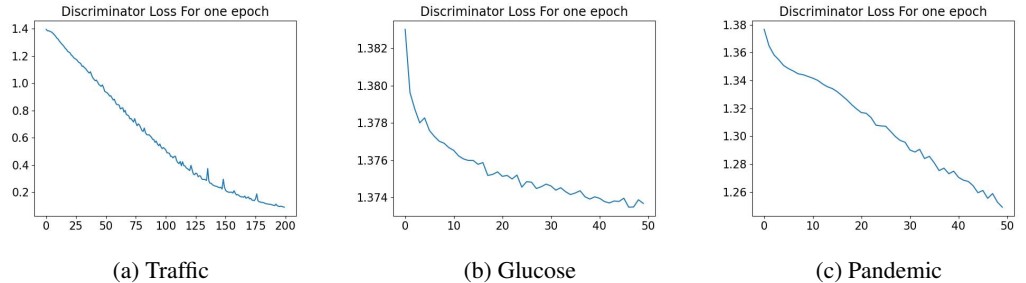

(a) Traffic        (b) Glucose        (c) Pandemic

Figure 3: Discriminator loss over training steps *within each RL iteration* in our implementation. The loss decreases rapidly from its initial value (e.g., 1.4 to values near 0.2 in the Traffic environment), indicating successful training and improved accuracy of occupancy-ratio estimates.

As for the discriminator network architecture, we follow the same structure described in (Laidlaw et al., 2025). For each environment, we employ a fully connected neural network with two hidden layers, each consisting of 256 units and ReLU activations. Table 2 summarizes the hyperparameters used for discriminator training across different environments.

Table 2: Hyperparameters used for discriminator network training across different environments.

| Hyperparameter | Traffic | Glucose | Pandemic |
|---|---|---|---|
| Learning rate | $5 \times 10^{-3}$ | $1 \times 10^{-2}$ | $5 \times 10^{-4}$ |
| SGD epochs per iteration | 200 | 20 | 15 |
| Batch size | 40000 | 100000 | 3860 |
| SGD minibatch size | 16384 | 1024 | 64 |

### F.2 DERIVATION OF MAX-MIN POLICY OPTIMIZATION

Using the estimated $d_\phi(s, a)$ from trained discriminator as discussed in Appendix F.1, we can compute the $\chi^2$ divergence via:

$$\chi^2(\mu_\pi \parallel \mu_{\pi_{\text{ref}}}) = \mathbb{E}_{\mu_\pi} \left[ \frac{\mu_\pi(s, a)}{\mu_{\pi_{\text{ref}}}(s, a)} - 1 \right] \approx \mathbb{E}_{D_\pi} \left[ e^{d_\phi(s, a)} - 1 \right]. \tag{39}$$

For environments where both state and action spaces are discrete, we directly estimate the occupancy measure via empirical sampling. Specifically, given the same batch of trajectories $D$ collected from policy $\pi$ (as used for training the discriminator), we approximate the discounted occupancy measure as follows (Schlaginhaufen & Kamgarpour, 2023; Abbeel & Ng, 2004):

$$\tilde{\mu}_\pi^D(s, a) := (1 - \gamma) \frac{1}{N} \sum_{i=1}^{N} \sum_{t=0}^{T} \gamma^t \mathbb{I}\{s_t^i = s, a_t^i = a\}, \tag{40}$$

where $\mathbb{I}\{\cdot\}$ is the indicator function. Using this empirical estimate, we can compute the Radon-Nikodym derivative and $\chi^2$ divergence without training a discriminator.

In our formulation, we assume that the proxy reward is normalized with respect to the reference policy $\pi_{\text{ref}}$. To achieve this, we reuse the same batch of trajectories $D_{\pi_{\text{ref}}}$ sampled from $\pi_{\text{ref}}$ to estimate the expected return $\tilde{J}(\pi_{\text{ref}}, R_{\text{proxy}})$ using:

$$\tilde{J}(\pi_{\text{ref}}, R_{\text{proxy}}) = (1 - \gamma) \frac{1}{N} \sum_{i=1}^{N} R_{\text{proxy}}(\tau^{(i)}) \tag{41}$$

where each $\tau^{(i)} = (s_0, a_0, s_1, a_1, ..., s_T) \sim D_{\pi_{\text{ref}}}$ is a trajectory sampled from $\pi_{\text{ref}}$, $N$ is the number of sampled trajectories, and $R_{\text{proxy}}(\tau^{(i)}) = \sum_{t=0}^{T} \gamma^t R_{\text{proxy}}(s_t^{(i)}, a_t^{(i)})$. This estimation is unbiased when using trajectories generated by the policy $\pi$. To estimate the empirical variance of the proxy reward, we use:

$$\tilde{\sigma}_{R_{\text{proxy}}}^2 = \tilde{\mathbb{E}}_{\mu_{\pi_{\text{ref}}}}^2[R_{\text{proxy}}] - \tilde{\mathbb{E}}_{\mu_{\pi_{\text{ref}}}}[R_{\text{proxy}}^2] \tag{42}$$

We estimate $\tilde{\mathbb{E}}_{\mu_{\pi_{\text{ref}}}}[R_{\text{proxy}}^2]$ using:

$$\tilde{\mathbb{E}}_{\mu_{\pi_{\text{ref}}}}[R_{\text{proxy}}^2] = (1 - \gamma) \frac{1}{N} \sum_{i=1}^{N} R_{\text{proxy}}^2(\tau^{(i)})$$

where $R_{\text{proxy}}^2(\tau^{(i)}) = \sum_{t=0}^{T} \gamma^t R_{\text{proxy}}^2(s_t^{(i)}, a_t^{(i)})$. However, estimating $\tilde{\mathbb{E}}_{\mu_{\pi_{\text{ref}}}}^2[R_{\text{proxy}}]$ directly from a single batch introduces bias, because the square of an empirical mean is not an unbiased estimator of the square of the true mean. To obtain an unbiased estimate of $\tilde{\mathbb{E}}_{\mu_{\pi_{\text{ref}}}}^2[R_{\text{proxy}}]$, we apply the double-sampling technique (Di Castro et al., 2012; Xie et al., 2018). Specifically, we independently sample another batch of trajectories, denoted $D_{\pi_{\text{ref}}}^*$, from the reference policy $\pi_{\text{ref}}$, and compute:

$$\tilde{\mathbb{E}}_{\mu_{\pi_{\text{ref}}}}^2[R_{\text{proxy}}] = \tilde{\mathbb{E}}_{\mu_{\pi_{\text{ref}}}}^{D_{\pi_{\text{ref}}}}[R_{\text{proxy}}] \times \tilde{\mathbb{E}}_{\mu_{\pi_{\text{ref}}}}^{D_{\pi_{\text{ref}}}^*}[R_{\text{proxy}}]$$

where $\tilde{\mathbb{E}}_{\mu_{\pi_{\text{ref}}}}^{D_{\pi_{\text{ref}}}}[R_{\text{proxy}}]$ and $\tilde{\mathbb{E}}_{\mu_{\pi_{\text{ref}}}}^{D_{\pi_{\text{ref}}}^*}[R_{\text{proxy}}]$ denote the empirical returns computed from the two independent batches using Equation 41. This ensures an unbiased estimation of $\mathbb{E}_{\mu_\pi}^2[R_{\text{proxy}}]$, which is critical for correctly computing the regularization term in the objective 27.

We then normalize the proxy reward for each state-action pair in $D_\pi$ as:

$$R_{\text{proxy}}^{\text{norm}}(s, a) = \frac{R_{\text{proxy}}(s, a) - \tilde{J}(\pi_{\text{ref}}, R_{\text{proxy}})}{\tilde{\sigma}_{R_{\text{proxy}}}} \tag{43}$$

For notational simplicity, we will continue to use $R_{\text{proxy}}$ to denote the normalized proxy reward throughout the remainder of this section.

We use the same batch of sampled trajectories $D_\pi$ from current policy $\pi$ to estimate $\mathbb{E}_{\mu_\pi}[R_{\text{proxy}}]$ using:

$$\tilde{\mathbb{E}}_{\mu_\pi}[R_{\text{proxy}}] = \tilde{J}(\pi, R_{\text{proxy}}) = (1 - \gamma)\frac{1}{N}\sum_{i=1}^{N} R_{\text{proxy}}(\bar{\tau}^{(i)}) \tag{44}$$

where each $\bar{\tau}^{(i)} = (s_0, a_0, s_1, a_1, ..., s_T) \sim D_\pi$ is a trajectory sampled from $\pi$. To estimate $\tilde{\mathbb{E}}_{\mu_\pi}^2[R_{\text{proxy}}]$, we apply the same double-sampling technique. Specifically, we independently sample another batches of trajectories, denoted $D_\pi^*$, from the current policy $\pi$, and compute:

$$\tilde{\mathbb{E}}_{\mu_\pi}^2[R_{\text{proxy}}] = \tilde{\mathbb{E}}_{\mu_\pi}^{D_\pi}[R_{\text{proxy}}] \times \tilde{\mathbb{E}}_{\mu_\pi}^{D_\pi^*}[R_{\text{proxy}}], \tag{45}$$

where $\tilde{\mathbb{E}}_{\mu_\pi}^{D_\pi}[R_{\text{proxy}}]$ and $\tilde{\mathbb{E}}_{\mu_\pi}^{D_\pi^*}[R_{\text{proxy}}]$ denote the empirical returns computed from the two independent batches using Equation 44.

Putting all the steps together, the maxmin algorithm is in Algorithm 2:

---

**Algorithm 2** Max-Min Policy Optimization

---

1: Initialize policy parameters $\theta$
2: Initialize reference policy $\pi_{\text{ref}}$ and collect trajectories $D_{\pi_{\text{ref}}}$
3: Estimate $J(\pi_{\text{ref}}, R_{\text{proxy}})$ using Equation 41 and $\sigma_{R_{\text{proxy}}}^2$ using Equation 42
4: **for** each iteration **do**
5:    Sample trajectories $D_\pi$ from current policy $\pi_\theta$
6:    **if** discrete environment **then**
7:       Estimate occupancy measure using Equation 40
8:    **else**
9:       Train discriminator $d_\phi$ by minimizing Equation 38
10:    **end if**
11:    Estimate $\chi^2$ divergence using Equation 39
12:    Normalize proxy reward for each state-action pair in $D_\pi$ using Equation 43
13:    Estimate proxy reward expectation $\mathbb{E}_{\mu_\pi}[R_{\text{proxy}}]$ using Equation 44
14:    Estimate $\mathbb{E}_{\mu_\pi}^2[R_{\text{proxy}}]$ via double-sampling using Equation 45
15:    Update policy $\pi_\theta$ using PPO to maximize robust objective in Equation 27
16: **end for**

---

## F.3 DERIVATION OF LINEAR MAX-MIN POLICY OPTIMIZATION

As for the Linear Max-Min optimization problem, following the discussion in E.4–E.7, we first need to estimate $Q$:

$$Q = \sum_{(s,a)} C(s, a)\phi(s, a)\phi(s, a)^\top$$

where $C(s, a) = \mu_{\pi_{\text{ref}}}(s, a)$, $\phi(s, a)$ is a vector of known feature functions sampled under the current policy $\pi$. To recover the reference occupancy $\mu_{\text{ref}}$, define

$$\bar{d}_\phi(s, a) = \log \frac{\mu_{\pi_{\text{ref}}}(s, a)}{\mu_\pi(s, a)}$$

we can rewrite $Q$ via importance sampling (Sutton & Barto, 2018):

$$Q = \mathbb{E}_{\mu_\pi}\left[e^{\bar{d}_\phi(s,a)}\phi(s, a)\phi(s, a)^\top\right]$$

Note that $\bar{d}_\phi(s,a)$ differs slightly from $d_\phi(s,a) = \log \frac{\mu_\pi(s,a)}{\mu_{\pi_{\text{ref}}}(s,a)}$ used in the Max-Min optimization algorithm discussed in Appendix F.2, where the numerator and denominator are reversed. To estimate $\bar{d}_\phi(s,a)$, we again train a similar discriminator network as described in Appendix F.1 by minimizing:

$$\phi = \arg\min_\phi \ \mathbb{E}_{\mu_{\pi_{\text{ref}}}}[\log(1 + e^{-d_\phi(s,a)})] + \mathbb{E}_{\mu_\pi}[\log(1 + e^{d_\phi(s,a)})]$$

$$\approx \arg\min_\phi \ \mathbb{E}_{D_{\pi_{\text{ref}}}}[\log(1 + e^{-d_\phi(s,a)})] + \mathbb{E}_{D_\pi}[\log(1 + e^{d_\phi(s,a)})] \tag{46}$$

At optimality, the discriminator satisfies:

$$\bar{d}^*(s,a) = \log \frac{\mu_{\pi_{\text{ref}}}(s,a)}{\mu_\pi(s,a)}$$

And we use $\bar{d}_\phi(s,a) \approx \bar{d}^*(s,a)$. We then estimate $Q$:

$$\tilde{Q} = (1-\gamma)\mathbb{E}_{D_\pi}\left[\sum_{t=0}^\infty \gamma^t e^{\bar{d}_\phi(s_t,a_t)} \phi(s_t,a_t)\phi(s_t,a_t)^\top\right] \tag{47}$$

We then perform feature whitening by applying a linear transformation:

$$\tilde{\phi}(s,a) = \tilde{W}\phi(s,a)$$

where $\tilde{W} = \tilde{Q}^{-1/2}$ is the matrix square root inverse of $\tilde{Q}$.

All subsequent quantities are computed using the transformed features $\tilde{\phi}$. As before, we also normalize the proxy reward for each state-action pair in $D_\pi$ using Equation 43.

After whitening, we need to estimate the gradient of each dual variables $(\lambda_1, \lambda_2, \lambda_3)$ as derived in Appendix E.7.

**Estimating $C_{\phi,j}$ and $D_{\phi,j}$.** Recall that

$$C_{\phi,j} = \sum_{(s,a)} C(s,a)\phi_j(s,a)$$

which can be rewritten via importance sampling as:

$$C_{\phi,j} = \mathbb{E}_{\mu_\pi}\left[e^{\bar{d}_\phi(s,a)}\tilde{\phi}_j(s,a)\right]$$

and then can be approximated via:

$$\tilde{C}_{\phi,j} = (1-\gamma)\mathbb{E}_{D_\pi}\left[\sum_{t=0}^\infty \gamma^t e^{\bar{d}_\phi(s_t,a_t)}\tilde{\phi}_j(s_t,a_t)\right]$$

Similarly, recall that

$$D_{\phi,j} = \sum_{(s,a)} D(s,a)\phi_j(s,a)$$

where $D(s,a) = \mu_{\pi_{\text{ref}}}(s,a) \cdot R_{\text{proxy}}(s,a)$. Using importance sampling, we can write:

$$D_{\phi,j} = \mathbb{E}_{\mu_\pi}\left[e^{\bar{d}_\phi(s,a)}R_{\text{proxy}}(s,a)\tilde{\phi}_j(s,a)\right]$$

and can be approximated in practice by:

$$\tilde{D}_{\phi,j} = (1-\gamma)\mathbb{E}_{D_\pi}\left[\sum_{t=0}^\infty \gamma^t e^{\bar{d}_\phi(s_t,a_t)}R_{\text{proxy}}(s_t,a_t)\tilde{\phi}_j(s_t,a_t)\right]$$

**Estimating $q_j(\lambda_1, \lambda_2)$.** Recall that

$$q_j(\lambda_1, \lambda_2) = \sum_{(s,a)} \left( v(s,a) - \lambda_1 D(s,a) - \lambda_2 C(s,a) \right) \phi_j(s,a)$$

$$= \sum_{(s,a)} v(s,a)\phi_j(s,a) - \lambda_1 \sum_{(s,a)} D(s,a)\phi_j(s,a) - \lambda_2 \sum_{(s,a)} C(s,a)\phi_j(s,a)$$

$$= \mathbb{E}_{\mu_\pi}[\phi_j(s,a)] - \lambda_1 D_{\phi,j} - \lambda_2 C_{\phi,j}$$

where $v(s,a) = \mu_\pi(s,a)$ and $\mathbb{E}_{\mu_\pi}[\phi_j(s,a)]$ is the discounted feature expectation under the policy $\pi$. We can estimate the first term using:

$$\tilde{\mathbb{E}}_{\mu_\pi}[\phi_j(s,a)] = (1-\gamma)\frac{1}{N}\sum_{i=1}^{N}\tilde{\phi}_j(\bar{\tau}^i)$$

where each $\bar{\tau}^{(i)} = (s_0, a_0, s_1, a_1, ..., s_T) \sim D_\pi$ is a trajectory sampled from $\pi$, and $\tilde{\phi}_j(\bar{\tau}^{(i)}) = \sum_{t=0}^{T} \gamma^t \tilde{\phi}_j(s_t^{(i)}, a_t^{(i)})$ . Given the above estimates, we can finally compute:

$$\tilde{q}_j(\lambda_1, \lambda_2) = \tilde{\mathbb{E}}_{\mu_\pi}[\phi_j(s,a)] - \lambda_1 \tilde{D}_{\phi,j} - \lambda_2 \tilde{C}_{\phi,j}$$

With the above estimation, we can compute the gradient and solve for the optimal dual variables $\lambda = (\lambda_1, \lambda_2, \lambda_3)$ using the Levenberg-Marquardt algorithm (Moré, 1978), a damped least-squares method designed for solving nonlinear systems of equations. Specifically, we use the root solver in SciPy (Virtanen et al., 2020) to find the stationary point of the gradient $\nabla_\lambda g(\lambda) = 0$. We initialize the optimization with $\lambda_1 = 0$, $\lambda_2 = 0$, and $\lambda_3 = -1$, and enforce $\lambda_3 < 0$ throughout training to ensure concavity of the dual objective $g(\lambda)$. To enforce the non-negativity constraint $\boldsymbol{\theta} \geq 0$ as required by the analytical form in Equation 33, we manually clip each $\theta_i$ to ensure it remains non-negative. Future work may explore alternative solvers better suited to constrained optimization.

Recall that the optimal primal variable $\theta_j^*$ is:

$$\theta_j^*(\lambda) = \max\left(0, \frac{q_j(\lambda_1, \lambda_2)}{2\lambda_3}\right)$$

After optimizing for the dual variables, we can substitute the optimal $(\lambda_1^*, \lambda_2^*, \lambda_3^*)$ back into the above equation and get:

$$\tilde{\theta}_j^*(\lambda) = \max\left(0, \frac{\tilde{q}_j(\lambda_1^*, \lambda_2^*)}{2\lambda_3^*}\right)$$

for all features. Then we can substitute the optimal $\tilde{\boldsymbol{\theta}}^*$ back in the robust reward objective in Equation 28 and train the policy $\pi$ to maximize the outer problem using the standard reinforcement learning algorithm proximal policy optimization (PPO) (Schulman et al., 2017).

Putting all the steps together, the linear maxmin algorithm is in Algorithm 3:

### F.4 ENVIRONMENT DESCRIPTION AND REWARD HACKING TYPES

**Traffic.** This environment simulates a highway merging scenario, adapted from (Pan et al., 2022; Wu et al., 2021; Vinitsky et al., 2018), where a group of autonomous vehicles (AVs) controlled by an RL agent must merge into human-driven traffic. Each AV observes its own state (position and velocity) and those of nearby vehicles, and outputs continuous acceleration actions. The true reward is designed to ensure smooth and efficient traffic flow, encouraging low commute times and gentle accelerations. The reference policy $\pi_{\text{ref}}$ is a behavioral cloning (BC) policy trained on demonstrations generated by the Intelligent Driver Model (IDM) (Treiber et al., 2000).

**Pandemic.** Based on the PandemicSimulator (Kompella et al., 2020), this environment models infection dynamics using an extended SEIR model. At each timestep, the agent selects a lockdown policy to control the spread of disease while minimizing societal costs. The true reward balances infection severity, political disruption, and policy smoothness over time. The reference policy is trained via behavioral cloning on a mixture of realistic and hand-crafted policy trajectories.

---

**Algorithm 3** Linear Max-Min Policy Optimization

---

1: Initialize policy parameters $\theta$
2: Initialize reference policy $\pi_{\text{ref}}$ and collect trajectories $D_{\pi_{\text{ref}}}$
3: Estimate $J(\pi_{\text{ref}}, R_{\text{proxy}})$ using Equation 41 and $\sigma^2_{R_{\text{proxy}}}$ using Equation 42
4: **for** each iteration **do**
5:    Sample trajectories $D_\pi$ from current policy $\pi_\theta$
6:    **if** discrete environment **then**
7:       Estimate occupancy measure using Equation 40
8:    **else**
9:       Train discriminator $d_\phi$ by minimizing Equation 46
10:    **end if**
11:    Normalize proxy reward for each state-action pair in $D_\pi$ using Equation 43
12:    Estimate $\tilde{Q}$ using Equation 47
13:    Compute feature transformation $\tilde{W} = \tilde{Q}^{-1/2}$ and transform features $\tilde{\phi}(s,a) = \tilde{W}\phi(s,a)$
14:    Estimate $\tilde{C}_{\phi,j}, \tilde{D}_{\phi,j}$ using transformed features
15:    Compute $\tilde{q}_j(\lambda_1, \lambda_2)$ for all features
16:    Solve for optimal dual variables $(\lambda_1, \lambda_2, \lambda_3)$ by maximizing the dual objective (Equation 34)

17:    Compute the optimal primal variable $\tilde{\theta}_j^*$ for all features
18:    Update policy $\pi_\theta$ using PPO to maximize the robust objective in Equation 28
19: **end for**

---

**Glucose Monitoring.** This environment uses the SimGlucose simulator (Man et al., 2014; Fox et al., 2020), where an RL agent administers insulin doses to a simulated patient with Type 1 Diabetes. The goal is to maintain safe blood glucose levels and minimize long-term health risk. The reference policy is trained via behavioral cloning using data generated by a PID controller with clinically tuned parameters (Steil, 2013). Proxy rewards in this setting often reflect surrogate objectives such as treatment cost or patient burden.

**Tomato Watering GridWorld.** This environment presents a simple spatial grid where the agent waters tomato plants. The true reward corresponds to the number of tomatoes correctly watered. However, the proxy reward includes an artificially high bonus at a specific grid location (a "sprinkler state"), which causes the agent to overfit by remaining in that region despite little actual benefit to overall tomato growth. The reference policy follows (Laidlaw et al., 2025), with 10% random actions added to allow for policy improvement.

**RLHF.** This environment builds on prior work (Laidlaw et al., 2025; Coste et al.) that studies overoptimization of LLM-based reward models. The proxy reward function is derived from a reward model fine-tuned on the AlpacaFarm preference dataset (Dubois et al., 2023), using the Pythia-70M model (Biderman et al., 2023), a comparatively small model. For the true reward signal, we adopt the Llama 3 Tulu V2 8B reward model released by AI2 (Ivison et al., 2024). The reference policy corresponds to the supervised fine-tuned (SFT) model from (Laidlaw et al., 2025; Coste et al.), which was trained on the AlpacaFarm SFT dataset using Pythia-1.4B.

**Types of Reward Hacking.** We adopt the taxonomy proposed in (Pan et al., 2022) to classify the kinds of proxy reward misalignments that lead to reward hacking. Our selected environments span all three major categories:

- **Misweighting:** The proxy reward includes all relevant objectives but uses incorrect relative weights. Our Linear Max-Min method specifically seeks the most adversarial weighting in this space.

- **Ontological:** The proxy captures the correct high-level goal using different or incomplete features. In the **Traffic** environment, the true reward combines commute time, acceleration, and headway, whereas the proxy replaces commute time with velocity. In the **Pandemic** environment, the true reward penalizes infections, political cost, lower stage changes, and non-smooth policies, while the proxy omits the political cost entirely. Similarly, in **Glu-**

**cose**, the proxy reward only considers the expected patient costs while the true reward only measures the health risk.

- **Scope:** The proxy evaluates behavior over a limited domain. In the **Tomato** environment, the true reward reflects the number of tomatoes successfully watered. However, the proxy introduces a large bonus at a specific state (the sprinkler), incentivizing the agent to pursue this location at the expense of fulfilling the intended watering task. In the **RLHF** environment, the proxy reward is produced by a comparatively small model with limited evaluative capacity, whereas the true reward is derived from a much larger, stronger model. Consequently, the proxy reward provides a less reliable evaluation signal.

### F.5  ADDITIONAL EXPERIMENT SETUP

**Non-LLM Experiments.** For the policy networks, we follow the architectures described in (Laidlaw et al., 2025). In the Pandemic, Traffic, and Tomato environments, we use fully connected neural networks with 2 layers of 128 units, 4 layers of 512 units, and 4 layers of 512 units, respectively. For the Glucose environment, we employ a three-layer LSTM network, where each LSTM layer has 64 units. We use the pre-trained policies provided in the ORPO repository[2] as the reference policies $\pi_{\text{ref}}$. We initialize the policy network with the corresponding pre-trained checkpoint for the Traffic, Glucose, and Pandemic environments, and initialize a random policy for the Tomato environment. Table 3 summarizes the hyperparameters used for PPO training across all models and environments.

Table 3: Hyperparameters used for PPO training across different environments.

| Hyperparameter | Traffic | Glucose | Pandemic | Tomato |
|---|---|---|---|---|
| Training iterations | 250 | 500 | 260 | 500 |
| Batch size | 40000 | 100000 | 3860 | 3000 |
| Optimizer | Adam | Adam | Adam | Adam |
| Learning rate | $5 \times 10^{-5}$ | $1 \times 10^{-5}$ | 0.0003 | $1 \times 10^{-3}$ |
| Gradient clipping | N/A | 10 | 10 | 0.1 |
| Discount factor ($\gamma$) | 0.99 | 0.99 | 0.99 | 0.99 |
| Random seed | 0 | 0 | 0 | 0 |
| GAE coefficient ($\lambda$) | 0.97 | 0.98 | 0.95 | 0.98 |
| Entropy coefficient (start) | 0.01 | 0.01 | 0.1 | 0.01 |
| Entropy coefficient (end) | 0.01 | 0.01 | 0.01 | 0.01 |
| KL target | 0.02 | $1 \times 10^{-3}$ | 0.01 | $1 \times 10^{-3}$ |
| Value function loss clipping | 10000 | 100 | 20 | 10 |
| Value function loss coefficient | 0.5 | 0.0001 | 0.5 | 0.1 |
| Share value function layers | True | True | True | False |

As for the reward used during training and evaluation, we follow the same setup as ORPO (Laidlaw et al., 2025). All policies are trained using **only the proxy reward**, while both the true and proxy rewards are used for evaluation.

In the **Traffic** environment, the proxy reward is a weighted combination of *velocity*, *acceleration*, and *headway*, with weights 1, 1, and 0.1, respectively. The true reward, on the other hand, uses *commute time*, *acceleration*, and *headway*, also weighted 1, 1, and 0.1.

In the **Pandemic** environment, the proxy reward is composed of *infection summary absolute*, *lower stage*, and *smooth stage changes*, with weights 10, 0.1, and 0.01. The true reward adds a *political* component to these three features and is weighted with 10.

For the **Glucose** environment, the proxy reward includes only one feature: *expected patient cost*. The true reward is based on *magni_bg*, which measures the health risk of the patient.

---

[2]`https://github.com/cassidylaidlaw/orpo/tree/main/data/base_policy_checkpoints`

In the **Tomato** environment, the true reward counts the number of *watered tomato*. The proxy reward adds a large bonus at a specific state (*sprinkler*), incentivizing the agent to reach that location regardless of its impact on the primary task.

**LLM Experiments.** We adopt the common formulation for RLHF as a *contextual bandit* problem, where the environment is modeled as a Markov Decision Process (MDP) with a discount factor $\gamma = 0$. In this setup, the return of a policy $\pi$ under a given reward function $R$ simplifies to:

$$J(\pi, R) = \mathbb{E}_{s \sim \mu_0, a \sim \pi}[R(s, a)],$$

where $\mu_0$ denotes the distribution over initial states. In the context of RLHF, each state corresponds to a prompt sampled from a dataset, and the action is the model's generated response. The reward is then computed based on this prompt–response pair.

Under this contextual bandit assumption, the $\chi^2$ divergence between occupancy measures reduces to the divergence between action distributions conditioned on prompts:

$$\chi^2(\mu_\pi \| \mu_{\pi_{\text{ref}}}) = \mathbb{E}_{s \sim \mu_0} \left[ \chi^2(\pi(\cdot|s) \| \pi_{\text{ref}}(\cdot|s)) \right],$$

as established in Lemma A.6 of (Laidlaw et al., 2025). This allows us to avoid discriminator-based estimation of occupancy ratios in this setting. Instead, we directly estimate the $\chi^2$ divergence using the following estimator:

$$\tilde{\chi}^2(\mu_\pi \| \mu_{\pi_{\text{ref}}}) = \mathbb{E}_{s \sim \mu_0, a \sim \pi} \left[ \frac{\pi(a|s)}{\pi_{\text{ref}}(a|s)} + \frac{\pi_{\text{ref}}(a|s)}{\pi(a|s)} - 2 \right],$$

as proposed in (Laidlaw et al., 2025).

For policy optimization, we apply the same Max-Min training algorithm described in Appendix F.2, adapting it to the contextual bandit structure without discriminator training. We evaluate our approach in the RLHF setting using a setup consistent with prior work (Laidlaw et al., 2025; Coste et al.). The proxy reward function is derived from a reward model fine-tuned on the AlpacaFarm preference dataset (Dubois et al., 2023), using the Pythia-70M model (Biderman et al., 2023). For the true reward signal, we adopt the Llama 3 Tulu V2 8B reward model released by AI2 (Ivison et al., 2024). The reference policy corresponds to the supervised fine-tuned (SFT) model from (Laidlaw et al., 2025; Coste et al.), which was trained on the AlpacaFarm SFT dataset using Pythia-1.4B. All policy evaluations, both proxy and true rewards, are conducted on the same set of prompts used in (Laidlaw et al., 2025).

To further strengthen our experimental evaluation regarding the RLHF setting, we also compare against the Reward Ensemble method (referred to as Ensemble for brevity) (Eisenstein et al., 2023). Specifically, we adopt their *finetune ensembles* setting: we fine-tune five reward models on the AlpacaFarm preference dataset, all initialized from the same pre-trained Pythia-70M model but using five different random seeds, and aggregate their outputs using the mean rule. This setup is directly comparable to our RLHF configuration for ORPO and our methods, where both ORPO and our approach use a single fine-tuned Pythia-70M reward model.

**Selection of $r$.** For our `Max-Min` and `Linear Max-Min` policy optimization algorithms, the correlation parameter $r$ serves as an additional hyperparameter. In practice, as with `ORPO` (Laidlaw et al., 2025), $r$ may only be approximately estimated, and there is currently no principled method for selecting its optimal value. To address this, we perform a grid search over $r \in \{0.1, 0.2, \ldots, 0.9\}$ for each environment and measure the resulting `Max-Min` and `Linear Max-Min` policy expected returns under the worst-case or linear worst-case reward. Results on all searched $r$ can be found in Appendix H.5. Additional analysis of how different training values of $r$ affect robustness under varying evaluation $r$ values is provided in Appendix H.2. Unless otherwise noted, we use the following $r$ values for training and evaluation throughout our experiments: $r = 0.3$ for **Traffic**, $r = 0.7$ for **Pandemic**, $r = 0.9$ for **Glucose**, $r = 0.4$ for **Tomato**, and $r = 0.4$ for RLHF. As for `ORPO` policy, we trained with occupancy-measure $\chi^2$ regularization, using the official implementation from (Laidlaw et al., 2025). All hyperparameters are set as recommended to ensure optimal performance. The `ORPO*` shares the exact same setting as the `ORPO` policy with the full discriminator training schedule as in our algorithms.

**Evaluation of the worst-case performance.** Theoretically, in the absence of structural constraints on the reward function, as opposed to the case of linear rewards, the worst-case reward of a policy

in state-action pairs unvisited by $\pi_{\text{ref}}$ can be arbitrarily negative without violating the correlation constraint. However, assigning extremely negative values is impractical in real-world scenarios due to domain constraints. Moreover, doing so would render all policies with at least one unseen state-action pair equally poor in terms of worst-case reward, obscuring meaningful comparisons. To address this, we define a minimal feasible reward value $R_{\text{min}}$ and assign it to all unseen state-action pairs. The actual expected worst-case reward (**Worst***) is thus calculated as:

$$\sum_{(s,a):\mu_{\pi_{\text{ref}}}(s,a)>0} \mu_\pi(s,a)R_{\text{worst}} \ + \sum_{(s',a'):\mu_{\pi_{\text{ref}}}(s',a')=0} \mu_\pi(s',a')R_{\text{min}}$$

where the first part is derived from the adversarial reward function given by our inner minimization solution, and the second part applies to state-action pairs unvisited by $\pi_{\text{ref}}$.

In practice, however, environments like Traffic, Pandemic, and Glucose are continuous with large state-action spaces, making it difficult to reliably estimate $\mu_\pi(s,a)$ and $\mu_{\pi_{\text{ref}}}(s,a)$ from a limited number of trajectories. As a result, identifying unvisited or low-density regions in these environments is far more ambiguous. Therefore, for these continuous environments, we rely on the output of the discriminator as a signal for detecting unseen state-action pairs. Specifically, if the discriminator outputs infinity (or diverges numerically) for a given state-action, we treat this as an indication that the state-action was never visited by the reference policy $\pi_{\text{ref}}$. We approximate the total occupancy (**Occ**) over such state-action pairs by computing their frequency in the sampled trajectories, and use the expected worst-case reward (**Worst**) of a policy $\pi$ over the remaining state-action pairs as the default worst-case performance metric: $\sum_{(s,a):d_\phi(s,a)<\infty} \mu_\pi(s,a)R_{\text{worst}}(s,a)$. In contrast, for the discrete Tomato environment, we directly estimate the occupancy measure by sampling state-action pairs and then compute **Worst*** accordingly. Further details on this procedure are provided in Appendix H.2.

To compare the worst-case performance of different policies, we sample 200 trajectories in the **Traffic** and **Glucose** environments, 20 trajectories in **Pandemic**, 1000 trajectories in **Tomato**, and 8 answers per prompt in **RLHF** to estimate the worst-case performance of a policy.

**Evaluation of policy robustness.** To evaluate robustness across different correlation levels, we uniformly sample candidate vectors $\boldsymbol{\theta}$, where each component $\theta_i$ is drawn from the interval $[0,1]$. We use the same number of trajectories sampled from the reference policy $\pi_{\text{ref}}$ to determine whether it satisfies the correlation constraint:

$$\mathbb{E}_{\mu_{\pi_{\text{ref}}}}\left[\frac{\boldsymbol{\theta}^\top\boldsymbol{\phi}-M}{V}\cdot R_{\text{proxy}}\right]=r,$$

where $M$ and $V$ denote the mean and standard deviation of $\boldsymbol{\theta}^\top\boldsymbol{\phi}$ under the reference policy.

**Note:** For our worst-case performance evaluation, we explicitly normalize the reward to have zero mean and unit variance under the reference policy (enforcing $M=0$ and $V=1$). In contrast, for the robustness evaluation across correlation levels, we do not apply such normalization. This allows us to report the average reward under each $\boldsymbol{\theta}$ in its original scale, reflecting variability more comparable to the original true reward landscape.

### F.6 Training Time and Complexity

Table 4 reports the total training time for each algorithm across different environments. All experiments were conducted on a single NVIDIA RTX 4090 GPU (24GB memory) and a 13th Gen Intel Core i9-13900KF CPU (32 threads). We implemented all methods in Python 3.9 using PyTorch 2.6.0 (Paszke, 2019), RLlib (Liang et al., 2018) and trlX (Havrilla et al., 2023).

The training times and memory usages across different environments are summarized in Table 4 and Table 5. Since the training durations for `ORPO*`, `Max-Min`, and `Linear Max-Min` differ by less than one hour in each setting, and the memory usages differ by less than 10MB, we group them together for brevity. Since the RLHF environment does not require training a discriminator, the training times and memory usages for `ORPO` and our `Max-Min` are identical. We therefore exclude RLHF from the runtime analysis. As shown in Table 4, all three methods require more training time compared to the original `ORPO` implementation. The increased training time primarily results from additional gradient steps used to more thoroughly train the discriminator network. Specifically, the

per-iteration training time is approximately 2.5 minutes for Traffic, 4.6 minutes for Pandemic, and 8.9 minutes for Glucose. This leads to a total training time increase from roughly 7 hours to 37 hours in Glucose. However, the added cost is environment-dependent and remains moderate in simpler settings, for example, increasing from 5 to 10 hours in Traffic. In contrast, the memory footprint of our methods is very close to that of `ORPO`: the peak CPU memory usage differs by less than 30–50MB across environments (within a few percent of `ORPO` in all cases). Overall, these results indicate that our methods introduce a modest runtime overhead and negligible memory overhead, achieving a practical trade-off between computational cost and the improved quality of divergence estimation.

Table 4: Approximate training time for each algorithm across different environments.

| Algorithm | Traffic | Glucose | Pandemic | Tomato |
|---|---|---|---|---|
| `ORPO` | $\approx$5h | $\approx$7h | $\approx$14h | $\approx$1h |
| `ORPO*` / `Max-Min` / `Linear Max-Min` | $\approx$10h | $\approx$37h | $\approx$19h | $\approx$1h |

Table 5: Approximate memory usage for each algorithm across different environments.

| Algorithm | Traffic | Glucose | Pandemic | Tomato |
|---|---|---|---|---|
| `ORPO` | $\approx$1679MB | $\approx$1662MB | $\approx$1813MB | $\approx$2148MB |
| `ORPO*` / `Max-Min` / `Linear Max-Min` | $\approx$1706MB | $\approx$1674MB | $\approx$1864MB | $\approx$1903MB |

**Complexity.** At first glance, regularization-based approaches like `ORPO` may appear more computationally efficient than max-min optimization, which often involves iterative procedures to solve both inner and outer objectives. However, in practice, `ORPO` requires repeatedly estimating the $\chi^2$ divergence between policy distributions during each policy update step. This estimation is done by training a discriminator network, which itself involves multiple optimization steps per iteration. In contrast, our `Max-Min` formulation admits a closed-form solution for the inner minimization over reward functions. This allows us to avoid iterative solving in the inner loop entirely. For the `Linear Max-Min` variant, although a closed-form expression for the dual variables is not available, the corresponding dual optimization problem is smooth and well-posed, and can be solved efficiently using standard gradient-based methods. Therefore, despite the max-min structure, our method does not incur higher practical complexity compared to `ORPO`. In fact, both approaches rely on discriminator-based divergence estimation and perform comparable amounts of computation per iteration. The main difference lies in the structure of the objective, not in the asymptotic or empirical complexity. In summary, `ORPO` does not inherently enjoy a complexity advantage over our `Max-Min` or `Linear Max-Min` algorithms.

## G  CONVERGENCE ANALYSIS

In this section, we study the convergence of our Max-Min and Linear Max-Min algorithms. As both methods rely on accurately estimating the occupancy measure, we begin by analyzing the sample complexity of this estimation via the discriminator described in Appendix F.1.

### G.1  SAMPLE COMPLEXITY OF OCCUPANCY MEASURE ESTIMATION

In this section, we analyze the sample complexity of estimating the occupancy measure via the discriminator described in Appendix F.1. Our argument adapts techniques from Huang et al. (2023); Barakat et al. (2024) to our setting. To start with, we define the following notations for convenience. Let $x = (s, a)$ range over the space $\mathcal{X}$ of all possible state-action pairs. We consider two reference distributions on $\mathcal{X}$: $\mu_{\pi_{\mathrm{ref}}}$ and $\mu_\pi$. The (true) density ratio of $\mu_\pi$ with respect to $\mu_{\pi_{\mathrm{ref}}}$ is

$$L^\star(x) := \frac{\mu_\pi(x)}{\mu_{\pi_{\mathrm{ref}}}(x)} \qquad \text{on the support of } \mu_{\pi_{\mathrm{ref}}},$$

and its log-ratio is $d^\star(x) := \log L^\star(x)$.

We work with a parametric log-ratio class $\mathcal{D} = \{d_\phi : \mathcal{X} \to \mathbb{R}\}$ and the induced ratio class $\mathcal{L} = \{L_\phi := e^{d_\phi}\}$. Following Equation 38 in Appendix F.1, we learn $d_\phi$ by minimizing the following loss:

$$\mathcal{R}(d) := \tfrac{1}{2} \mathbb{E}_{x \sim \mu_{\pi_{\text{ref}}}}\big[ \log(1 + e^{d(x)}) \big] + \tfrac{1}{2} \mathbb{E}_{x \sim \mu_\pi}\big[ \log(1 + e^{-d(x)}) \big]. \tag{48}$$

Given $n_{\text{ref}}$ i.i.d. samples $\{x_i^{\text{ref}}\}_{i=1}^{n_{\text{ref}}} \sim \mu_{\pi_{\text{ref}}}$ and $n_\pi$ i.i.d. samples $\{x_j^\pi\}_{j=1}^{n_\pi} \sim \mu_\pi$, which are independent, we can minimize the empirical loss as follows in practice:

$$\widehat{\mathcal{R}}(d) := \tfrac{1}{2} \cdot \frac{1}{n_{\text{ref}}} \sum_{i=1}^{n_{\text{ref}}} \log\big(1 + e^{d(x_i^{\text{ref}})}\big) + \tfrac{1}{2} \cdot \frac{1}{n_\pi} \sum_{j=1}^{n_\pi} \log\big(1 + e^{-d(x_j^\pi)}\big). \tag{49}$$

Let the true loss minizer be

$$d^\star \in \arg\min_{d \in \mathcal{D}} \mathcal{R}(d), \qquad L^\star := e^{d^\star}.$$

Let the empirical loss minimizer be

$$\widehat{d} \in \arg\min_{d \in \mathcal{D}} \widehat{\mathcal{R}}(d), \qquad \widehat{L} := e^{\widehat{d}}.$$

For convenience, we also define the mixture distribution $\mu_{\text{mix}} := \tfrac{1}{2} \mu_{\pi_{\text{ref}}} + \tfrac{1}{2} \mu_\pi$, which will be used later. We make the following assumption throughout the analysis.

**Assumption 1** (Modeling, boundedness and cover). *The following conditions hold throughout the analysis:*

1. *Realizability. The true log-ratio belongs to the model class: $d^\star \in \mathcal{D}$ (equivalently, $L^\star \in \mathcal{L}$).*

2. *Bounded. There exist constants $0 < \alpha \le \beta < \infty$ such that*

$$\alpha \le L_\phi(x) \le \beta \quad \text{for all } x \in \mathcal{X}, \ \phi,$$

   *and hence $\alpha \le L^\star(x) \le \beta$ as well. Equivalently, $d_\phi(x) \in [\log \alpha, \log \beta]$.*

3. *$L_1$ optimistic cover (Definition 3 of (Huang et al., 2023)). There exists a finite set $\overline{\mathcal{L}} \subset (0, \infty)^{\mathcal{X}}$ with cardinality $|\overline{\mathcal{L}}| = M$ and a scale $\gamma > 0$ such that for every $L \in \mathcal{L}$ there is $\overline{L} \in \overline{\mathcal{L}}$ with*

$$\overline{L}(x) \ge L(x) \text{ for all } x, \qquad \mathbb{E}_{x \sim \mu_{\text{ref}}}\big[ |\overline{L}(x) - L(x)| \big] \le \gamma, \qquad \alpha \le \overline{L}(x) \le \beta.$$

   *We denote $\overline{\mathcal{D}} := \{\overline{d} := \log \overline{L} : \overline{L} \in \overline{\mathcal{L}}\}$.*

Assumption 1 collects the conditions used throughout our analysis. First, *realizability* is standard in likelihood-based occupancy estimation and allows us to control the estimation error through the complexity of the parametric class rather than the size of $\mathcal{X}$. In practice, a sufficiently expressive neural discriminator makes this assumption reasonable. Second, *boundedness* guarantees well-posedness on the support of $\mu_{\pi_{\text{ref}}}$ and prevents divisions by zero. It can be enforced by restricting attention to the support of $\mu_{\pi_{\text{ref}}}$ or by applying ratio clipping during training. Finally, the $L_1$ *optimistic cover* (adopted from Definition 3 of Huang et al. (2023)) is the technical device that enables uniform concentration and converts control of the loss in Equation 48 into an $L_1$ error with clean constants. We instantiate this cover for our discriminator class later in the proof.

We begin by stating some auxiliary lemmas that formalize the structural claims used later.

**Lemma 1** (Strong convexity of $\mathcal{R}(d)$). *Let Assumption 1 hold true. Define*

$$\lambda := \frac{\min\{\alpha, \beta\}}{(1 + \max\{\alpha, \beta\})^2} > 0.$$

*Then for any measurable $d \in \mathcal{D}$ and for the unique minimizer $d^\star$ to Equation 48, we have*

$$\mathcal{R}(d) - \mathcal{R}(d^\star) \ge \frac{\lambda}{2} \mathbb{E}_{x \sim \mu_{\text{mix}}}\big[ (d(x) - d^\star(x))^2 \big].$$

*Proof.* Fix $x$ and define the pointwise loss as:

$$r_x(d) := (1 - \eta(x)) \log(1 + e^d) + \eta(x) \log(1 + e^{-d}), \quad \eta(x) := \frac{\mu_\pi(x)}{\mu_\pi(x) + \mu_{\pi_{\mathrm{ref}}}(x)}.$$

Its derivatives w.r.t. the scalar $d$ are $r'_x(d) = \sigma(d) - \eta(x)$ and $r''_x(d) = \sigma(d)(1 - \sigma(d)) > 0$, where $\sigma(d) = \frac{e^d}{1+e^d}$. We notice that $r''_x(d)$ is independent of $\eta(x)$. Therefore, at every $x$, the pointwise loss $r_x$ is strictly convex in $d$.

Now we estimate the lower bound for $r''_x(d)$. On the range $d \in [\log \alpha, \log \beta]$ (boundedness from Assumption 1), let $y = e^d \in [\alpha, \beta]$; then

$$r''_x(d) = \sigma(d)(1 - \sigma(d)) = \frac{y}{(1+y)^2} =: f(y)$$

Let's consider the monotonicity of $f(y)$, $f'(y) = \frac{1-y}{(1-y)^3}$, so $f$ increases on $(0, 1]$ and decreases on $[1, \infty)$. Therefore,

$$\min_{y \in [\alpha, \beta]} f(y) = \min\left\{ \frac{\alpha}{(1+\alpha)^2}, \frac{\beta}{(1+\beta)^2} \right\}.$$

We can consider a slightly more conservative but simpler bound:

$$\lambda := \frac{\min\{\alpha, \beta\}}{(1 + \max\{\alpha, \beta\})^2} \leq \min_{t \in [\alpha, \beta]} f(y),$$

Thus, for all $x$ and all $d \in [\log \alpha, \log \beta]$, we have $r''_x(d) \geq \lambda$. Notice that strong convexity (with parameter $\lambda$) of a $C^2$ univariate function $g$ satisfies:

$$g(u) \geq g(v) + g'(v)(u - v) + \frac{\lambda}{2}(u - v)^2 \qquad \text{for all } u, v.$$

Applying this with $g(\cdot) = r_x(\cdot)$, $u = d(x)$, and $v = d^\star(x)$, where $d^\star$ is the unique pointwise minimizer (so $r'_x(d^\star(x)) = 0$). We get for every $x$,

$$r_x(d(x)) - r_x(d^\star(x)) \geq \frac{\lambda}{2}(d(x) - d^\star(x))^2.$$

Recall that $\mu_{\mathrm{mix}} = \frac{1}{2}\mu_{\pi_{\mathrm{ref}}} + \frac{1}{2}\mu_\pi$ and

$$\mathcal{R}(d) = \frac{1}{2} \mathbb{E}_{x \sim \mu_{\pi_{\mathrm{ref}}}}[\log(1 + e^{d(x)})] + \frac{1}{2}\mathbb{E}_{x \sim \mu_\pi}[\log(1 + e^{-d(x)})] = \mathbb{E}_{x \sim \mu_{\mathrm{mix}}}[r_x(d)]$$

Taking expectation with respect to the mixture $\mu_{\mathrm{mix}}$ gives

$$\mathcal{R}(d) - \mathcal{R}(d^\star) = \mathbb{E}_{x \sim \mu_{\mathrm{mix}}}[r_x(d(x)) - r_x(d^\star(x))] \geq \frac{\lambda}{2}\mathbb{E}_{x \sim \mu_{\mathrm{mix}}}[(d(x) - d^\star(x))^2],$$

which is the desired inequality. $\qquad \square$

We now establish three Lipschitz bounds that will be used repeatedly in the analysis.

**Lemma 2** (Lipschitz bounds). *Let $L_+ := \frac{\beta}{1+\beta}$ and $L_- := \frac{1}{1+\alpha}$. For all $d, \tilde{d} \in [\log \alpha, \log \beta]$, the following hold:*

1. $\left| \log(1 + e^d) - \log(1 + e^{\tilde{d}}) \right| \leq L_+ |d - \tilde{d}|.$

2. $\left| \log(1 + e^{-d}) - \log(1 + e^{-\tilde{d}}) \right| \leq L_- |d - \tilde{d}|.$

3. $\left| e^d - e^{\tilde{d}} \right| \leq \beta |d - \tilde{d}|.$

*Proof.* (1) Define $f_+(u) = \log(1 + e^u)$. Then $f'_+(u) = \sigma(u) = \frac{e^u}{1+e^u}$. On $u \in [\log \alpha, \log \beta]$ we have $e^u \in [\alpha, \beta]$, hence

$$|f'_+(u)| = \frac{e^u}{1 + e^u} \leq \sup_{y \in [\alpha, \beta]} \frac{y}{1 + y} = \frac{\beta}{1 + \beta} = L_+.$$

By the mean value theorem, $\left| f_+(d) - f_+(\tilde{d}) \right| \le L_+ \left| d - \tilde{d} \right|$.

(2) Define $f_-(u) = \log(1 + e^{-u})$. Then $f'_-(u) = -\sigma(-u) = -\frac{1}{1+e^u}$. For $u \in [\log \alpha, \log \beta]$,

$$|f'_-(u)| = \frac{1}{1 + e^u} \le \sup_{y \in [\alpha, \beta]} \frac{1}{1 + y} = \frac{1}{1 + \alpha} = L_-.$$

Again by the mean value theorem, $\left| f_-(d) - f_-(\tilde{d}) \right| \le L_- \left| d - \tilde{d} \right|$.

(3) For $g(u) = e^u$ we have $g'(u) = e^u$. On $[\log \alpha, \log \beta]$, $e^u \le \beta$, hence $|g'(u)| \le \beta$. The mean value theorem yields $\left| e^d - e^{\tilde{d}} \right| \le \beta \left| d - \tilde{d} \right|$. $\qquad \square$

**Lemma 3** (Uniform deviation over the finite cover). *Let Assumption 1 hold true. Define*

$$B := \tfrac{1}{2} \log(1 + \beta) + \tfrac{1}{2} \log\left(1 + \tfrac{1}{\alpha}\right).$$

*Let $\overline{\mathcal{D}}$ be a finite cover ($L_1$ optimistic cover from Assumption 1) with cardinality $|\overline{\mathcal{D}}| = M$. Define $n := \min\{n_{\mathrm{ref}}, n_\pi\}$ and $\eta := \sqrt{\frac{\log(M/\delta)}{n}}$ for any $\delta \in (0, 1)$. Then, with probability at least $1 - \delta$,*

$$\sup_{\overline{d} \in \overline{\mathcal{D}}} \left| \widehat{\mathcal{R}}(\overline{d}) - \mathcal{R}(\overline{d}) \right| \le 2 B \eta. \tag{50}$$

*Proof.* Fix $\overline{d} \in \overline{\mathcal{D}}$. Define

$$\Delta_{\mathrm{ref}}(\overline{d}) := \sum_{i=1}^{n_{\mathrm{ref}}} \left[ \log(1 + e^{\overline{d}(x)}) \right] - \mathbb{E}_{\mu_{\pi_{\mathrm{ref}}}} \left[ \log(1 + e^{\overline{d}(x)}) \right],$$

$$\Delta_\pi(\overline{d}) := \sum_{i=1}^{n_\pi} \left[ \log(1 + e^{-\overline{d}(x)}) \right] - \mathbb{E}_{\mu_\pi} \left[ \log(1 + e^{-\overline{d}(x)}) \right].$$

Then

$$\widehat{\mathcal{R}}(\overline{d}) - \mathcal{R}(\overline{d}) = \tfrac{1}{2} \Delta_{\mathrm{ref}}(\overline{d}) + \tfrac{1}{2} \Delta_\pi(\overline{d}), \quad \Rightarrow \quad \left| \widehat{\mathcal{R}}(\overline{d}) - \mathcal{R}(\overline{d}) \right| \le \tfrac{1}{2} \left| \Delta_{\mathrm{ref}}(\overline{d}) \right| + \tfrac{1}{2} \left| \Delta_\pi(\overline{d}) \right|.$$

By the boundedness of Assumption 1, we have $d(x) \in [\log \alpha, \log \beta]$, each summand satisfies

$$0 \le \log(1 + e^{\overline{d}(x)}) \le \log(1 + \beta), \qquad 0 \le \log(1 + e^{-\overline{d}(x)}) \le \log\left(1 + \tfrac{1}{\alpha}\right).$$

Hence, by the Hoeffding's inequality, for any $t > 0$,

$$\mathbb{P}\left( \left| \Delta_{\mathrm{ref}}(\overline{d}) \right| \ge t \right) \le 2 \exp\left( -\frac{2 n_{\mathrm{ref}} t^2}{\log(1 + \beta)^2} \right), \qquad \mathbb{P}\left( \left| \Delta_\pi(\overline{d}) \right| \ge t \right) \le 2 \exp\left( -\frac{2 n_\pi t^2}{\log(1 + \tfrac{1}{\alpha})^2} \right).$$

Choose

$$t_{\mathrm{ref}} := \log(1 + \beta) \sqrt{\frac{\log(2M/\delta)}{2 n_{\mathrm{ref}}}}, \qquad t_\pi := \log\left(1 + \tfrac{1}{\alpha}\right) \sqrt{\frac{\log(2M/\delta)}{2 n_\pi}}.$$

Then $\mathbb{P}\left( \left| \Delta_{\mathrm{ref}}(\overline{d}) \right| \ge t_{\mathrm{ref}} \right) \le \delta/M$ and $\mathbb{P}\left( \left| \Delta_\pi(\overline{d}) \right| \ge t_\pi \right) \le \delta/M$. Taking a union bound over all $\overline{d} \in \overline{\mathcal{D}}$ yields, with probability at least $1 - \delta$,

$$\sup_{\overline{d} \in \overline{\mathcal{D}}} \left| \widehat{\mathcal{R}}(\overline{d}) - \mathcal{R}(\overline{d}) \right| \le \tfrac{1}{2} t_{\mathrm{ref}} + \tfrac{1}{2} t_\pi.$$

Finally, since $n = \min\{n_{\mathrm{ref}}, n_\pi\}$, we have

$$\sqrt{\frac{\log(2M/\delta)}{2 n_{\mathrm{ref}}}} \le \sqrt{\frac{\log(M/\delta)}{n}}, \qquad \sqrt{\frac{\log(2M/\delta)}{2 n_\pi}} \le \sqrt{\frac{\log(M/\delta)}{n}},$$

up to benign constant factors that we absorb into the front constant. Using the definition of $B$ and setting $\eta = \sqrt{\log(M/\delta)/n}$ gives

$$\sup_{\overline{d} \in \overline{\mathcal{D}}} \left| \widehat{\mathcal{R}}(\overline{d}) - \mathcal{R}(\overline{d}) \right| \le 2 B \eta,$$

which is Equation 50. $\qquad \square$

We now prove the transfer bounds that move deviations on the cover element $\overline{d}$ back to an arbitrary $d$, measured either in the original loss or the empirical loss. These inequalities will let us relate risk differences to $L_1$ discrepancies between ratio functions.

**Lemma 4** (Transfer bounds from a cover element to an arbitrary discriminator). *Let Assumption 1 hold true. Let $L_+ := \frac{\beta}{1+\beta}$, $L_- := \frac{1}{1+\alpha}$, and define*

$$C_\triangle := \frac{L_+ + \beta L_-}{2\alpha}.$$

*Then:*

$$\big| \mathcal{R}(\overline{d}) - \mathcal{R}(d) \big| \leq C_\triangle \, \gamma, \tag{51}$$

$$\big| \widehat{\mathcal{R}}(\overline{d}) - \widehat{\mathcal{R}}(d) \big| \leq C_\triangle \, \gamma. \tag{52}$$

*Proof.* Start from the definition

$$\mathcal{R}(d) = \tfrac{1}{2} \mathbb{E}_{\mu_{\pi_{\mathrm{ref}}}}\big[ \log(1 + e^d) \big] + \tfrac{1}{2} \mathbb{E}_{\mu_\pi}\big[ \log(1 + e^{-d}) \big].$$

By the triangle inequality,

$$\big| \mathcal{R}(\overline{d}) - \mathcal{R}(d) \big| \leq \tfrac{1}{2} \mathbb{E}_{\mu_{\pi_{\mathrm{ref}}}} \big| \log(1 + e^{\overline{d}}) - \log(1 + e^d) \big| + \tfrac{1}{2} \mathbb{E}_{\mu_\pi} \big| \log(1 + e^{-\overline{d}}) - \log(1 + e^{-d}) \big|. \tag{53}$$

Recall the Lipschitz bounds derived in Lemma 2:

$$\big| \log(1 + e^u) - \log(1 + e^v) \big| \leq L_+ \, |u - v|, \qquad \big| \log(1 + e^{-u}) - \log(1 + e^{-v}) \big| \leq L_- \, |u - v|$$

yield from Equation 53

$$\big| \mathcal{R}(\overline{d}) - \mathcal{R}(d) \big| \leq \tfrac{1}{2} L_+ \, \mathbb{E}_{\mu_{\pi_{\mathrm{ref}}}}\big[ |\overline{d} - d| \big] + \tfrac{1}{2} L_- \, \mathbb{E}_{\mu_\pi}\big[ |\overline{d} - d| \big]. \tag{54}$$

Since $\mu_\pi = L \cdot \mu_{\pi_{\mathrm{ref}}}$ and $L \leq \beta$ by boundedness from Assumption 1,

$$\mathbb{E}_{\mu_\pi}\big[ |\overline{d} - d| \big] = \mathbb{E}_{\mu_{\pi_{\mathrm{ref}}}}\big[ L \, |\overline{d} - d| \big] \leq \beta \, \mathbb{E}_{\mu_{\pi_{\mathrm{ref}}}}\big[ |\overline{d} - d| \big]. \tag{55}$$

Plug Equation 55 back into Equation 54:

$$\big| \mathcal{R}(\overline{d}) - \mathcal{R}(d) \big| \leq \frac{L_+ + \beta L_-}{2} \, \mathbb{E}_{\mu_{\pi_{\mathrm{ref}}}}\big[ |\overline{d} - d| \big]. \tag{56}$$

Next, we need to conver $\mathbb{E}_{\mu_{\pi_{\mathrm{ref}}}}\big[ |\overline{d} - d| \big]$ to $\mathbb{E}_{\mu_{\pi_{\mathrm{ref}}}}\big[ |\overline{L} - L| \big]$. Use the mean value theorem for $\log$ on $[\alpha, \beta]$,

$$|\overline{d} - d| = |\log \overline{L} - \log L| = \frac{1}{\xi} |\overline{L} - L| \leq \frac{1}{\alpha} |\overline{L} - L|, \qquad \xi \text{ between } \overline{L} \text{ and } L.$$

Therefore, we have

$$\mathbb{E}_{\mu_{\pi_{\mathrm{ref}}}}\big[ |\overline{d} - d| \big] \leq \frac{1}{\alpha} \mathbb{E}_{\mu_{\pi_{\mathrm{ref}}}}\big[ |\overline{L} - L| \big]. \tag{57}$$

Combining Equation 56 and Equation 57:

$$\big| \mathcal{R}(\overline{d}) - \mathcal{R}(d) \big| \leq C_\triangle \, \mathbb{E}_{x \sim \mu_{\pi_{\mathrm{ref}}}}\big[ |\overline{L}(x) - L(x)| \big], \quad C_\triangle = (L_+ + \beta L_-)/(2\alpha).$$

Finally, according to the $L_1$ optimistic cover of Assumption 1, we have

$$\mathbb{E}_{\mu_{\pi_{\mathrm{ref}}}}\big[ |\overline{L} - L| \big] \leq \gamma$$

We get

$$\big| \mathcal{R}(\overline{d}) - \mathcal{R}(d) \big| \leq C_\triangle \, \gamma.$$

The same derivation holds if we replace expectations by empirical averages (sample means). Every inequality we used above (triangle inequality and the Lipschitz bounds) is pointwise and hence holds averaging over a finite sample instead of the distribution. Concretely,

$$\big| \widehat{\mathcal{R}}(\overline{d}) - \widehat{\mathcal{R}}(d) \big| \leq C_\triangle \sum_{i=1}^{n_{\mathrm{ref}}} \big[ |\overline{L}(x) - L(x)| \big] \leq C_\triangle \, \gamma$$

where the last inequality uses the empirical $L_1$ closeness of $\overline{L}$ and $L$.

$\square$

We now control the excess loss of the empirical minimizer by combining the uniform deviation bound over the optimistic cover with the transfer inequalities.

**Lemma 5** (Excess-loss bound for the empirical minimizer). *Let Assumption 1 hold true. Let $B = \frac{1}{2}\log(1+\beta) + \frac{1}{2}\log(1+1/\alpha)$, $C_\triangle = (L_+ + \beta L_-)/(2\alpha)$ with $L_+ = \beta/(1+\beta)$ and $L_- = 1/(1+\alpha)$, and $\eta = \sqrt{\log(M/\delta)/n}$ where $n = \min\{n_{\mathrm{ref}}, n_\pi\}$ defined as before. Then, with probability at least $1 - \delta$,*

$$\mathcal{R}(\widehat{d}) - \mathcal{R}(d^\star) \ \leq \ 3\,C_\triangle\,\gamma \ + \ 4\,B\,\eta. \tag{58}$$

*Proof.* Since $\widehat{d} \in \arg\min_{d \in \mathcal{D}} \widehat{\mathcal{R}}(d)$, we have:

$$\widehat{\mathcal{R}}(\widehat{d}) \ \leq \ \widehat{\mathcal{R}}(d) \qquad \text{for all } d,$$

in particular for $d = d^\star$ and for $d = \bar{d}^\star$ (the cover of $d^\star$, $\bar{d}^\star = \log \bar{L}^\star$). Start with a standard add-subtract trick:

$$\mathcal{R}(\widehat{d}) - \mathcal{R}(d^\star) = \left(\mathcal{R}(\widehat{d}) - \widehat{\mathcal{R}}(\widehat{d})\right) + \underbrace{\left(\widehat{\mathcal{R}}(\widehat{d}) - \widehat{\mathcal{R}}(\bar{d}^\star)\right)}_{\leq 0} + \left(\widehat{\mathcal{R}}(\bar{d}^\star) - \mathcal{R}(\bar{d}^\star)\right) + \left(\mathcal{R}(\bar{d}^\star) - \mathcal{R}(d^\star)\right), \tag{59}$$

For the first difference in Equation 59, insert the cover element $\bar{d} = \log \bar{L}$ of $\widehat{d}$ and apply the transfer bounds (Lemma 4) and the uniform deviation bound over the finite cover (Lemma 3):

$$\begin{aligned}
\mathcal{R}(\widehat{d}) - \widehat{\mathcal{R}}(\widehat{d}) &= \left(\mathcal{R}(\widehat{d}) - \mathcal{R}(\bar{d})\right) + \left(\mathcal{R}(\bar{d}) - \widehat{\mathcal{R}}(\bar{d})\right) + \left(\widehat{\mathcal{R}}(\bar{d}) - \widehat{\mathcal{R}}(\widehat{d})\right) \\
&\leq \ \left|\mathcal{R}(\widehat{d}) - \mathcal{R}(\bar{d})\right| + \left|\mathcal{R}(\bar{d}) - \widehat{\mathcal{R}}(\bar{d})\right| + \left|\widehat{\mathcal{R}}(\bar{d}) - \widehat{\mathcal{R}}(\widehat{d})\right| \\
&\leq \ C_\triangle\gamma \ + \ 2B\eta \ + \ C_\triangle\gamma.
\end{aligned}$$

The first term uses the transfer bounds from $\widehat{d}$ to its cover $\bar{d}$. The middle term uses the uniform deviation bound on the finite cover. It applies directly because $\bar{d} \in \overline{\mathcal{D}}$. The third term uses the empirical transfer bounds. Thus, we have

$$\mathcal{R}(\widehat{d}) - \widehat{\mathcal{R}}(\widehat{d}) \ \leq \ 2C_\triangle\gamma + 2B\eta. \tag{60}$$

Returning to Equation 59, the middle term is nonpositive by optimality, and the remaining two terms are bounded by the same two lemmas:

$$\left|\widehat{\mathcal{R}}(\bar{d}^\star) - \mathcal{R}(\bar{d}^\star)\right| \leq 2B\eta, \qquad \left|\mathcal{R}(\bar{d}^\star) - \mathcal{R}(d^\star)\right| \leq C_\triangle\gamma.$$

Combining with Equation 60 yields

$$\mathcal{R}(\widehat{d}) - \mathcal{R}(d^\star) \ \leq \ (2C_\triangle\gamma + 2B\eta) + 2B\eta + C_\triangle\gamma \ = \ 3C_\triangle\gamma + 4B\eta,$$

which is Equation 58. $\square$

Finally, we derive the occupancy ratio error bound.

**Theorem 3** (Occupancy ratio $L_1$ error bound). *Let Assumption 1 hold true. Let*

$$B \ := \ \tfrac{1}{2}\log(1 + \beta) + \tfrac{1}{2}\log\left(1 + 1/\alpha\right), \qquad L_+ \ := \ \frac{\beta}{1 + \beta}, \qquad L_- \ := \ \frac{1}{1 + \alpha},$$

$$C_\triangle \ := \ \frac{L_+ + \beta L_-}{2\alpha}, \qquad \lambda \ := \ \frac{\min\{\alpha, \beta\}}{(1 + \max\{\alpha, \beta\})^2} > 0,$$

*and $n := \min\{n_{\mathrm{ref}}, n_\pi\}$, $\eta := \sqrt{\log(M/\delta)/n}$ for any $\delta \in (0, 1)$. Let $\widehat{d} \in \arg\min_{d \in \mathcal{D}} \widehat{\mathcal{R}}(d)$ be the empirical minimizer, $\widehat{L} := e^{\widehat{d}}$, and $L^\star = e^{d^\star}$ be the true ratio. Then, with probability at least $1 - \delta$,*

$$\mathbb{E}_{x \sim \mu_{\pi_{\mathrm{ref}}}}\left[\left|\widehat{L}(x) - L^\star(x)\right|\right] \ \leq \ \beta\sqrt{\frac{4}{\lambda}}\ \sqrt{3\,C_\triangle\,\gamma \ + \ 4\,B\,\eta}. \tag{61}$$

*Proof.* By Lemma 5 (excess-risk bound for the empirical minimizer),

$$\mathcal{R}(\widehat{d}) - \mathcal{R}(d^\star) \leq 3\,C_\triangle\,\gamma + 4\,B\,\eta \quad \text{with probability at least } 1 - \delta.$$

From Lemma 1 we have,

$$\mathbb{E}_{x \sim \mu_{\mathrm{mix}}}\big[\big(\widehat{d}(x) - d^\star(x)\big)^2\big] \;\leq\; \frac{2}{\lambda}\big(\mathcal{R}(\widehat{d}) - \mathcal{R}(d^\star)\big) \;\leq\; \frac{2}{\lambda}\big(3\,C_\triangle\,\gamma + 4\,B\,\eta\big),$$

where $\mu_{\mathrm{mix}} = \frac{1}{2}\,\mu_{\pi_{\mathrm{ref}}} + \frac{1}{2}\,\mu_\pi$. Since $\mu_{\mathrm{mix}} \geq \frac{1}{2}\,\mu_{\pi_{\mathrm{ref}}}$, we have $\mathbb{E}_{\mu_{\pi_{\mathrm{ref}}}}[\cdot] \leq 2\,\mathbb{E}_{\mu_{\mathrm{mix}}}[\cdot]$, and

$$\mathbb{E}_{x \sim \mu_{\pi_{\mathrm{ref}}}}\big[\big(\widehat{d}(x) - d^\star(x)\big)^2\big] \;\leq\; \frac{4}{\lambda}\big(3\,C_\triangle\,\gamma + 4\,B\,\eta\big).$$

Finally, by the third point of Lemma 2 (the exponential map is $\beta$–Lipschitz on $[\log\alpha, \log\beta]$) and Cauchy–Schwarz,

$$\mathbb{E}_{\mu_{\pi_{\mathrm{ref}}}}\big[|\widehat{L} - L^\star|\big] \;\leq\; \beta\,\mathbb{E}_{\mu_{\pi_{\mathrm{ref}}}}\big[|\widehat{d} - d^\star|\big] \;\leq\; \beta\,\sqrt{\mathbb{E}_{\mu_{\pi_{\mathrm{ref}}}}[(\widehat{d} - d^\star)^2]} \;\leq\; \beta\,\sqrt{\frac{4}{\lambda}}\,\sqrt{3\,C_\triangle\,\gamma + 4\,B\,\eta},$$

which is Equation 61. $\qquad\square$

## G.2 Guarantees for Max-Min with Occupancy Measure Approximation

In this section, we establish convergence guarantees for our Max-Min Algorithm 2. Our analysis follows the Reinforcement Learning with General Utility (RLGU) (Zhang et al., 2022; Barakat et al., 2024), where given a utility function $F(\cdot), \theta \mapsto F(\mu_{\pi_\theta})$ over the policy-induced occupancy measure $\mu_{\pi_\theta}$, the goal of RLGU is to find a policy $\pi_\theta^\star$ such that $\pi_\theta^\star \in \arg\max_\theta F(\mu_{\pi_\theta})$. In RLGU, there is no reward function. Instead, we can view $\nabla_\theta F(\mu_\theta)$ as a pseudo-reward depending on the unknown occupancy measure induced by the policy. The procedure for solving RLGU follows three steps: (i) estimate the occupancy $\mu_{\pi_\theta}$ (e.g., by MLE), (ii) form the pseudo-reward from this estimate, and (iii) update the policy. Our Max-Min algorithm mirrors this pipeline. Specifically, we first estimate the occupancy ratio by training a discriminator. Then we construct the worst-case reward using Equation 19 from the estimation. Finally, we perform a policy update. Consequently, the general RLGU sample complexity guarantees apply to our algorithm after replacing the pseudo-reward $\nabla_\theta F(\mu_\theta)$ with our worst-case reward and substituting their occupancy-estimation error with our occupancy-ratio error obtained above. We formalize this correspondence and state the resulting bounds below.

For each iteration of our Max-Min algorithm $t = 1, 2, ..., T$, let the pseudo-reward $r_t(s, a)$ is defined as in Equation 19. Let's define

$$F(\mu_t) := \frac{1}{4\lambda_3} \int \frac{\mu_t(s,a)^2}{\mu_{\pi_{\mathrm{ref}}}(s,a)}\,\mathrm{d}(s,a) \;-\; \int c(s,a)\,\mu_t(s,a)\,\mathrm{d}(s,a), \tag{62}$$

where $c(s,a) := \frac{1}{2\lambda_3}\big(\lambda_1 \frac{R_{\mathrm{proxy}}(s,a)}{V} + \lambda_2\big)$. By construction, we have $r_t(s,a) = \nabla_\mu F(\mu_t)(s,a)$, which means the utility gradient in $\mu$ is exactly the pseudo-reward. Let's $\widehat{\mu}_t := \widehat{L}_t \cdot \mu_{\pi_{\mathrm{ref}}}$ be the occupancy estimator, $\widehat{r}_t(s,a) := \nabla_\mu F(\widehat{\mu}_t)(s,a)$ be the estimated pseudo-reward and $F^\star \in \max_\theta F(\mu_{\pi_\theta})$ be the maximum.

We next introduce some assumptions that are required for our results, which are adapted from (Barakat et al., 2024).

**Assumption 2** (Policy parametrization, Assumption 6 from (Barakat et al., 2024)). *For every $(s,a) \in \mathcal{S} \times \mathcal{A}$ and every $\theta \in \mathbb{R}^d$, the policy has full support, i.e, $\pi_\theta(a \mid s) > 0$. Moreover, the mapping $\theta \mapsto \pi_\theta(a \mid s)$ is continuously differentiable, and the score function $\theta \mapsto \nabla_\theta \log \pi_\theta(a \mid s)$ is uniformly bounded:*

$$\big\|\nabla_\theta \log \pi_\theta(a \mid s)\big\| \;\leq\; l_\psi \qquad \text{for some constant } l_\psi > 0 \text{ and all } (s,a),\ \theta.$$

This assumption typically holds in practice, for instance, with the standard softmax policy parameterization. Next, we make a smoothness assumption on the utility function, which is crucial in deriving the final convergence bound. We also verify that the defined utility function in Equation 62 satisfies the smoothness assumption.

**Assumption 3** (General utility smoothness, Assumption 7 from (Barakat et al., 2024)). *For utility function $F(\cdot), \theta \mapsto F(\mu_{\pi_\theta})$, there exist constants $L_\mu > 0$ such that for all $\mu_1, \mu_2 \in \mathcal{X}$,*

$$\left\|\nabla_\mu F(\mu_1)\right\|_2 \leq \ell_\mu \qquad and \qquad \left\|\nabla_\mu F(\mu_1) - \nabla_\mu F(\mu_2)\right\|_2 \leq L_\mu \left\|\mu_1 - \mu_2\right\|_2.$$

Notice that Assumption 3 holds in our setting since Hessian is diagonal with entries at most $\nabla_\mu^2 F(\mu)(s,a) = 1/(2\lambda_3 \mu_{\pi_{\mathrm{ref}}}(s,a))$. Thus, if $\mu_{\pi_{\mathrm{ref}}}(s,a) \geq \rho_{\min} > 0$ on the support of all $(s,a)$, which we assume it holds, then we have that $\nabla_\mu F(\mu)$ is $L_\mu$-Lipschitz with

$$\left\|\nabla_\mu F(\mu) - \nabla_\mu F(\mu')\right\|_2 \leq L_\mu \|\mu - \mu'\|_2, \qquad L_\mu = \frac{1}{2|\lambda_3|} \rho_{\min}^{-1},$$

Under Assumptions 2 and 3, the utility function $\theta \mapsto F(\mu_{\pi_\theta})$ is $L_\theta$-smooth. Using these properties, our Max-Min algorithm admits the following first-order stationarity guarantee:

**Theorem 4** (Guarantee for the Max-Min update). *Assume Assumptions 2 and 3 hold. Let $N$ be the batch size for estimating the policy gradient at each iteration, $\alpha$ be the stepsizes satisfying $\alpha_t \leq 1/(2L_\theta)$, $K_{\mathrm{conv}} := \|\mu_{\pi_{\mathrm{ref}}}\|_\infty (\beta - \alpha)$ and*

$$\varepsilon_L := \beta \sqrt{\frac{4}{\lambda}} \sqrt{3\, C_\triangle \,\gamma + 4\, B\, \eta} \geq \mathbb{E}_{x \sim \mu_{\pi_{\mathrm{ref}}}}\left[\,|\widehat{L}(x) - L^\star(x)|\,\right]$$

*Then we have:*

$$\mathbb{E}\Big[\big\|\nabla_\theta F\big(\mu_{\pi_{\theta_\tau}}\big)\big\|^2\Big] \leq \frac{16\left(F^\star - \mathbb{E}[\,F(\mu_{\pi_{\theta_1}})\,]\right)}{\alpha\, T} + \frac{C_1}{N} + C_2\, K_{\mathrm{conv}}\, \varepsilon_L, \qquad (63)$$

*where $\tau$ is drawn uniformly from $\{1, \ldots, T\}$, expectation is w.r.t all randomness (in $(\theta_t)$ and $\tau$), $C_1 = \frac{8\ell_\mu^2 l_\psi^2}{(1-\gamma')}$ and $C_2 = \frac{8l_\psi^2 L_\mu^2}{(1-\gamma')^4}$ with $\gamma'$ be the discount factor in RL.*

*Proof.* Since we already verified that Assumptions 2 and 3 hold in our setting, according to Theorem 8 in (Barakat et al., 2024)), we directly have

$$\frac{1}{T}\sum_{t=1}^T \mathbb{E}\big[\|\nabla_\theta F\big(\mu_{\pi_{\theta_\tau}}\big)\|^2\big] \leq \frac{16\left(F^\star - \mathbb{E}[F(\mu_{\pi_{\theta_1}})]\right)}{\alpha\, T} + \frac{C_1}{N} + C_2\, \frac{1}{T}\sum_{t=1}^T \mathbb{E}[\|\widehat{\mu}_t - \mu_t\|_2^2].$$

To control the last term via the ratio error derived in Theorem 3, note that with $\widehat{\mu}_t = \widehat{L}_t\, \mu_{\pi_{\mathrm{ref}}}$ and $\mu_t = L_t\, \mu_{\pi_{\mathrm{ref}}}$,

$$\|\widehat{\mu}_t - \mu_t\|_2^2 = \sum_{s,a} \mu_{\pi_{\mathrm{ref}}}(s,a)^2 \left(\widehat{L}_t(s,a) - L_t(s,a)\right)^2 \leq \|\mu_{\pi_{\mathrm{ref}}}\|_\infty (\beta - \alpha) \sum_{s,a} \mu_{\pi_{\mathrm{ref}}}(s,a)\, |\widehat{L}_t - L_t|,$$

using $x^2 \leq (\beta - \alpha)|x|$ for $|x| \leq \beta - \alpha$ and $\mu_{\pi_{\mathrm{ref}}}^2 \leq \|\mu_{\pi_{\mathrm{ref}}}\|_\infty \mu_{\pi_{\mathrm{ref}}}$. Therefore,

$$\mathbb{E}\big[\|\widehat{\mu}_t - \mu_t\|_2^2\big] \leq K_{\mathrm{conv}}\, \mathbb{E}_{x \sim \mu_{\pi_{\mathrm{ref}}}}\big[\,|\widehat{L}_t(x) - L_t(x)|\,\big] \leq K_{\mathrm{conv}}\, \varepsilon_L.$$

Averaging over $t$ and drawing $\tau$ uniformly from $\{1, \ldots, T\}$ yields Equation 63. $\qquad\square$

### G.3 CONVERGENCES FOR LINEAR MAX-MIN ALGORITHM

For our Linear Max-Min algorithm, it is challenging to derive a convergence bound directly. However, as discussed in Appendix E.6, the inner optimization problem, i.e., finding the worst-case reward for a given policy, admits a **globally optimal closed-form solution** under our formulation in the tabular setting. Therefore, for any given policy $\pi$, we have access to an oracle that outputs the optimal worst-case reward $R^*$, and our Linear Max-Min algorithm can be viewed as alternating between gradient ascent on $\pi$ and the optimal minimization on $R^*$. As shown in Section 4 of (Jin et al., 2020), our algorithm converges, and the resulting policy $\pi$ corresponds to an approximate stationary point of the outer optimization problem.

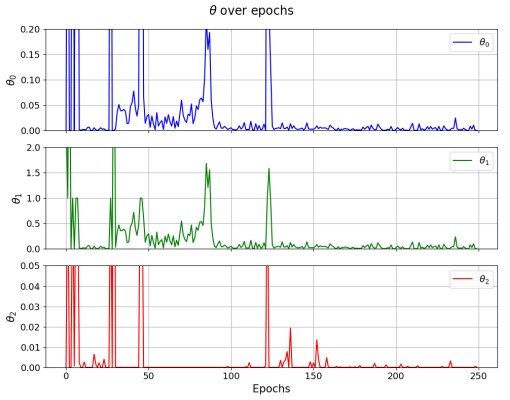 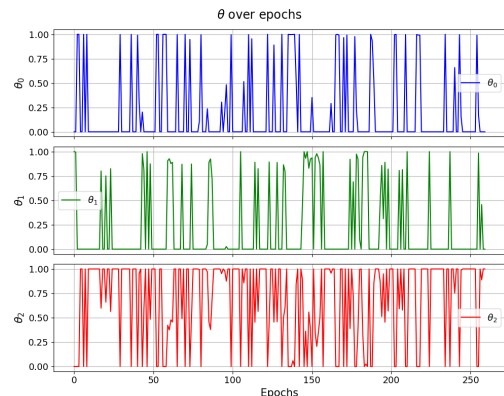

(a) Traffic environment: $\theta_0$ (velocity), $\theta_1$ (acceleration), $\theta_2$ (headway).

(b) Pandemic environment: $\theta_0$ (infection summary), $\theta_1$ (early-stage), $\theta_2$ (smoothness).

Figure 4: Evolution of adversarial reward weights $\boldsymbol{\theta}$ over training epochs for different environments using the Linear Maxmin method.

## H  ADDITIONAL EXPERIMENT RESULTS

### H.1  FEATURE WEIGHTS IN LINEAR MAX-MIN OPTIMIZATION DURING TRAINING

Figure 4 visualizes the evolution of each component of the linear worst-case reward weight vector $\boldsymbol{\theta}$ during training in the Traffic and Pandemic environments. We observe distinct behaviors in the dynamics of $\boldsymbol{\theta}$ across tasks.

In the Traffic environment (Figure 4a), we observe that the three $\theta$ parameters vary significantly in scale. Specifically, $\theta_1$ (acceleration) exhibits the largest magnitude, ranging from 0 to 2, while $\theta_2$ (headway) has the smallest scale, ranging from 0 to 0.05. This highlights how the linear max-min algorithm assigns different levels of penalization to each feature. Moreover, we also observe distinct phases in the dynamics of $\boldsymbol{\theta}$ over the course of training. In the early epochs ($<$50), all components, especially $\theta_1$ (acceleration) and $\theta_2$ (headway), exhibit high-frequency fluctuations. At this point, the dual optimization problem is not yet well-conditioned, and the adversarial reward is highly sensitive to small changes in occupancy or feature values. As training progresses ($\sim$epochs 100–250), the parameters begin to stabilize. Most notably, $\theta_2$ (headway) converges close to zero and remains suppressed, indicating that the worst-case reward does not emphasize this feature. This may suggest that headway is less harmful under adversarial reweighting compared to others (velocity or accel) or is already well aligned with the reference policy $\pi_{\text{ref}}$. Meanwhile, $\theta_1$ (acceleration) consistently exhibits higher values and sharper spikes than the other components. This indicates that acceleration plays a dominant role in the adversarial reward, likely because policies that optimize for the proxy reward tend to exploit aggressive acceleration patterns that diverge significantly from the behavior of $\pi_{\text{ref}}$. In contrast, $\theta_0$ (velocity) remains small and relatively stable throughout training, suggesting that speed alone is not strongly penalized under adversarial interpretations.

Overall, the observed pattern reflects the interpretability and sparsity benefits of the linear max-min formulation. The model is able to selectively emphasize features that are most vulnerable to reward hacking, while suppressing those that are either irrelevant or well-aligned. This structured behavior supports the practical value of using linearly parameterized worst-case rewards to improve policy robustness.

In the Pandemic environment (Figure 4b), unlike the Traffic environment, where $\theta$ converged to a sparse and interpretable solution, we observe high variability across all components throughout training. In particular, we find the following pattern:

1. **Persistent fluctuations.** All three components exhibit frequent oscillations over the course of 260 epochs. This ongoing instability suggests that the adversarial reward continually

adapts as the policy changes, likely due to the environment's temporal sensitivity and complex dynamics.

2. $\theta_2$ **(smoothness) remains active.** The smoothness-related component $\theta_2$ is frequently non-zero and relatively stable compared to the others. This indicates that the worst-case reward consistently emphasizes penalizing erratic or unstable responses in the infection trajectory — a behavior often neglected by naive proxy metrics.

3. $\theta_1$ **(early-stage transitions) is highly volatile.** The component associated with early infection stage changes spikes intermittently. This suggests that early-stage mismanagement is a recurring vulnerability in the learned policy that the adversarial reward seeks to exploit.

4. $\theta_0$ **(overall infection) activates intermittently.** Although $\theta_0$ sometimes spikes, it does not dominate the adversarial reward. This may indicate that the learned policy already accounts for infection magnitude reasonably well, or that smoothness and early-stage control offer more leverage for reward hacking under the proxy constraint.

Overall, this pattern highlights that in more dynamic and temporally complex environments like Pandemic, the worst-case reward remains non-sparse and adapts to different policy weaknesses throughout training. In contrast to the Traffic environment, adversarial emphasis here is broader and more reactive.

## H.2 ADDITIONAL WORST-CASE PERFORMANCE RESULTS

Table 6: Evaluation results on Traffic, Pandemic, Glucose, and RLHF environments. All policies are trained using **only the proxy reward**. In Traffic, the proxy reward is based on *vel*, *accel*, *headway* (1, 1, 0.1), while the true reward uses *commute*, *accel*, *headway* (1, 1, 0.1). In Pandemic, the proxy reward includes *infection*, *lower stage*, *smooth changes* (10, 0.1, 0.01), while the true reward additionally includes *political* with weight 10 after *infection*. In Glucose, the proxy uses *expected patient cost*, and the true reward uses *magni_bg*. In RLHF, the proxy uses a 70M LLM, and the true reward uses a 8B LLM. We report $\theta$ in the same order as feature weights. **Occ** denotes total occupancy over state-action pairs unseen by $\pi_{ref}$, where discriminator outputs infinity.

| Env | Traffic | | | | | |
|---|---|---|---|---|---|---|
| **Method** | **True** | **Proxy** | **Worst** | **Linear Worst ($\theta$)** | **Linear Worst* ($\theta$)** | **Occ $\downarrow$** |
| ORPO | 16.91±0.12 | 3.41±0.13 | -1.96e+04±0.02e+04 | -0.69±0.01 (0.71, 0.21, 0.69) | -0.83±0.02 (0.63, 0.12, 0.97) | 3.82e-04 ±0.13e-04 |
| ORPO* | 10.26±0.09 | 1.35±0.09 | -1.35e+04±0.02e+04 | -0.44±0.02 (0.46, 0.18, 0.86) | -0.45±0.01 (0.58, 0.06, 0.81) | 1.84e-04±0.07e-04 |
| Max-Min | 12.70±0.06 | 3.63±0.09 | -268.31±4.14 | -0.06±0.01 (0.01, 0.02, 0.96) | -0.06±0.01 (0.001, 0.02, 0.99) | 0.00±0.00 |
| Linear Max-Min | 16.46±0.10 | 2.40±0.11 | -1.19e+04±0.01e+04 | 0.20±0.01 (0.64, 0.07, 0.76) | -0.12±0.01 (0.91, 0.01, 0.67) | 0.00±0.00 |
| **Env** | **Pandemic** | | | | | |
| **Method** | **True** | **Proxy** | **Worst** | **Linear Worst ($\theta$)** | **Linear Worst* ($\theta$)** | |
| ORPO | -1.04±0.21 | 1.75±0.19 | -5.31e+06±0.01e+06 | -2.41±0.02 (0.23, 0.95, 0.17) | -2.65±0.02 (0.02, 0.95, 0.92, 0.08) | |
| ORPO* | 1.18±0.19 | 1.18±0.19 | -4.46e+06±0.03e+06 | -1.36±0.01 (0.25, 0.97, 0.13) | -1.36±0.01 (0.25, 0, 0.97, 0.13) | |
| Max-Min | 1.25±0.18 | 1.25±0.18 | -63.29±3.35 | -1.11±0.01 (0.14, 0.99, 0.01) | -1.11±0.01 (0.14, 0, 0.99, 0.01) | |
| Linear Max-Min | 3.65±0.11 | 7.60±0.13 | -6.82e+05±0.01e+05 | 0.65±0.01 (0.001, 0.23, 0.02) | -0.17±0.02 (0.01, 0.97, 0.22, 0.09) | |
| **Env** | **Glucose** | | | **RLHF** | | |
| **Method** | **True($\times 10^3$)** | **Proxy** | **Worst** | **True** | **Proxy** | **Worst** |
| ORPO | 6.0±0.1 | 100.48±0.54 | -27.54±0.32 | 8.30 ± 1.07 | 0.63±0.21 | -1.84±0.03 |
| ORPO* | 6.3±0.2 | 116.36±0.56 | -8.79±0.27 | N/A | N/A | N/A |
| Max-Min | 6.3±0.1 | 102.66±0.58 | -1.71±0.25 | 5.38 ± 0.92 | 0.84±0.11 | -0.10±0.01 |

**Adversarial Weight Analysis.** In Table 6, we also report the adversarial weight vectors $\theta$ for each policy. These weights reveal which features are most vulnerable to proxy exploitation under the learned policy and can be used to diagnose and revise the proxy reward function, thereby improving robustness. This highlights the interpretability benefits of our framework. Moreover, several patterns emerge from the results. In the Traffic environment, first, we observe a clear dominance of the headway feature, with all methods assigning it the highest weight. This suggests that headway is the most critical component exposed to reward hacking under correlation constraints. Second, the acceleration feature is consistently downweighted across all methods. This indicates that acceleration may be less prone to exploitation or already well aligned with the reference policy. Third, the velocity feature is moderately emphasized by Linear Max-Min and ORPO (e.g., 0.64 and 0.71), while Max-Min nearly suppresses it (0.01). This contrast suggests that Linear Max-Min anticipates some vulnerability from velocity deviations, while Max-Min focuses almost entirely on headway. In the Pandemic environment, first, both ORPO* and Max-Min assign zero weight to the political feature. This occurs because the expected feature value under their policies is exactly zero, making the correlation constraint inactive for that dimension. Interestingly, this feature plays a significant role in the adversarial rewards for both ORPO and Linear Max-Min, with their corresponding $\theta$ assigning non-negligible weight to it (e.g., 0.95 and 0.97 respectively). This suggests

that these policies expose themselves to vulnerability in feature dimensions that are entirely ignored by `Max-Min` and `ORPO*`. Second, the lower stage feature consistently receives the highest weight across all methods, indicating it is the most sensitive component under proxy misalignment.

Table 7: Evaluation results on Tomato environments. All policies are trained using **only the proxy reward**. In Tomato, the proxy includes *number of watered tomatoes* plus a bonus at a specific state (sprinkler), while the true reward only measures *watered tomatoes*. **Occ** in the Tomato environment denotes total occupancy over state-action pairs unseen by $\pi_{\text{ref}}$, based on 1000 sampled trajectories. **Worst** refers to the expected worst-case reward computed while excluding those unseen state-action pairs. **Worst\*** denotes the actual expected worst-case reward, while $R_{\text{min}}$ represents the minimum possible reward of any state-action pair. All rewards are normalized according to the reference policy $\pi_{\text{ref}}$.

| Method | Tomato | | | | |
|---|---|---|---|---|---|
| | **True** | **Proxy** | **Worst** | **Occ** $\downarrow$ | **Worst\*** |
| ORPO | 6.28±0.22 | 6.83±0.28 | -1.51±0.09 | 2.50e-04±0.63e-04 | -1.51+$R_{\text{min}}$·2.50e-04 |
| ORPO* | 4.00±0.18 | 3.98±0.23 | -1.09±0.10 | 3.09e-05±0.59e-05 | -1.09+$R_{\text{min}}$·3.09e-05 |
| Max-Min | 4.56±0.20 | 4.68±0.25 | -1.37±0.06 | 1.01e-05±0.43e-05 | -1.37+$R_{\text{min}}$·1.01e-05 |

**Worst-Case Performance in Tomato Environment.** Table 7 reports worst-case performance results for the Tomato environment. We omit the `Linear Max-Min` policy from these experiments for the following reasons. In the Tomato environment, the reward structure is difficult to express in a clean feature-based form suitable for linear modeling. Therefore, we report only the results for the `Max-Min` policy alongside the baselines.

The results for the Tomato environments exhibit trends similar to those observed in other environments (Section 4.2). In particular, `ORPO*` appears to outperform others in the Tomato environment in terms of worst-case performance. Recall that these results are reported under **Worst**, the expected worst-case reward restricted to state-action pairs observed under $\pi_{\text{ref}}$. Since the Tomato environment is discrete, we can explicitly identify which state-action pairs are unseen through sampling, enabling clearer interpretation of their physical meaning as well as the evaluation of the actual worst-case performance **Worst\***. The latter corresponds to the **Worst** value plus the product of the occupancy in unseen regions (**Occ**) and $R_{\text{min}}$. Because `Max-Min` exhibits the lowest occupancy among all methods, it demonstrates greater robustness under varying assumptions about $R_{\text{min}}$.

Nevertheless, `ORPO*` still shows marked improvement over `ORPO`, both in worst-case return and in reducing occupancy over unseen state-action pairs. As previously noted, in the Glucose environment, the discriminator fails to detect any state-action pairs missed by the reference policy. This reinforces our earlier concern that the current discriminator training procedures may have limited capacity to identify rare or out-of-distribution events.

**Impact of Correlation Parameter Selection on Robustness.** In this section, we present additional experiment results to examine how the proxy-true reward correlation parameter $r$ used during training affects the robustness under varying evaluation $r$.

Table 8: Evaluation of robustness in the Tomato environment across different training-time correlation levels $r$. **Occ** denotes total occupancy over state-action pairs unseen by $\pi_{\text{ref}}$, based on 1000 sampled trajectories. **Worst** refers to the expected worst-case reward computed while excluding those unseen state-action pairs.

| $r$ | **Occ** | **Worst** ($r=0.1$) | **Worst** ($r=0.4$) | **Worst** ($r=0.7$) | **Worst** ($r=0.9$) |
|---|---|---|---|---|---|
| 0.1 | 1.36e-03±0.12e-03 | -1.34±0.05 | -1.12±0.04 | -0.74±0.03 | -0.27±0.02 |
| 0.4 | 1.01e-05±0.43e-05 | -1.66±0.07 | -1.37±0.06 | -0.70±0.03 | -0.05±0.02 |
| 0.7 | 1.05e-02±0.10e-02 | -2.10±0.08 | -1.82±0.07 | -1.33±0.06 | -0.66±0.04 |
| 0.9 | 1.29e-05±0.41e-05 | -9.10±0.20 | -8.92±0.18 | -7.60±0.15 | -5.49±0.12 |

Table 8 and Table 9 report the robustness evaluation results in the **Tomato** and **Traffic** environments under different training-time values of the correlation parameter $r$. Several consistent patterns emerge across both environments.

Table 9: Evaluation of robustness in the Traffic environment across different training-time correlation levels $r$. **Occ** denotes total occupancy over state-action pairs unseen by $\pi_{\text{ref}}$, based on 200 sampled trajectories. **Worst** refers to the expected worst-case reward computed while excluding those unseen state-action pairs.

| $r$ | **Occ** | **Worst** ($r = 0.1$) | **Worst** ($r = 0.3$) | **Worst** ($r = 0.5$) | **Worst** ($r = 0.9$) |
|-----|---------|------------------------|------------------------|------------------------|------------------------|
| 0.3 | 0.00±0.00 | -2794.63±42.10 | -268.31±4.14 | -82.07±2.04 | -22.03±0.88 |
| 0.5 | 0.00±0.00 | -7.71e+04±1.20e+03 | -1.95e+04±3.10e+02 | -6168.40±124.75 | -1350.22±27.95 |
| 0.9 | 9.66e-05±1.84e-05 | -3.01e+05±6.05e+03 | -9.51e+04±1.89e+03 | -2.73e+04±5.45e+02 | -9.33e+03± 1.88e+02 |

First, for any fixed policy (i.e., fixed training $r$), we observe that the expected worst-case return monotonically increases as the evaluation $r$ increases. This aligns with intuition: higher correlation levels correspond to smaller uncertainty sets over rewards, meaning the worst-case reward functions are less adversarial. In contrast, low $r$ values expand the reward uncertainty set, allowing more pathological or implausible reward functions, and thus lead to more pessimistic evaluations. However, this does not hold universally. For a fixed policy, the expected worst-case return in Equation 27 is monotone in $r$ only when the policy has a positive expected proxy return (which is the case here). If the policy's expected proxy return is negative, this monotonicity no longer holds.

Second, we find that training with a moderate correlation level, particularly around $r = 0.3$ to $0.4$, yields better robustness across a wide range of evaluation $r$ values. In contrast, training with overly small (e.g., $r = 0.1$ for Tomato) or large (e.g., $r = 0.9$ for Tomato and Traffic) correlation levels degrades robustness. A small $r$ leads to overly conservative training, anticipating extreme forms of reward hacking and thus hurting general performance. On the other hand, a high $r$ overly trusts the proxy reward and fails to hedge against potential deviations, resulting in poor worst-case behavior under reward misspecification. This trade-off highlights that intermediate values of $r$ may strike a better balance between conservativeness and optimism, enabling the policy to generalize to a broader and more plausible spectrum of reward functions. Therefore, in the absence of prior knowledge of $r$, starting with a moderate $r$ is a practical heuristic.

### H.3 Additional Results for Robustness Across Correlation Levels

As discussed previously (Appendix H.2), we do not include linear worst-case evaluation for the Glucose and Tomato environments. Consequently, we cannot perform a uniform search over $\theta$ as we do for the Traffic and Pandemic environments (Appendix F.5). As noted in Appendix B, it is generally difficult, and often infeasible, to sample a full reward function over all state-action pairs, particularly in high-dimensional or continuous environments, such as the Glucose environment. To approximate this process for the Tomato environment, which is a discrete environment, we instead sample 1000 trajectories using the reference policy $\pi_{\text{ref}}$. We then restrict the search to the visited state-action pairs. For each such pair, we perturb the original proxy reward by adding Gaussian noise with zero mean and variance sampled uniformly from the interval $[0.001, 1]$. We then check whether the resulting perturbed reward $\tilde{R}$ satisfies the constraint $\tilde{R} \in \mathcal{R}_{\text{corr}}$. As in previous evaluations, we do not explicitly constrain $M$ and $V$. We sample 20 perturbed reward functions and then use them to evaluate each policy. **Note:** Some policies, such as ORPO, may visit state-action pairs that are not included in the sampled set from $\pi_{\text{ref}}$. For these unseen state-action pairs, the proxy-true correlation constraint does not apply, as no corresponding reference data is available. In such cases, we default to using the original proxy reward to evaluate those portions of the trajectory.

We emphasize that this procedure is neither optimal nor efficient, and is employed solely for evaluation purposes in the Tomato environment. Designing more principled and scalable methods for reward sampling under correlation constraints remains an important direction for future work.

Figure 5 presents the average reward and standard deviation across varying correlation levels $r$ for the Tomato environment. As expected, the reference policy $\pi_{\text{ref}}$ (blue) consistently underperforms across all values of $r$, though it exhibits the lowest variance—indicating stable yet suboptimal behavior. Interestingly, ORPO* (purple) performs worse than ORPO (red) throughout, suggesting that improving the accuracy of occupancy measure estimation does not necessarily enhance robustness in this environment. In contrast, our Max-Min method (green) achieves the highest average reward across all correlation levels, highlighting its better robustness under reward uncertainty.

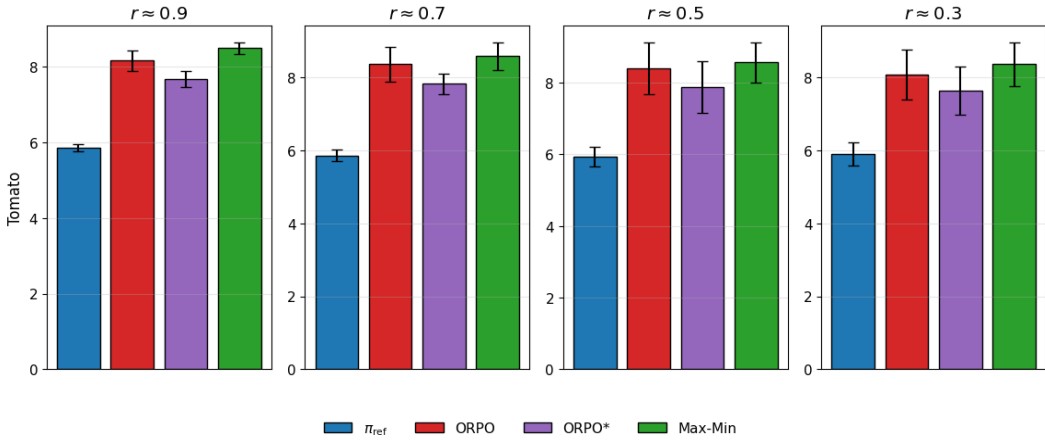

Figure 5: Mean reward and standard deviation under sampled reward functions at different proxy-true reward correlation levels $r$ for the Tomato environment. Our methods (`Max-Min`) yield higher average performance across all choices of $r$.

## H.4    ADDITIONAL UNNORMALIZED RESULTS

To ensure a fair comparison with prior work (Laidlaw et al., 2025), which reports results in the unnormalized reward scale, we also include the raw (unnormalized) expected proxy and true rewards. However, for worst-case reward metrics, it is nontrivial to reverse the normalization transformation, as our formulation explicitly constrains the reward to have zero mean and unit variance under the reference policy. Therefore, we omit worst-case results in the unnormalized setting.

Table 10: Unnormalized performance comparison across all environments.

| Method | Traffic | | Pandemic | | Glucose | | Tomato | | RLHF | |
|---|---|---|---|---|---|---|---|---|---|---|
| | True | Proxy | True | Proxy | True ($\times 10^3$) | Proxy | True | Proxy | True | Proxy |
| $\pi_{\text{ref}}$ | -1004.33±0.00 | 1474.30±0.00 | -12.01±0.00 | -12.01±0.00 | -79.7±0.0 | -117.75±0.00 | 5.96±0.00 | 6.37±0.00 | 15.97±0.00 | -0.29±0.00 |
| ORPO | -666.13±2.34 | 1542.57±2.62 | -12.84±0.17 | -10.61±0.15 | -49.7±0.6 | -67.51±0.27 | 9.10±0.11 | 9.10±0.11 | 16.51±0.07 | -0.23±0.02 |
| ORPO* | -799.02±1.77 | 1501.29±1.83 | -11.06±0.15 | -11.06±0.15 | -48.3±1.0 | -59.57±0.28 | 7.96±0.09 | 7.96±0.09 | N/A | N/A |
| Max-Min | -750.32±1.28 | 1546.86±1.82 | -11.01±0.14 | -11.01±0.14 | -48.2±0.5 | -66.42±0.29 | 8.24±0.10 | 8.24±0.10 | 16.32±0.06 | -0.21±0.01 |
| Linear Max-Min | -675.12±2.04 | 1522.34±2.19 | -9.90±0.09 | -5.93±0.10 | N/A | N/A | N/A | N/A | N/A | N/A |
| Ensemble | N/A | N/A | N/A | N/A | N/A | N/A | N/A | N/A | 16.12±0.08 | -0.17±0.01 |

Table 10 presents the unnormalized performance results across all environments. We observe that both our `Max-Min` and `Linear Max-Min` policies achieve comparable performance to `ORPO` on most tasks. Interestingly, the `ORPO*` variant (with a fully trained discriminator) outperforms the original `ORPO` in some environments (e.g., Pandemic and Glucose), but performs worse in others, such as Traffic and Tomato. While our earlier analysis (Section 4.2) shows that better discriminator training generally improves worst-case robustness, these results suggest that accurate discriminator estimation does not always translate to improved performance for every specific reward function. Understanding the nuanced effects of discriminator optimization on various reward metrics is beyond the scope of this paper and remains an important direction for future research.

## H.5    RESULTS FOR ALL $r$

Here, we report the results of a uniform grid search over $r \in [0.1, 0.9]$ for our Max-Min algorithm across all training-time correlation levels $r$ on the **Traffic**, **Tomato**, **Pandemic** and **Glucose** environments in Tables 11, 12, 13 and 14. Each table presents the mean and standard deviation of the expected true reward, expected worst-case reward, and occupancy measure achieved by the learned policy over five random seeds. Here, the "true reward" refers to the original (unnormalized) true reward, while the "worst reward" refers to the normalized reward. We also include the corresponding ORPO results for a fair comparison.

Note that the original ORPO algorithm performs a grid search over $\lambda$, where $\lambda = \sigma_{R_{\text{proxy}}} \sqrt{1 - r^2}$ and $\sigma_{R_{\text{proxy}}}$ is the standard deviation of $R_{\text{proxy}}$ under $\pi_{\text{ref}}$, which is generally unknown and must be

estimated. Thus, ORPO effectively searches over $\lambda$, whereas our method searches directly over $r$. For a fair comparison, we estimate $\sigma_{R_{\text{proxy}}}$ and use it to map each value of $r$ in our grid to a corresponding $\lambda$ for ORPO. However, across all environments we find that the resulting $\lambda$ values occupy a much narrower scale than $r$: while $r$ spans the full range from 0.1 to 0.9, the induced $\lambda$ values are confined to a small interval (e.g., approximately 0.021–0.05 in **Tomato** and 0.035–0.08 in **Pandemic**). As a consequence, the ORPO policies change only marginally across the mapped $r$ values, and their expected true and worst-case returns appear similar in the tables. In practice, ORPO would need to search over a broader range of $\lambda$ values. By contrast, the expected true and worst-case returns for our Max-Min method vary meaningfully across the full span of $r$. This highlights that, in practice, our method and ORPO naturally operate on different hyperparameter scales when tuning their respective robustness parameters.

Table 11: Evaluation in the Tomato environment across training-time correlation levels $r$ for ORPO and Max–Min. $\lambda$ denotes ORPO's coefficient, with $\lambda = \sigma_{R_{\text{proxy}}}\sqrt{1-r^2}$. We use $\sigma_{R_{\text{proxy}}} = 0.05$ in this environment, consistent with the ORPO setting. **Occ** denotes the total occupancy over state-action pairs unseen by $\pi_{\text{ref}}$. **Worst** refers to the expected worst-case reward computed under the training $r$ while excluding those unseen state-action pairs.

| $r$ | ORPO | | | | Max–Min | | |
|-----|-------|------|-------|-----|---------|-------|-----|
| | $\lambda$ | True | Worst | Occ | True | Worst | Occ |
| 0.1 | 0.050 | 0.46±0.14 | -6.08±0.07 | 8.22e-03±0.27e-03 | 0.13±0.15 | -1.34±0.05 | 1.36e-03±0.12e-03 |
| 0.2 | 0.049 | 0.66±0.03 | -6.88±0.08 | 2.8e-03±0.14e-03 | 7.79±0.10 | -0.72±0.05 | 2.18e-03±0.25e-03 |
| 0.3 | 0.048 | 0.70±0.03 | -7.55±0.04 | 1.87e-03±0.16e-03 | 7.68±0.11 | -0.96±0.04 | 1.85e-03±0.16e-03 |
| 0.4 | 0.046 | 0.16±0.02 | -9.20±0.06 | 1.51e-03±0.13e-03 | 8.24±0.10 | -1.37±0.06 | 1.01e-05±0.43e-05 |
| 0.5 | 0.043 | 0.52±0.08 | -6.53±0.05 | 0.028±0.0010 | 7.38±0.12 | -1.21±0.04 | 2.15e-03±0.37e-03 |
| 0.6 | 0.040 | 0.51±0.07 | -7.44±0.06 | 0.028±0.0011 | 6.65±0.17 | -1.22±0.06 | 1.18e-03±0.32e-03 |
| 0.7 | 0.035 | 0.84±0.12 | -5.85±0.07 | 0.027±0.0010 | 0.16±0.08 | -1.33±0.06 | 1.05e-02±0.10e-02 |
| 0.8 | 0.030 | 0.16±0.03 | -7.01±0.06 | 2.88e-03±0.19e-03 | 1.02±0.13 | -2.86±0.04 | 5.77e-04±0.35e-04 |
| 0.9 | 0.021 | 0.11±0.09 | -7.30±0.08 | 0.032±0.0009 | 0.37±0.13 | -5.49±0.12 | 1.29e-05±0.41e-05 |

Table 12: Evaluation in the Traffic environment across training-time correlation levels $r$ for ORPO and Max–Min. $\lambda$ denotes ORPO's coefficient, with $\lambda = \sigma_{R_{\text{proxy}}}\sqrt{1-r^2}$. We use $\sigma_{R_{\text{proxy}}} = 2e-4$ in this environment, consistent with the ORPO setting. **Occ** denotes the total occupancy over state-action pairs unseen by $\pi_{\text{ref}}$. **Worst** refers to the expected worst-case reward computed under the training $r$ while excluding those unseen state-action pairs.

| $r$ | ORPO | | | | Max–Min | | |
|-----|-------|------|-------|-----|---------|-------|-----|
| | $\lambda$ | True | Worst | Occ | True | Worst | Occ |
| 0.1 | 1.99e-4 | -1063.75±1.08 | 1.20e+03±0.02e+03 | 5.53e-03±0.03e-03 | -1428.21±5.36 | -3.43e+04±0.66e+04 | 2.29e-04±0.36e-04 |
| 0.2 | 1.95e-4 | -775.79±1.15 | 4.62e+04±0.04e+04 | 8.71e-04±0.03e-04 | -1312.67±9.14 | -2.83e+04±0.71e+04 | 1.49e-04±0.42e-04 |
| 0.3 | 1.91e-4 | -689.31±1.12 | -5.13e+04±0.04e+04 | 3.98e-04±0.02e-04 | -750.32±1.28 | -268.31±4.14 | 0.00±0.00 |
| 0.4 | 1.83e-4 | -1109.41±1.59 | -1.84e+03±0.04e+03 | 4.94e-03±0.05e-03 | -732.86±1.19 | -314.14±5.37 | 0.00±0.00 |
| 0.5 | 1.73e-4 | -673.45±1.36 | -1.51e+04±0.03e+04 | 6.85e-04±0.01e-04 | -1034.42±2.32 | -6168.40±124.75 | 0.00±0.00 |
| 0.6 | 1.60e-4 | -768.04±1.55 | -4.65e+04±0.04e+04 | 5.43e-03±0.03e-03 | -1322.74±3.92 | -6.73e+03±0.28e+03 | 4.50e-05±1.27e-05 |
| 0.7 | 1.43e-4 | -816.62±1.03 | -4.93e+04±0.04e+04 | 7.60e-03±0.03e-03 | -1398.63±2.73 | -3.82e+04±1.74e+04 | 2.38e-04±0.35e-04 |
| 0.8 | 1.20e-4 | -782.01±1.08 | -8.74e+04±0.09e+04 | 5.97e-03±0.02e-03 | -1359.23±2.08 | -4.94e+04±1.45e+04 | 3.48e-04±0.25e-04 |
| 0.9 | 8.72e-5 | -669.89±1.01 | -1.39e+04±0.04e+04 | 4.41e-03±0.03e-03 | -1337.41±2.45 | -9.33e+03±1.88e+02 | 9.66e-05±1.84e-05 |

## I  HOW TO CHOOSE $r$ IN PRACTICE?

When $r$ is unknown, both our method and ORPO lack a principled mechanism for selecting an appropriate value. Besides the simple heuristics derived from our experiments as discussed in Appendix H.2, we outline two potential approaches to this important problem below.

**Statistical inference of $r$.** If we have access to the true reward on a subset of state-action pairs, or if such labels can be acquired through active learning, we can estimate $r$ using the definition:

$$\mathbb{E}_{\mu_{\pi_{\text{ref}}}}\left[\left(\frac{R_{\text{proxy}} - J(\pi_{\text{ref}}, R_{\text{proxy}})}{\sigma_{R_{\text{proxy}}}}\right)\left(\frac{R_{\text{true}} - J(\pi_{\text{ref}}, R_{\text{true}})}{\sigma_{R_{\text{true}}}}\right)\right] = r, \tag{64}$$

Table 13: Evaluation in the Pandemic environment across training-time correlation levels $r$ for ORPO and Max–Min. $\lambda$ denotes ORPO's coefficient, with $\lambda = \sigma_{R_{\text{proxy}}}\sqrt{1 - r^2}$. We use $\sigma_{R_{\text{proxy}}} = 0.08$ in this environment, consistent with the ORPO setting. **Worst** refers to the expected worst-case reward computed under the training $r$ while excluding those unseen state-action pairs.

| $r$ | ORPO | | | Max–Min | |
|---|---|---|---|---|---|
| | $\lambda$ | True | Worst | True | Worst |
| 0.1 | 0.080 | -12.22$\pm$0.14 | -7.29e+06$\pm$0.05e+06 | -18.48$\pm$0.37 | -7.44e+04$\pm$0.19e+04 |
| 0.2 | 0.078 | -11.77$\pm$0.11 | -1.70e+07$\pm$0.10e+07 | -16.31$\pm$0.49 | -6.03e+04$\pm$0.12e+04 |
| 0.3 | 0.076 | -12.49$\pm$0.20 | -2.49e+06$\pm$0.05e+06 | -19.27$\pm$0.38 | -7.04e+04$\pm$0.05e+04 |
| 0.4 | 0.073 | -12.17$\pm$0.17 | -1.25e+06$\pm$0.05e+06 | -19.21$\pm$0.15 | -2.03e+03$\pm$0.16e+03 |
| 0.5 | 0.069 | -12.26$\pm$0.24 | -1.07e+06$\pm$0.04e+06 | -13.75$\pm$0.14 | -2.58e+03$\pm$0.15e+03 |
| 0.6 | 0.064 | -12.08$\pm$0.28 | -2.65e+06$\pm$0.09e+06 | -13.15$\pm$0.24 | -104.00$\pm$0.22 |
| 0.7 | 0.057 | -11.45$\pm$0.23 | -2.92e+05$\pm$0.10e+05 | -11.01$\pm$0.14 | -63.29$\pm$3.35 |
| 0.8 | 0.048 | -12.22$\pm$0.14 | -9.37e+05$\pm$0.06e+05 | -11.20$\pm$0.22 | -123.65$\pm$0.15 |
| 0.9 | 0.035 | -12.02$\pm$0.20 | -3.29e+04$\pm$0.09e+04 | -11.05$\pm$0.13 | -77.20$\pm$2.08 |

Table 14: Evaluation in the Glucose environment across training-time correlation levels $r$ for ORPO and Max–Min. $\lambda$ denotes ORPO's coefficient, with $\lambda = \sigma_{R_{\text{proxy}}}\sqrt{1 - r^2}$. We use $\sigma_{R_{\text{proxy}}} = 0.05$ in this environment, consistent with the ORPO setting. **Worst** refers to the expected worst-case reward computed under the training $r$ while excluding those unseen state-action pairs.

| $r$ | ORPO | | | Max–Min | |
|---|---|---|---|---|---|
| | $\lambda$ | True($\times 10^3$) | Worst | True($\times 10^3$) | Worst |
| 0.1 | 0.050 | -90.2$\pm$0.7 | -350.94$\pm$0.37 | -169.3$\pm$0.6 | -317.97$\pm$0.12 |
| 0.2 | 0.049 | -88.1$\pm$0.8 | -199.26$\pm$0.59 | -150.2$\pm$0.6 | -304.15$\pm$0.34 |
| 0.3 | 0.048 | -79.1$\pm$0.6 | -225.71$\pm$0.45 | -118.3$\pm$0.4 | -139.47$\pm$0.46 |
| 0.4 | 0.046 | -72.2$\pm$0.4 | -206.40$\pm$0.27 | -113.5$\pm$0.8 | -123.11$\pm$0.32 |
| 0.5 | 0.043 | -94.4$\pm$0.4 | -215.00$\pm$0.27 | -125.1$\pm$0.7 | -126.41$\pm$0.43 |
| 0.6 | 0.040 | -68.0$\pm$0.9 | -266.50$\pm$0.47 | -95.9$\pm$0.8 | -84.67$\pm$0.17 |
| 0.7 | 0.035 | -71.6$\pm$0.5 | -314.48$\pm$0.23 | -51.7$\pm$0.8 | -18.84$\pm$0.43 |
| 0.8 | 0.030 | -53.5$\pm$0.6 | -227.79$\pm$0.28 | -33.3$\pm$0.4 | -11.25$\pm$0.27 |
| 0.9 | 0.021 | -50.9$\pm$0.5 | -255.07$\pm$0.18 | -48.2$\pm$0.5 | -1.71$\pm$0.25 |

In fact, Equation 64 defines the Pearson correlation coefficient $r$ between the true reward $R_{\text{true}}$ and the proxy reward $R_{\text{proxy}}$ under the occupancy measure $\mu_{\pi_{\text{ref}}}$. Given a batch of $n$ state-action pairs $\{(s_i, a_i)\}_{i=1}^n$ sampled from $\pi_{\text{ref}}$ for which we have both $R_{\text{true}}^{(i)}$ and $R_{\text{proxy}}^{(i)}$, we can estimate this correlation using the sample correlation coefficient:

$$\hat{r} = \frac{\sum_{i=1}^n (R_{\text{true}}^{(i)} - \bar{R}_{\text{true}})(R_{\text{proxy}}^{(i)} - \bar{R}_{\text{proxy}})}{\sqrt{\sum_{i=1}^n (R_{\text{true}}^{(i)} - \bar{R}_{\text{true}})^2} \cdot \sqrt{\sum_{i=1}^n (R_{\text{proxy}}^{(i)} - \bar{R}_{\text{proxy}})^2}}$$

We can then use **Fisher's z-transformation** to compute the confidence intervals for $r$. After getting this bounded range, we can plug this bound into our framework to define a tighter reward uncertainty set. For example, we can use $r_{\text{lower}}$ for more pessimistic robustness. Or we can redefine the correlation constraint in Equation 64 to be bounded by both $r_{\text{lower}}$ and $r_{\text{upper}}$. The optimal solution under this new constraint can be similarly obtained using the approach in the paper.

**A min-max regret approach.** A more principled approach to addressing the uncertainty in $r$ may come from a regret-based perspective. Let $J_r(\pi)$ denote the worst-case return for a given policy $\pi$ under a specific correlation level $r$, i.e., $J_r(\pi) = \min_{R \in R_{\text{corr}}(r)} J(\pi, R)$. The regret can then be defined as $\text{Reg}(\pi, r) = \max_{\pi^*} J_r(\pi^*) - J_r(\pi)$, which quantifies the performance gap between the optimal policy under $r$ and the current policy. With this formulation, a robust objective

can be expressed as $\min_\pi \max_r \text{Reg}(\pi, r)$, aiming to find a policy that minimizes the worst-case regret across all possible values of $r$. This framework enables us to train policies that are robust to uncertainty in the correlation parameter $r$. We think this is a promising future direction, especially for cases where $r$ may be misspecified during training. As studied in (Sadek et al., 2025), minimax-regret may provide strong robustness guarantees under distribution shifts for $r$. In such settings, methods like Prioritized Level Replay (Jiang et al., 2021) and recent progress in (Monette et al., 2025) could be adapted to solve the problem by sampling multiple $r$ and solving Equation 27 in our paper. We should note that the reason these frameworks are potentially applicable is that our formulation admits a closed-form solution for the inner minimization. However, the main challenge lies in estimiating the occupancy measure. An interesting direction for future work is to investigate whether policy gradients can be approximated without explicitly occupancy estimation.

