# OpenReview forum: "Robust Optimization for Mitigating Reward Hacking with Correlated Proxies"
_ICLR.cc/2026/Conference — ICLR 2026 Poster_

### Official Review · Reviewer_46uB · 2025-10-26

**Soundness:** 3
**Presentation:** 3
**Contribution:** 3
**Rating:** 6
**Confidence:** 3

**Summary:**

This paper aims to design reinforcement learning (RL) agents that remain robust to imperfect or proxy reward signals. The authors formulate reward hacking as a robust policy optimization problem over all proxy rewards that are $r$-correlated with the true reward, deriving a tractable max–min formulation. When rewards are linear in known features, the method incorporates this structure to yield more interpretable and robust policies. Experiments across several environments show improved worst-case performance compared to existing approaches.

**Strengths:**

The paper is well written and clearly presented.

It addresses an important problem in reinforcement learning (RL) where the reward function is imperfect. There have also been recent papers on similar settings in language models, such as:
- [1] https://arxiv.org/abs/2504.03784
- [2] https://arxiv.org/abs/2405.11204
Considering the growing popularity of LLMs, it would be interesting to discuss the potential applications of this approach.

The proposed approach appears principled, and the theoretical results effectively support the empirical findings.

**Weaknesses:**

- Equations (9) and (13) do not include the importance sampling ratio, whereas Section 3.3 does. I think this is because the chi-square divergence involves this ratio—if so, the authors could clarify this more explicitly.

- The authors claim to have developed efficient algorithms, but the computational complexity of the proposed method is not discussed. Including empirical comparisons of runtime and memory usage against benchmarks would make the paper more complete.

- How are the feature mappings selected? Are the theoretical/empirical results sensitive to the choice or number of mappings?

- The experiments could be strengthened by comparing with recent related works, such as:
  - InfoRM: Mitigating Reward Hacking in RLHF via Information-Theoretic Reward Modeling
  - RRM: Robust Reward Model Training Mitigates Reward Hacking
  - Helping or Herding? Reward Model Ensembles Mitigate but do not Eliminate Reward Hacking

**Questions:**

Please see the weaknesses above.

---

> ### Author Response · Authors · 2025-11-21
> **Response to Reviewer 46uB (1/2)**
>
> Dear Reviewer 46uB:
>
> **We sincerely thank the reviewer for their thorough review and many helpful suggestions. We are glad that they found the proposed approach is principled and the theoretical results support the empirical findings. Below we address the concerns point-by-point.**
>
> >Q1: Equations (9) and (13) do not include the importance sampling ratio, whereas Section 3.3 does.
>
> A1: We thank the reviewer for pointing out this ambiguity. To solve the max-min / linear max-min objectives and derive Equations (9) and (13), importance sampling is not required. However, during implementation, several quantities appearing in these equations, such as the $\chi^2$ divergence in Equation 9 and $Q$ in Equation 13, must be estimated from samples. In these cases, importance sampling is necessary to estimate the occupancy under $\pi_{\text{ref}}$ using trajectories sampled from $\pi$ and vice versa. We have now clarified this distinction more explicitly in Appendix F.3, where we show how $Q$ is estimated using importance sampling.
>
> >Q2: The computational complexity of the proposed method is not discussed.
>
> A2: We agree that including empirical comparisons of runtime and memory usage against benchmarks would make the paper more complete. In fact, we have discussed the runtime and complexity of all algorithms across different environments in Appendix F.6 (Table 4). We have now added additional memory usage for all methods in Appendix F.6 Table 5. Regarding complexity, as clarified in Appendix F.6, despite the max-min structure, our method does not incur higher practical complexity compared to regularization-based approaches such as ORPO. Our Max-Min formulation admits a closed-form solution for the inner minimization problem, and the inner problem of our Linear Max-Min is smooth and well-posed, allowing efficient optimization with standard gradient-based methods. In fact, both ORPO and our methods rely on discriminator-based divergence estimation and perform comparable amounts of computation per iteration. The main difference lies in the structure of the objective, not in the asymptotic or empirical complexity. **In summary, ORPO does not inherently enjoy a complexity advantage over our Max-Min or Linear Max-Min algorithms.**
>
> Empirically, as shown in Tables 4 and 5, our Max-Min and Linear Max-Min implementations require more training time/memory compared to the original ORPO implementation because our methods need additional gradient steps used to more thoroughly train the discriminator network. However, as demonstrated in Section 4.2 (Figure 1), ORPO’s robustness and stability degrade significantly under varying reward specifications when the discriminator is not fully trained. This highlights that fully training the discriminator, although marginally more expensive, is essential for achieving robust performance.

---

> ### Author Response · Authors · 2025-11-21
> **Response to Reviewer 46uB (2/2)**
>
> >Q3: How are the feature mappings selected? Are the theoretical/empirical results sensitive to the choice or number of mappings?
>
> A3: To ensure a fair comparison, our experiments use **exactly the same feature mappings** as those employed in ORPO.
>
> To study the sensitivity of both our methods and ORPO to the choice and number of feature mappings, we conduct additional evaluations in Section 4.2 (Table 1) using a new metric, Linear Worst*. This metric evaluates each policy under a reward function where some features are **unseen** during training. Specifically, in the Traffic environment, the training reward includes features of "velocity", "acceleration", and "headway", while the evaluation reward replaces "velocity" with "commute time", resulting in a different feature configuration during testing. In the Pandemic environment, the training reward consists of "infection", "lower stage", and "smooth changes", whereas the evaluation reward adds an additional "political feature", testing robustness to an expanded feature set. As shown in Table 1, we observe minimal degradation in Max-Min policy’s performance, indicating its strong robustness to feature variation. In contrast, the performance of Linear Max-Min declines in this case, suggesting its advantage diminishes when prior assumptions about feature structure are inaccurate.  Nevertheless, both Max-Min and Linear Max-Min outperform ORPO across all evaluated settings, suggesting that **our algorithms are overall less sensitive to the specific choice or number of feature mappings compared to ORPO.**
>
> >Q4: The experiments could be strengthened by comparing with recent related works.
>
> A4: We thank the reviewer for this insightful suggestion. To further strengthen our experimental evaluation regarding the RLHF setting, we compared against “Helping or Herding? Reward Model Ensembles Mitigate but do not Eliminate Reward Hacking” (referred to as “Ensemble” for brevity). Specifically, we adopt their “finetune ensembles” method: we fine-tune five reward models on the AlpacaFarm preference dataset, all initialized from the same pre-trained Pythia-70M model but using five different random seeds, and aggregate their outputs using the mean rule. This setup is directly comparable to our RLHF configuration for ORPO and our methods, where both ORPO and our approach use a single fine-tuned Pythia-70M reward model. The resulst for the Ensemble baseline are reported in Table 1: *the true reward is 2.31±1.23, the proxy reward is 1.26±0.11 and the worst reward is -1.70±0.04.* These results show that the Ensemble baseline achieves only limited improvement in expected true return over the reference policy and attains a lower expected worst-case return than our method. These results indicate that using reward ensembles alone is insufficient to effectively mitigate reward hacking compared to our approach. However, such reward-centric methods, including InfoRM and RRM, can be easily integrated into our framework. In particular, these approaches can be used to construct a stronger proxy reward, which can then be plugged into our method to further improve performance.
>
>
> **Please consider these and we sincerely hope you will be able to reconsider your scores.**

---

> ### Author Response · Authors · 2025-11-28
> **Seeking Feedback on Our Rebuttal Responses**
>
> Dear reviewer 46uB:
>
> We would like to thank you again for your constructive comments and helpful suggestions. Since we are nearly at the end of the discussion phase, we would like to post a follow-up discussion.
>
> In our previous response, we have clarified the raised questions and made corresponding improvements in the updated paper. We would greatly appreciate it if you could review our detailed responses when you have a chance. If any clarifications remain insufficient or if you have further questions, we are happy to provide additional information.
>
> Thank you for your time and valuable guidance.
>
> Sincerely,
>
> Authors of Submission 2483

---

### Official Review · Reviewer_uNZH · 2025-10-30

**Soundness:** 2
**Presentation:** 2
**Contribution:** 2
**Rating:** 4
**Confidence:** 3

**Summary:**

The paper studies robustness to reward misspecification by assuming access to a proxy reward $R_{\text{proxy}}$ and a reference policy $\pi_{\text{ref}}$.
It defines an uncertainty set of true rewards whose Pearson correlation with the proxy reward under $\pi_{\text{ref}}$ equals $r$, and trains a policy to maximize worst-case return over that set.
A change-of-measure via the Radon--Nikodym derivative (density ratio) was used, which results in a regularized optimization objective the authors note "closely resembles" occupancy-regularized policy optimization (ORPO) objective, differing by a regularization scaling and an extra penalty term inside the square root.
Convergence guarantee of the proposed method is provided, and empirical evidence on four "real-world inspired" reward hacking environments is given.

**Strengths:**

The paper gives a clean derivation of a max-min robust RL objective under a correlation-constrained reward set and reduces it to a tractable form via duality and change of measure.
The resulting penalty can be read as an "orthogonalized" $\chi^2$ term, penalizing the component of occupancy shift not aligned with proxy improvement, which is a neat geometric refinement relative to ORPO's regularization.
The implementation details for ratio estimation and the comparison to ORPO will be useful to practitioners.

**Weaknesses:**

## Reward-space vs. behavior-space

First, the core uncertainty model is reward-space, not behavior-space.
Correlation under $\mu_{\pi_{\text{ref}}}$ ignores the well-known fact that many behaviorally equivalent rewards (e.g., potential-based shapings) can have arbitrary correlation with the proxy, while tiny changes in reward can flip the optimal policy near decision boundaries.
As a result, the method may penalize or ignore the wrong directions in practice; a behavior-centric formulation (e.g., safe improvement stated in terms of behaviorial metrics) would better match the problem that matters.

The paper repeatedly motivates "guarding against reward hacking," yet provides no behavioral safe-improvement guarantee and even asserts that the correction term "enforces robustness to potential reward hacking," which is not supported by a theorem.
Over-pessimistic and over-optimistic rewards can both lead to suboptimal behaviors, and the agent may just learn a different reward hacking behavior.

## Reference policy

Second, the entire construction is anchored to a reference policy.
How's it chosen? Does it need to be a good policy?
The correlation constraints are imposed under $\mu_{\pi_{\text{ref}}}$, and are vacuous off its support; the adversary may set arbitrarily bad rewards in unvisited regions.
Consequently, the guarantee weakens precisely where a newly learned policy might go.
The paper assumes $\pi_{\text{ref}}$ is given and does not offer a principled selection rule, beyond practical choices.

## Correlation hyperparameter

Third, $r$ is a hyperparameter that must be known or tuned. The authors state there is "no principled method" to pick it and resort to grid search over $r \in \\{0.1,\dots,0.9\\}$, which is unintuitive for a Pearson correlation meant to represent true-proxy alignment.
Results also depend on the training/evaluation $r$ mismatch.

## Succesor representation

Finally, the linear-reward assumption $R(s,a)=\theta^\top\phi(s,a)$ is long-established (**successor representations/features/measures**); the section omits direct attribution even though the subsequent optimization uses discounted feature expectations exactly as in that literature.

**Questions:**

In its current form, the paper offers a tidy derivation and a small geometric refinement of ORPO, but it does not resolve the central mismatch between reward-space uncertainty and behavior-space objectives, and it relies on an unintuitive correlation hyperparameter and a reference-policy anchor.
I encourage the authors to
- clarify the relationship between reward function correlation under a reference policy and behavior difference;
- clarify the implications of off-support vacuity and provide coverage-aware constraints; and
- justify or estimate $r$ more principledly.

---

> ### Author Response · Authors · 2025-11-21
> **Response to Reviewer uNZH (1/3)**
>
> Dear Reviewer uNZH:
>
> **We sincerely thank the reviewer for their very thorough review and are glad that our method is a neat geometric refinement of ORPO and is useful to practitioners. Below we address the concerns point-by-point.**
>
> >Q1: Clarify the relationship between reward function correlation under a reference policy and behavior difference.
>
> A1: We thank the reviewer for these insightful and critical points. Below we address this concern point by point.
> - $\textbf{Definition of reward hacking and how our method mitigates it.}$. We clarify that, as defined in Definition 4.2 of the ORPO paper and discussed in our preliminaries (Section 2), reward hacking occurs when $J(\pi, R_{\text{true}}) < J(\pi_{\text{ref}}, R_{\text{true}})$. Both ORPO and our work aim to prevent this phenomenon, namely, to ensure that optimizing a proxy reward leads to an improvement in the true reward over the reference policy. To further illustrate how our framework (in particular, Equation 9) helps guard against reward hacking, we formalize Theorem 1 in Section 3.1. From the theorem, we see that our objective optimizes a pessimistic lower bound on the true improvement over the reference policy. Although $J(\pi, R_{\text{true}}) - J(\pi_{\text{ref}}, R_{\text{true}})$ is precisely the quantity we would like to maximize, we cannot do so directly because the true reward is unobserved, and therefore we must instead optimize the max-min objective in Equation 9. Theorem 1 shows that our objective is always lower than (but anchored to) the true improvement, which explains why our framework can promote robustness against potential reward hacking: improving our surrogate objective necessarily improves a conservative bound on $J(\pi, R_{\text{true}}) - J(\pi_{\text{ref}}, R_{\text{true}})$. Moreover, in our experiments, we empirically verify that our method does not exhibit reward hacking across different environments and remains robust across a wide range of correlation levels, supporting our claim of “guarding against reward hacking.”
>
> - $\textbf{We model uncertainty in reward space for studying reward hacking.}$ We acknowledge that many behaviorally equivalent rewards (e.g., potential-based shaping) can have arbitrary correlation with the proxy, and our uncertainty set will treat such rewards as different even though they are behaviorally indistinguishable. A behavior-centric formulation would indeed be more invariant to shaping. While there is substantial work on risk-sensitive objectives or risk constraints in RL that operate in behavior space, these are orthogonal to our setting. We explicitly study reward hacking/misspecification, so the uncertainty must be defined in **reward** space rather than behavior space. We therefore view the correlation constraint as a modeling assumption on a particular reward representation chosen by the designer (e.g., the proxy), rather than on abstract equivalence classes under potential-based shaping.
> Designing an ambiguity set that is explicitly defined over behaviorally equivalent reward classes, for example, via shaping-invariant reward metrics, would be an interesting and complementary direction, but it is orthogonal to our current contribution and would significantly complicate the optimization problem. To the best of our knowledge, such behavior-centric formulations have been largely ignored in the robust/safe RL and reward-hacking literature and represent an appealing avenue for future work. Importantly, however, our guarantees remain valid even in the presence of potential-based shaping: although behaviorally equivalent versions of the true reward can have different correlations with the proxy, as long as **one** behaviorally equivalent representative lies in our uncertainty set, our framework and Theorem 1 still apply, since potential-based shaping does not change the value of the return difference $J(\pi, R_{\text{true}}) - J(\pi_{\text{ref}}, R_{\text{true}})$.
>
> - $\textbf{Tiny reward changes and their effect on expected return.}$ We also acknowledge that tiny changes in the reward can indeed flip the optimal policy near decision boundaries. However, this does not undermine our goal of optimizing expected return. Even though small perturbations in the reward may change which action is optimal, they typically do not significantly affect the return, since the value function is Lipschitz continuous with respect to the reward function, a standard result in MDP theory (Theorem 12 in [1]).
>
> - $\textbf{Linear Max-Min to mitigate over-pessimistic rewards.}$ As stated in Section 3.2, we acknowledge that our Max-Min framework, where we ignore any prior knowledge of the reward function, can be overly pessimistic. To solve this, we introduce the Linear Max-Min variant, which leverages the linear structure of the reward function to eliminate implausible reward functions.
>
> [1] Csáji, B. C., & Monostori, L. "Value function based reinforcement learning in changing Markovian environments." JMLR, 2008.

---

> ### Author Response · Authors · 2025-11-21
> **Response to Reviewer uNZH (2/3)**
>
> >Q2: Clarify the implications of off-support vacuity and provide coverage-aware constraints.
>
> A2: We thank the reviewer for raising this important point. We acknowledge that, as in ORPO, our construction is anchored to a reference policy $\pi_{\text{ref}}$, which is assumed to be given. In principle, $\pi_{\text{ref}}$ can be any policy for the environment. Both our method and ORPO view $\pi_{\text{ref}}$ as a (suboptimal) baseline, and the goal is to optimize a proxy reward in a way that improves the true reward relative to this baseline. Thus, **the reference policy does not need to be optimal.** The guarantees are explicitly stated relative to $\pi_{\text{ref}}$.
>
> However, as we discuss in Section 3.1 and as the reviewer correctly notes, the correlation constraints are imposed under $\mu_{\pi_{\text{ref}}}$ and off the support of $\pi_{\text{ref}}$ the adversarial reward can indeed be arbitrarily bad. To mitigate this, we train an auxiliary discriminator network to identify regions that are rarely visited by $\pi_{\text{ref}}$, and then penalize deviation from the occupancy of $\pi_{\text{ref}}$, effectively discouraging the learned policy from venturing too far into regions that are unsupported by the reference policy. We agree that this weakens guarantees precisely in regions that a newly learned policy might explore, and this is a limitation shared by both our method and ORPO.
>
> This observation suggests a more “principled” preference over reference policies: we should favor $\pi_{\text{ref}}$ with **sufficient coverage of the state-action space**, so that the adversarial freedom in off-support regions is limited. In practice, this is more likely when $\pi_{\text{ref}}$ is derived from rich **expert demonstrations or from stochastic/exploratory training** (e.g., entropy-regularized policies or policies trained with exploration bonuses). In our experiments (Appendix F.4), the reference policies for the Traffic, Pandemic, and Glucose environments are obtained by behavioral cloning on large, diverse trajectories generated by human experts or hand-crafted controllers, which cover a wide range of task-relevant behaviors and induce broad state-action occupancy. For Tomato, the reference policy is trained with an exploration-bonus objective, again encouraging wider coverage.
>
> More broadly, we agree that developing a fully principled selection rule for $\pi_{\text{ref}}$ is an interesting open problem. One natural direction is to select among candidate reference policies that **maximize a coverage criterion**, such as the entropy of the induced state-action occupancy. While we do not instantiate such a procedure in this work, we view this as an important direction for future work. We would also like to clarify that, as with ORPO, our framework assumes a given reference policy. Designing or learning such a policy is an important but orthogonal problem and not the primary focus of this work.

---

> ### Author Response · Authors · 2025-11-21
> **Response to Reviewer uNZH (3/3)**
>
> >Q3: Justify or estimate $r$ more principledly.
>
> A3: We agree that $r$ is defined as the Pearson correlation intended to capture the alignment between the true and proxy rewards. Our statement that there is “no principled method” to select $r$ is based on the assumption that **we do not have access to the true reward function**. Without ground-truth rewards, we cannot estimate the exact correlation $r$ and thus must treat it as a hyperparameter and rely on grid search. This also naturally leads to a potential mismatch between the value of $r$ used during training and the true correlation at evaluation time. There is currently no principled method for selecting the optimal $r$ for either our method or ORPO, and this issue is a common challenge in the broader robust RL literature.
>
> However, if we can assume access to the true reward on a subset of state-action pairs, it becomes possible to estimate the Pearson correlation directly using standard statistical techniques, as discussed in Appendix I. In that case, the estimated $r$ can be plugged into our framework to provide a more informed choice.
>
> In fully unsupervised settings where such information is unavailable, we also investigate how to choose a reasonable $r$ empirically.  Empirically, we find that **moderate values of  $r$ tend to work well.** In Appendix H.2 (Tables 8, 9), we observe that values of $r$ in the range of 0.3 to 0.4 generally yield robust policies. As extremely small $r$ values tend to be overly conservative, while high $r$ values overly trust the proxy reward. Therefore, in the absence of prior knowledge of $r$, starting with a moderate $r$ is a practical heuristic.
>
>
> >Q4: The linear-reward assumption is long-established (successor representations/features/measures); the section omits direct attribution even though the subsequent optimization uses discounted feature expectations exactly as in that literature.
>
> A4: We thank the reviewer for pointing out this omission. Our linear reward assumption and the use of discounted feature expectations are indeed closely related to the literature on successor features. We now explicitly acknowledge this connection in the Introduction, Section 3.2 and Related Work in Appendix D.
>
> **Please consider these and we sincerely hope you will be able to reconsider your scores.**

---

> ### Author Response · Authors · 2025-11-28
> **Seeking Feedback on Our Rebuttal Responses**
>
> Dear reviewer uNZH:
>
> We would like to thank you again for your constructive comments and helpful suggestions. Since we are nearly at the end of the discussion phase, we would like to post a follow-up discussion.
>
> In our previous response, we have clarified the raised questions and made corresponding improvements in the updated paper. We would greatly appreciate it if you could review our detailed responses when you have a chance. If any clarifications remain insufficient or if you have further questions, we are happy to provide additional information.
>
> Thank you for your time and valuable guidance.
>
> Sincerely,
>
> Authors of Submission 2483

---

### Official Review · Reviewer_CXL3 · 2025-11-03

**Soundness:** 3
**Presentation:** 3
**Contribution:** 3
**Rating:** 6
**Confidence:** 3

**Summary:**

Building off Laidlaw et al (2025)'s definition of "r-correlated" proxy reward functions, the authors introduce a robust optimization framework aimed at reducing reward hacking. Their method focuses on maximizing the minimum reward achievable across all possible r-correlated reward functions. This leads to a loss function that, while similar to that of Laidlaw et al, is distinct in its exact regularization term. Across several environments, they show that their approach achieves higher worst-case reward, compared to ORPO from Laidlaw et al, while typically maintaining similar expected reward.

**Strengths:**

The paper has several strengths:
- Intuitive conceptual contribution of optimizing worst-case reward over all reward functions that are r-correlated to the given proxy reward function. This seems like the natural progression from Laidlaw et al (2025).
- Theoretically sound framework. They apply several tools from robust optimization to derive a tractable loss function, which though ultimately similar to Laidlaw et al (2025), differs in the exact regularization term, and they show this leads to improved worst-case rewards in their experiments
- Improvement for linear reward setting. They are able to derive a refinement for the case where the reward function is known to be linear.
- Better occupancy measure estimates lead to also improving ORPO. Both this work and Laidlaw et al (2025) use a discriminator network to measure the occupancy measure divergence between a given policy and the reference policy. The authors note that in the original ORPO implementation, the discriminator network is not fully optimized, and thus, the resultant policies still end up visiting states that are low frequency under the original policy. By training the discriminator for longer, they also obtain improved results compared to the original ORPO implementation.
- Extensive experiments and ablations. The authors test their method on five environments that were designed for studying reward hacking, and include several additional results in the appendices (e.g. results with unnormalized rewards, results on robustness to "r", etc)

**Weaknesses:**

One paragraph that I thought was a bit strange was:
> "We adopted a similar approach used by ORPO (Laidlaw et al., 2025). For each environment, we first performed a grid search over several different values of r, and for each fixed r, we trained the policy using our algorithm. We then selected the rvalue that leads to the policy with the best expected worst-case return ... Notice that ORPO selects the optimal r that yields the best expected return under the true reward, which is infeasible in practice when the true reward is unknown during training. In contrast, our approach for choosing r is more practical."

It's true that the authors' approach is computable without the true reward unlike the process used in ORPO, but it's not much more practical, as without knowing the "correct" value of r, it's not clear which measure of the worst-case reward is most meaningful.  Therefore, I would hesitate to describe this approach as more "practical" than ORPO. Instead, it might be more helpful for the authors to directly reference the two practical methods for selecting r discussed in Appendix I.

By the way, the authors say they include results for all r that they considered in H.5, but then H5 only includes results for the Traffic and Tomato environments. Also strangely, they consider r \in {0.3, 0.5, 0.9} for Traffic and r \in {0.1, 0.4, 0.7, 0.9} for Tomato. Why do they not use the same grid for all environments? Also, it is somewhat strange that the grid for Traffic is not uniform either. Can the authors please include results for all environments?

The authors' process for selecting "r" also highlights one aspect of their framework that I found counter-intuitive. The set R_corr(r) as defined in Eq 4 includes all reward functions that are *exactly* r-correlated with the proxy. Apriori, I would have expected them to define this set as including all reward functions that are *at least* r-correlated with the proxy. Then, increasing r would monotonically increase the maximal worst-case reward over the functions in R_corr(r). As I understand it, with the current defn, this kind of monotonicity does not necessarily hold? And that is also why in the selection process for r that they use in the experiments (noted above), the highest r is not always picked (though, noise in the occupancy measure estimation and policy training process may also affect this empirically)? It seems odd that, under this framework, selecting r = 0.3 makes the policy robust to reward functions with exactly 0.3 correlation, but not to those with higher correlation, such as 0.6. Could the authors clarify this design choice?

**Questions:**

- Why is the same grid for r not used in all environments?
- Why did the authors not define R_corr as all reward functions that are *at least* r-correlated to the proxy, rather than *exactly* r-correlated to the proxy?
- How would one obtain the "reference policy" in practice?

---

> ### Author Response · Authors · 2025-11-21
> **Response to Reviewer CXL3 (1/3)**
>
> Dear Reviewer CXL3:
>
> **We sincerely thank the reviewer for their thorough review and many helpful suggestions. We appreciate that they found our paper has several strengths, including its intuitive conceptual contribution, solid theoretical foundation, and extensive experiments and ablations. Below we address the concerns point-by-point.**
>
> >Q1: I would hesitate to describe this approach as more "practical" than ORPO.
>
> A1: We agree that when the exact correlation $r$ is unknown, our approach also raises concern about how to interpret which worst-case reward is actually meaningful. We have revised the paragraph accordingly to soften the claim of being “more practical” and now explicitly acknowledge this limitation. In addition, we now directly reference the two practical methods for selecting $r$ discussed in Appendix I in this paragraph. We thank the reviewer for highlighting this concern.

---

> ### Author Response · Authors · 2025-11-21
> **Response to Reviewer CXL3 (2/3)**
>
> >Q2: Why is the same grid for r not used in all environments?
>
> A2: We sincerely apologize for the confusion regarding whether the same grid of $r$ values was used across all environments. We now include the **complete grid search results**, including those for the remaining two environments, in Appendix H.5.  For illustration, we also present the full results for $\textbf{Tomato}$ and $\textbf{Glucose}$ here. Each table presents the mean and standard deviation of the expected true reward, expected worst-case reward, and occupancy measure achieved by the learned policy over five random seeds. We also include the corresponding ORPO results for a fair comparison.
>
> | r   | ORPO λ  | ORPO True        | ORPO Worst        | ORPO Occ               | Max–Min True      | Max–Min Worst      | Max–Min Occ               |
> |-----|---------|------------------|-------------------|------------------------|-------------------|--------------------|---------------------------|
> | 0.1 | 0.050   | 0.46±0.14        | -6.08±0.07        | 8.22e-03±0.27e-03      | 0.13±0.15         | -1.34±0.05         | 1.36e-03±0.12e-03         |
> | 0.2 | 0.049   | 0.66±0.03        | -6.88±0.08        | 2.80e-03±0.14e-03      | 7.79±0.10         | -0.72±0.05         | 2.18e-03±0.25e-03         |
> | 0.3 | 0.048   | 0.70±0.03        | -7.55±0.04        | 1.87e-03±0.16e-03      | 7.68±0.11         | -0.96±0.04         | 1.85e-03±0.16e-03         |
> | 0.4 | 0.046   | 0.16±0.02        | -9.20±0.06        | 1.51e-03±0.13e-03      | 8.24±0.10         | -1.37±0.06         | 1.01e-05±0.43e-05         |
> | 0.5 | 0.043   | 0.52±0.08        | -6.53±0.05        | 2.80e-02±1.00e-03      | 7.38±0.12         | -1.21±0.04         | 2.15e-03±0.37e-03         |
> | 0.6 | 0.040   | 0.51±0.07        | -7.44±0.06        | 2.80e-02±1.10e-03      | 6.65±0.17         | -1.22±0.06         | 1.18e-03±0.32e-03         |
> | 0.7 | 0.035   | 0.84±0.12        | -5.85±0.07        | 2.70e-02±1.00e-03      | 0.16±0.08         | -1.33±0.06         | 1.05e-02±1.00e-02         |
> | 0.8 | 0.030   | 0.16±0.03        | -7.01±0.06        | 2.88e-03±0.19e-03      | 1.02±0.13         | -2.86±0.04         | 5.77e-04±0.35e-04         |
> | 0.9 | 0.021   | 0.11±0.09        | -7.30±0.08        | 3.20e-02±0.90e-03      | 0.37±0.13         | -5.49±0.12         | 1.29e-05±0.41e-05         |
>
> Evaluation in the $\textbf{Tomato}$ environment is shown above.
>
> | r   | ORPO λ | ORPO True (×10³)   | ORPO Worst        | Max–Min True (×10³)   | Max–Min Worst      |
> |-----|--------|--------------------|-------------------|-----------------------|--------------------|
> | 0.1 | 0.050  | -90.2 ± 0.7        | -350.94 ± 0.37    | -169.3 ± 0.6          | -317.97 ± 0.12     |
> | 0.2 | 0.049  | -88.1 ± 0.8        | -199.26 ± 0.59    | -150.2 ± 0.6          | -304.15 ± 0.34     |
> | 0.3 | 0.048  | -79.1 ± 0.6        | -225.71 ± 0.45    | -118.3 ± 0.4          | -139.47 ± 0.46     |
> | 0.4 | 0.046  | -72.2 ± 0.4        | -206.40 ± 0.27    | -113.5 ± 0.8          | -123.11 ± 0.32     |
> | 0.5 | 0.043  | -94.4 ± 0.4        | -215.00 ± 0.27    | -125.1 ± 0.7          | -126.41 ± 0.43     |
> | 0.6 | 0.040  | -68.0 ± 0.9        | -266.50 ± 0.47    | -95.9 ± 0.8           | -84.67 ± 0.17      |
> | 0.7 | 0.035  | -71.6 ± 0.5        | -314.48 ± 0.23    | -51.7 ± 0.8           | -18.84 ± 0.43      |
> | 0.8 | 0.030  | -53.5 ± 0.6        | -227.79 ± 0.28    | -33.3 ± 0.4           | -11.25 ± 0.27      |
> | 0.9 | 0.021  | -50.9 ± 0.5        | -255.07 ± 0.18    | -48.2 ± 0.5           | -1.71 ± 0.25       |
>
> Evaluation in the $\textbf{Glucose}$ environment is shown above.
>
> Note that the original ORPO algorithm performs a grid search over $\lambda$, where $\lambda = \sigma_{R_{\text{proxy}}} \sqrt{1 - r^2}$ and $\sigma_{R_{\text{proxy}}}$ is the standard deviation of $R_{\text{proxy}}$ under $\pi_{\text{ref}}$, which is generally unknown and must be estimated. Thus, ORPO searches over $\lambda$, whereas our method searches directly over $r$. For a fair comparison, we estimate $\sigma_{R_{\text{proxy}}}$ and use it to map each value of $r$ in our grid to a corresponding $\lambda$ for ORPO. However, across all environments we find that the resulting $\lambda$ values occupy a much narrower scale than $r$: while $r$ spans the full range from $0.1$ to $0.9$, the induced $\lambda$ values are confined to a small interval (e.g., approximately $0.021$–$0.05$ in $\textbf{Tomato}$ and $\textbf{Glucose}$). As a consequence, the ORPO policies change only marginally across the mapped $r$ values, and their expected true and worst-case returns appear similar in the tables. In practice, ORPO would need to search over a broader range of $\lambda$ values. By contrast, the expected true and worst-case returns for our Max-Min method vary meaningfully across the full span of $r$. This highlights that, in practice, our method and ORPO naturally operate on different hyperparameter scales when tuning their respective robustness parameters.

---

> ### Author Response · Authors · 2025-11-21
> **Response to Reviewer CXL3 (3/3)**
>
> >Q3: Why did the authors not define R_corr as all reward functions that are at least r-correlated to the proxy, rather than exactly r-correlated to the proxy?
>
> A3: We thank the reviewer for this insightful observation. We agree that, from an intuitive standpoint, it is natural to define the uncertainty set as containing all rewards that are at least $r$ correlated with the proxy. In fact, our derivations continue to hold under this definition. The closed-form solution we obtain for the inner minimization always corresponds to a reward lying on the boundary of the “$\ge r$” set. Thus, switching from “=” to “$\ge$” in the definition of the uncertainty set **does not mathematically change the form of the inner minimizer or the resulting worst-case value.**
>
> Regarding monotonicity, there are two different things here. For a fixed policy, our Equation 9 gives the expected worst-case return for that policy as a function of $r$, and it is easy to show it is monotone in $r$.  However, in the implementation, we fix a training $r$ and train a policy to optimize Equation 9, where the trained policy also depends on $r$. There is no theoretical reason for the maximum worst-case reward (Equation 9) to be monotone in $r$ as we do not know the relationship between the trained policy and $r$. Changing $r$ changes the optimization landscape and thus the learned policy and worst-case reward. That is why the highest $r$ is not necessarily the best and we still need to search over $r$ in practice. Noise in occupancy estimation and training only adds more non-monotonicity, but the core reason is that the policy changes with $r$, not just the inner minimization. **In summary, it is correct that although the worst-case reward is monotonic in $r$ under a fixed policy (as evident from Equation 9), the maximum worst-case reward is generally not.**
>
> >Q4: How would one obtain the "reference policy" in practice?
>
> A4: We thank the reviewer for this important practical question. In practice, a reference policy can be obtained using **inverse reinforcement learning methods**, such as behavioral cloning from expert demonstrations. For example, in our experiments, the reference policies for the Traffic, Pandemic, and Glucose environments are trained via behavioral cloning on large, diverse trajectories generated by human experts or hand-crafted controllers.
>
> Alternatively, one can use **stochastic or exploratory policies** (e.g., entropy-regularized policies or policies trained with exploration bonuses) as the reference policy. For instance, the reference policy used for Tomato is trained with an exploration-bonus.
>
> These reference policies are standard and relatively simple to obtain, while still exhibiting rich task-relevant behaviors, making them a suitable and informative starting point for further improvement by our Max-Min and Linear Max-Min methods. We would also like to clarify that, as with ORPO, our framework assumes a given reference policy. Designing or learning such a policy is an important but orthogonal problem and not the primary focus of this work.
>
> **Please consider these and we sincerely hope you will be able to reconsider your scores.**

---

> > ### Comment · Reviewer_CXL3 · 2025-11-25
> > **Confusion on Appendix E.2**
> >
> > I thank the authors for their response and for providing the full grid search results.
> >
> > > Thus, switching from “=” to “$\ge$” in the definition of the uncertainty set does not mathematically change the form of the inner minimizer or the resulting worst-case value.
> >
> > This would intuitively make sense to me, however, I was reading Appendix E more closely regarding the derivation of the inner minimizer, and I now have a confusion. In Appendix E.2, the authors use stationarity and feasibility as sufficient conditions for global optimality although the problem has a nonconvex feasible set. Could the authors elaborate?

---

> > > ### Author Response · Authors · 2025-11-25
> > > **Response to Reviewer CXL3**
> > >
> > > Dear Reviewer CXL3:
> > >
> > > We sincerely appreciate your time to carefully review our paper and rebuttal. For the concern:
> > > >In Appendix E.2, the authors use stationarity and feasibility as sufficient conditions for global optimality although the problem has a nonconvex feasible set. Could the authors elaborate?
> > >
> > > We agree that in general, stationarity and feasibility are not sufficient for global optimality when the feasible set is nonconvex. Our statement at the end of Appendix E.2 is **relying on the structure of our specific inner problem**.
> > >
> > > Recall the inner minimization problem discussed in Appendix E.2, and we work in the Hilbert space $H=L^2(\mu_{\text{ref}})$. Recall our normalization assumptions: $E_{\mu_{\pi_{\text{ref}}}}[R_{\text{proxy}}]=0$ and $E_{\mu_{\pi_{\text{ref}}}}[R^2_{\text{proxy}}]=1$ . Then the constraints can be rewritten as inner products in $H$:
> > >
> > > $<R,1>=M$ (mean constraint),
> > >
> > > $<R,R_{\text{proxy}}>=rV$ (correlation constraint),
> > >
> > > $||R||^2=V^2+M^2$ (norm constraint).
> > >
> > > Let {$e_0, e_1, e_2, …$} be an orthonormal basis of $H$, where $e_0$ is proportional to the constant function of $1$, $e_1=R_{\text{proxy}}$ ($e_0$ and $e_1$ is orthonormal because $E_{\mu_{\pi_{\text{ref}}}}[R_{\text{proxy}}]=0$) and {$e_k$}, $k\ge 2$ spans the orthogonal complement of span {$1, R_{\text{proxy}}$}. Expanding
> > >
> > > $R=\alpha_0 e_0 + \alpha_1 e_1 +\sum_k \alpha_k e_k$.
> > >
> > > Notice that the mean constraint and correlation constraints uniquely fix $\alpha_0$ and $\alpha_1$. The norm constraint then forces: $\sum_k \alpha_k = \rho^2$ for some constant radius $\rho >0$. Hence the remaining degrees of freedom lie on a sphere in the subspace orthogonal to $1$ and $R_{\text{proxy}}$. This is to say, although $R_{\text{corr}}$ is not convex in the ambient space, it is a spherical manifold (the boundary of an $L^2$-ball intersected with an affine subspace), which is compact and smooth.
> > > Moreover, the objective is linear in $R$:
> > >
> > > $E_{\mu_{\pi_{\text{ref}}}} [L \cdot R]=<L, R>=\text{constant}+<L’, R’>$,
> > >
> > > where $L’$ is the projection of $L=\mu_\pi/\mu_{\pi_{\text{ref}}}$ onto the subspace spanned by {$e_k$} and $R’=\sum \alpha_k e_k$. Therefore the optimization reduces to
> > >
> > > $min_{||R’||=\rho} <L’,R’>$.
> > >
> > > This is simply minimizing a linear function over a Euclidean sphere. In this setting, it is well-known that the only stationary points of a linear functional on a sphere are its global maximum and global minimum. There are no other local minima or saddle points. Thus, on this particular nonconvex feasible set, any feasible stationary point is automatically a global optimizer.
> > > Our Lagrangian analysis in Appendix E.2 does exactly this:
> > > 1. For fixed ($\lambda_1,\lambda_2,\lambda_3$) with $\lambda_3 < 0$, the Lagrangian is a strictly convex quadratic in $R$, so its stationary point $R^\star(\lambda)$ is the unique global minimizer of the inner problem with those multipliers.
> > >
> > >
> > > 2. Solving the dual and enforcing feasibility recovers the specific choice of multipliers $\lambda^\star$ for which $R^\star(\lambda^\star)$ lies on the sphere defined by the norm constraint.
> > >
> > > 3. Because the reduced problem is linear over a sphere, this feasible stationary point $R^\star(\lambda^\star)$ must be the global minimizer of the original inner problem.
> > >
> > > **Please consider these and we sincerely hope you will be able to reconsider your scores.**

---

> > > > ### Comment · Reviewer_CXL3 · 2025-11-26
> > > >
> > > > I thank the authors for their reply and highly recommend that they include this explanation in the paper itself for clarity.
> > > >
> > > > With regards to the monotonicity point that the authors brought up in their previous response, I am not sure I follow still. Define $f(\pi, r)$ as the maximand in Eq 9. The authors say that, for a fixed policy $\pi$, the function $f$ is monotonically increasing in the correlation $r$. However, they claim that the worst-case maximal reward is not necessarily monotonic because the policy also depends on $r$. In other words, letting $r_1 < r_2$, it is not necessarily the case that $f(\pi^{\ast}(r_1), r_1) \leq f(\pi^\ast(r_2), r_2)$ where $\pi^*(r)$ is the optimal policy for the correlation $r$.
> > > >
> > > > However, consider the following:
> > > > - We have that $f(\pi^\ast(r_1), r_1) \leq f(\pi^\ast(r_1), r_2)$ because, as the authors said, for a fixed policy, $f$ is monotonically increasing in $r$
> > > > - Then, $f(\pi^\ast(r_1), r_2) \leq f(\pi^*(r_2), r_2)$ because $\pi^\ast(r)$ is optimal for $r$
> > > > - Therefore, $f(\pi^\ast(r_1), r_1) \leq f(\pi^*(r_2), r_2)$
> > > >
> > > > So, it seems to me that the maximal worst-case return $g(r) = f(\pi^\ast(r), r)$ should still be monotonic in $r$. Please let me know if I am mistaken somewhere.

---

> > > > > ### Author Response · Authors · 2025-11-26
> > > > > **Response to Reviewer CXL3**
> > > > >
> > > > > Dear Reviewer CXL3:
> > > > >
> > > > > We sincerely appreciate your careful follow-up and the time you took to think through our problem. We fully agree with the reviewer that, if for every fixed policy $\pi$ the objective $f(\pi,r)$ were monotone non-decreasing in $r$, then the maximal value $g(r)$ would also be monotone. We also thank the reviewer for the clear derivation, which we had not previously worked out in this way. However, in our experiments, both ORPO and our method do not exhibit monotonicity.
> > > > >
> > > > > After carefully re-examining Equation 9, we realized that our previous response was imprecise: we stated that $f(\pi,r)$ is monotone in $r$ for a fixed $\pi$, but this is not true in full generality.
> > > > >
> > > > > For clarity, let us rewrite Equation 9 (setting $V=1,M=0$ as in the implementation):
> > > > >
> > > > >  $f(\pi, r)=r m_\pi - \sqrt{1-r^2}\sqrt{\chi^2(\mu_\pi||\mu_{\pi_{\text{ref}}})-m_\pi^2}$,
> > > > >
> > > > > where $m_\pi=E_{\mu_\pi}[R_{\text{proxy}}]$ and we denote
> > > > >
> > > > > $a_\pi=\sqrt{\chi^2(\mu_\pi||\mu_{\pi_{\text{ref}}})-m_\pi^2}$.
> > > > >
> > > > > For a fixed policy $\pi$, $m_\pi$ and $a_\pi$ are constants, so we can write
> > > > >
> > > > > $f(\pi,r)=r m_\pi-a_\pi\sqrt{1-r^2}$, $r\in(0,1)$.
> > > > >
> > > > > Taking the derivative with respect to $r$ gives:
> > > > >
> > > > > $\frac{\partial f}{\partial r}=m_\pi+\frac{a_\pi r}{\sqrt{1-r^2}}$
> > > > >
> > > > > If $m_\pi\ge0$, then $\frac{\partial f}{\partial r}\ge0$ for all $r\in(0,1)$, so $f(\pi,r)$ is monotone non-decreasing.
> > > > >
> > > > > However, if $m_\pi<0$, the first term is negative and the second term is positive. In this case the derivative can be negative for small $r$ and positive for larger $r$, so $f(\pi,r)$ can decrease and then increase and is not globally monotone.
> > > > >
> > > > > Thus, monotonicity of $f(\pi,r)$ in $r$ only holds under an additional condition such as $m_\pi=E_{\mu_\pi}[R_{\text{proxy}}]\ge0$. We overlooked this condition in our earlier response. In practice, a policy can indeed have negative expected normalized proxy reward. Consequently, the maximal worst-case return is not required to be monotone in $r$ in general, and our empirical observation that neither ORPO nor our method exhibits clear monotonicity in $r$ is consistent with the theory.
> > > > >
> > > > > We sincerely apologize for our earlier incorrect statement and for the resulting confusion. We have also revised the paper to include a clearer explanation of the global optimality argument. We thank the reviewer again for the valuable and constructive feedback, which has deepened our understanding of this problem.

---

### Official Review · Reviewer_b81V · 2025-11-05

**Soundness:** 3
**Presentation:** 4
**Contribution:** 3
**Rating:** 6
**Confidence:** 3

**Summary:**

This paper addresses the issue of reward hacking in reinforcement learning, where an agent may optimize for an imperfect proxy of the reward function which could lead the agent to diverge from the intended true objective. To tackle this, the authors take a robust approach to optimize for the feature weights of the true underlying reward from an uncertainty set if feature weights defined by $\chi^2$-divergence. This allows them to model this as a max-min optimization problem over the proxy rewards which are constrained via correlation with the true rewards of the system, which the authors then extend to linear rewards. Ultimately, they prove convergence of their algorithm and sample complexity bounds on the occupancy estimation.

**Strengths:**

- The paper is relevant and highly motivated by practical implementation. It builds upon the state of the art to improve upon interpretability and derive a tractable objective for the non-convex robust optimization problem. The paper is very clear and progresses fluidly.
- A closed-form solution is proposed and derived for the worst-case reward feature vector of the adversary by transforming this vector into a whitened version of itself, $\tilde{\phi}$, such that the $Q$-function becomes the identity matrix.
- From what I could determine, the proposed method is backed up by a strong theoretical analysis. Subsequently, the authors provided much empirical validation of this theory to verify their claims as well as sufficient detail to reproduce these experiments.

**Weaknesses:**

- In practice the correlation between a proxy and rewards, $r$, is unknown. The authors briefly mention this in the appendix and use a grid search to find this. However, as the author's mention, there is not a principled method for selecting the optimal $r$ and thus it may not scale.
- An assumption is made that the true rewards lie within the defined uncertainty set. In practice, this may not always be the case. The proposed reference policy may not provide sufficient coverage of the feature space which could lead to the problem.

**Questions:**

How could this extend to other uncertainty sets?

Minor things:
- $\gamma$ should be defined prior to equation 1 where you define the objects in the MDP tuple.
- It would be nice to see one of the algorithms appear in the main text as well as the convergence bounds stated formally in section 3.3 to help highlight the contribution of this work.
- Missing space between "latent" and "(true)" on line 1026.

---

> ### Author Response · Authors · 2025-11-21
> **Response to Reviewer b81V (1/2)**
>
> Dear Reviewer b81V:
>
> **We sincerely thank the reviewer for the thoughtful and detailed review and appreciate that they found our paper to be highly motivated by practical implementation and backed up by a strong theoretical analysis.  Below we address the concerns point-by-point.**
>
> >Q1: In practice the correlation between a proxy and rewards, $r$, is unknown. There is not a principled method for selecting the optimal $r$ and thus it may not scale.
>
> A1: We agree that there is currently no principled method for selecting the optimal $r$ for either our method or ORPO, and this is a common challenge in the broader robust RL literature. While grid search over $r$ is a general and scalable strategy across environments, it can indeed become computationally expensive in complex settings.
>
> Empirically, we find that **moderate values of  $r$ tend to work well.** In Appendix H.2 (Tables 8, 9), we observe that values of $r$ in the range of 0.3 to 0.4 generally yield robust policies. Extremely small $r$ values tend to be overly conservative, while high $r$ values overly trust the proxy reward. Therefore, in the absence of prior knowledge of $r$, starting with a moderate $r$ is a practical heuristic.
>
> Besides the simple heuristics derived from our experiments, we also outline **two potential approaches** to this important problem in Appendix I. The first is a **statistical estimation approach**: if one has access to the true reward on a subset of state-action pairs, it is possible to estimate the correlation $r$ directly. The second is a **regret-based approach**, where we define regret as the performance gap between the optimal policy under $r$ and the current policy. The goal is to find a policy that minimizes the worst-case regret across all possible values of $r$. This framework enables us to train policies that are robust to uncertainty in the correlation parameter $r$.
>
>
>
>
> >Q2: An assumption is made that the true rewards lie within the defined uncertainty set. In practice, this may not always be the case.
>
> A2: We agree that, in practice, when the exact correlation $r$ between the proxy and the true reward is unknown, choosing a larger $r$ can violate the assumption that the true reward lies within the defined uncertainty set. Although, in theory, one can always choose a smaller $r$ to enlarge the uncertainty set so that it contains the true reward, this may yield an overly conservative worst-case reward and degrade the learned policy’s performance.
>
> To better understand the performance of both our approach and ORPO under such $r$ misspecification, in Section 4.2 (Figure 1) and Appendix H.3 (Figure 5), we present an extensive comparison under a wide range of testing $r$ values, using the fixed set of policies trained with the $r$ value specified in the paper. Specifically, for each $r$ during testing, we sample 1000 reward functions that satisfy the correlation constraint and report the average return of the learned policy under these sampled rewards. These experiments capture **two** forms of misspecification: (i) training with $r$ that is too large (the true reward falls outside the uncertainty set) and (ii) training with $r$ that is too small (the uncertainty set is overly large). As shown in Figures 1 and 5, our methods (Max-Min and Linear Max-Min) consistently outperform ORPO across nearly all values of $r$ tested, demonstrating that even when the training $r$ is misspecified, **our algorithms achieve more robust and stable performance under such uncertainty.**

---

> ### Author Response · Authors · 2025-11-21
> **Response to Reviewer b81V (2/2)**
>
> >Q3: The proposed reference policy may not provide sufficient coverage of the feature space which could lead to the problem.
>
> A3: We agree that the reference policy should provide sufficient coverage of the feature space, which is crucial to our Linear Max-Min algorithm. Intuitively, a useful reference policy should avoid collapsing to trivial or deterministic behavior and instead induce diverse feature visitation so that learning can improve upon it. Theoretically, as discussed in Appendix E.5, whitening the reward features requires the matrix $Q$ to be non-singular, i.e., the features visited by reference policy must span the full feature space. This is more likely when the reference policy is derived from either **expert demonstrations** that exhibit rich behavior or from **stochastic or exploratory policies** (e.g., entropy-regularized policies or policies trained with exploration bonuses).  In our experiments (Appendix F.4), the reference policies for the Traffic and Pandemic environments are trained via behavioral cloning on large, diverse trajectories generated by human experts or hand-crafted controllers. These demonstrations cover a wide range of task-relevant behaviors, and the induced reference policy has the occupancy spans a high-dimensional subspace of the feature space. We empirically verified that the resulting $Q$ matrices in our experiments are full-rank and numerically well-conditioned. Therefore, **the reference policy in our experiment provides sufficient coverage of the feature space.**
>
> We acknowledge that ensuring sufficient coverage of the feature space by the reference policy is generally challenging in practice. In cases where $Q$ is technically ill-conditioned because certain features are only rarely visited, a common remedy is to introduce a small ridge regularization term and replace $Q$ with $Q_\lambda=Q+\lambda I$ for $\lambda>0$ and $I$ being the identity matrix. This shifts the spectrum of $Q$ away from zero, improving the conditioning of the whitening step and enhancing numerical stability. The trade-off is that the whitening constraint then holds only approximately, and the resulting optimization becomes slightly more conservative in directions with low visitation, an effect that is often desirable from a robustness perspective.
>
>
> >Q4: How could this extend to other uncertainty sets?
>
> A4: We thank the reviewer for this insightful question. In robust RL, most uncertainty sets considered in the literature are centered around a known proxy reward, typically of the form $R_{\text{cent}}=${$R: f(R-R_{\text{proxy}})  \leq 0$}, and max-min formulations over such sets have been extensively studied [1]. In contrast, our uncertainty set is defined through a correlation constraint with the known proxy reward, which can be viewed as having the form $R_{\text{corr}}=${$R: f(R\times R_{\text{proxy}})\leq 0$}, where the proxy enters multiplicatively via its correlation with $R$. For the specific correlated uncertainty set defined in the paper, we are able to derive a closed-form solution. However, to the best of our knowledge, developing general-purpose methods for such correlation-based uncertainty sets in robust RL remains an open problem. Our recent analysis suggests that, under additional assumptions, a suitable change of measure on the policy might allow one to transform certain correlated uncertainty sets $R_{\text{corr}}$ into centered ones $R_{\text{cent}}$, thereby enabling the use of existing robust RL tools. We view this as an interesting direction for future work.
>
> [1] Esther Derman et al. "Robustness and regularization in reinforcement learning." NeurIPS 2023 Workshop on Generalization in Planning. 2023.
>
> >Q5: Minor things
>
> A5: We sincerely thank the reviewer for pointing out these minor issues. We have added the definition of $\gamma$, the discount factor, when introducing the MDP tuple. We also fully agree that including one of the algorithms and the convergence bounds in the main text helps to better highlight the contribution of our work. Accordingly, we now formally state the sample complexity result for the discriminator and present a simplified version of our Max-Min algorithm in Section 3.3. We have also corrected the missing space between “latent” and “(true)” on line 1026.
>
> **Please consider these and we sincerely hope you will be able to reconsider your scores.**

---

> ### Author Response · Authors · 2025-11-28
> **Seeking Feedback on Our Rebuttal Responses**
>
> Dear reviewer b81V:
>
> We would like to thank you again for your constructive comments and helpful suggestions. Since we are nearly at the end of the discussion phase, we would like to post a follow-up discussion.
>
> In our previous response, we have clarified the raised questions and made corresponding improvements in the updated paper. We would greatly appreciate it if you could review our detailed responses when you have a chance. If any clarifications remain insufficient or if you have further questions, we are happy to provide additional information.
>
> Thank you for your time and valuable guidance.
>
> Sincerely,
>
> Authors of Submission 2483

---

### Author Response · Authors · 2025-12-02
**Summary of rebuttals**

Dear AC:

Thank you once again for your time and effort in reviewing our paper and for providing insightful, constructive feedback. We would like to take this opportunity to summarize our work and formally conclude the author-reviewer discussion.

Our paper proposes a robust optimization framework for mitigating reward hacking. Specifically, we maximize performance under the worst-case proxy reward that is consistent with a correlation constraint. We derive a tractable max-min objective, and further extend it to linear rewards, improving both robustness and interpretability. From the initial reviews, we received supportive feedback from Reviewers b81V, CXL3, and 46uB, and we especially appreciate the detailed and constructive comments from Reviewer uNZH.

To summarize, all reviewers highlighted several strengths of our work, including:

- The paper is strongly motivated by the important problem of reward hacking and provides a natural progression beyond prior work such as ORPO.

- The paper is well written and clearly presented.

- The proposed method is backed up by a strong theoretical analysis, and the empirical findings effectively support the claims.


One major concern raised by reviewer uNZH, the only reviewer who gave a negative score, is that “The relationship between reward function correlation under a reference policy and behavior difference is unclear.” We address this concern through the following key observations.

- First, we deliberately model uncertainty in **reward space** rather than behavior space because our goal is to **study reward misspecification**, which is inherently defined in terms of discrepancies between proxy and true rewards. While directly modeling uncertainty in behavior space is an interesting direction, it is orthogonal to the focus of this work.

- Second, Theorem 1 shows that improving our objective (Equation 9) necessarily **improves a conservative bound on the true-reward improvement over the reference policy**, thereby guarding against reward hacking under our definition.
- Third, our framework and Theorem 1 remain valid under potential-based reward shaping, as noted by the reviewer, as long as **at least one** behaviorally equivalent true reward lies within our uncertainty set.

- Fourth, while small perturbations in the reward function may alter the optimal policy near decision boundaries, this does not undermine our goal of maximizing expected return. Moreover, our Linear Max-Min formulation explicitly mitigates implausible reward instantiations within the Max-Min framework.

Taken together, we believe these points adequately address the reviewer’s concern.

Other common concerns raised by reviewers are how to select the correlation parameter $r$ and the reference policy $\pi_{\text{ref}}$ in practice. Regarding $r$, we acknowledge that there is no fully principled way to choose this parameter, a limitation shared by our method, ORPO, and the broader robust RL literature. Empirically, we observe that **moderate values of $r$** perform well. We also outline two possible directions for choosing $r$: **a statistical estimation approach** and **a regret-based approach**. For the reference policy, we argue that a desirable $\pi_{\text{ref}}$ should provide sufficiently broad coverage over the state-action space. In practice, such a policy can be obtained via **inverse reinforcement learning** from expert demonstrations or from a **stochastic/exploratory training procedure**.

During the rebuttal period, we also provided detailed responses on several technical points, including: robustness of our method under misspecified $r$ and different choices/number of features; the global optimality of our solution; and the non-monotonicity properties of our objective. In addition, we presented further experimental results, including a full grid search over $r$ across all environments, empirical comparisons of runtime and memory usage, and comparisons with recent reward hacking methods in RLHF. We have updated the paper to incorporate all the above results. We believe these additions substantially strengthen the paper and **fully address all reviewer concerns**. We are particularly grateful for the exchange with Reviewer CXL3, which helped us refine our understanding of the non-monotonicity behavior of our objective.


Once again, we sincerely thank all reviewers and the AC for their valuable feedback throughout the review process.

Sincerely,

Authors of Submission 2483

---

### Meta-Review · Area_Chair_Tveq · 2025-12-22

**Summary:**

**Paper Summary**: This paper considers a robust policy optimization problem where the paper considers robustness across the reward is considered in order to avoid reward hacking.  In particular, compared to the ORPO, this paper derives a tighter lower bound to characterize the difference between the true reward and the proxy reward, and then it proposes an algorithm to maximize the lower bound.  The paper also shows theoretically the above for the linear model.  Experiments across several environments show that the algorithms consistently outperform ORPO in worst-case returns, and offer improved robustness and stability across different levels of proxy–true reward correlation.

**Reviewers' Concerns**: Reviewers appreciated the contributions. They also raised a few concerns. The main concerns were (i) knowing the value of $r$, the correlation term, (ii) the computational complexity, (iii) the reference policy, and (iv) the reward space and behavioral space robustness. The authors have provided responses to all the questions.

**AC's take**: According to the AC, most of the concerns have been addressed apart from selecting the value of $r$. While this is a drawback, however, this is also true for other works as well. Further, the authors indeed provided a good robust heuristic to choose the value of $r$. Hence, the AC feels that the contributions are enough to be accepted. I would suggest the authors to include all the results and the discussions.

**Reviewer Concerns:**

Reviewers appreciated the contributions. They also raised a few concerns. The main concerns were (i) knowing the value of $r$, the correlation term, (ii) the computational complexity, (iii) the reference policy, and (iv) the reward space and behavioral space robustness. The authors have provided responses to all the questions.

According to the AC, most of the concerns have been addressed apart from selecting the value of $r$. While this is a drawback, however, this is also true for other works as well. Further, the authors indeed provided a good robust heuristic to choose the value of $r$.

**Reviewer Scores:**

Only one reviewer's score is on the negative side, however, the AC feels that the authors have sufficiently addressed the concerns of that reviewer.

---

### Decision · Program_Chairs · 2026-01-26

Accept (Poster)